# Theoretical Understanding of Learning from Adversarial Perturbations

**Soichiro Kumano**
The University of Tokyo
kumano@cvm.t.u-tokyo.ac.jp

**Hiroshi Kera**
Chiba University
kera@chiba-u.jp

**Toshihiko Yamasaki**
The University of Tokyo
yamasaki@cvm.t.u-tokyo.ac.jp

## Abstract

It is not fully understood why adversarial examples can deceive neural networks and transfer between different networks. To elucidate this, several studies have hypothesized that adversarial perturbations, while appearing as noises, contain class features. This is supported by empirical evidence showing that networks trained on mislabeled adversarial examples can still generalize well to correctly labeled test samples. However, a theoretical understanding of how perturbations include class features and contribute to generalization is limited. In this study, we provide a theoretical framework for understanding learning from perturbations using a one-hidden-layer network trained on mutually orthogonal samples. Our results highlight that various adversarial perturbations, even perturbations of a few pixels, contain sufficient class features for generalization. Moreover, we reveal that the decision boundary when learning from perturbations matches that from standard samples except for specific regions under mild conditions. The code is available at https://github.com/s-kumano/learning-from-adversarial-perturbations.

## 1 Introduction

It is well known that a small malicious perturbation, or an adversarial perturbation, can change a classifier's prediction from the correct class to an incorrect class (Szegedy et al., 2014). An interesting observation by Ilyas et al. (2019) has shown that a neural network, trained on adversarial examples labeled by such incorrect classes, can generalize to correctly labeled test samples. Specifically, the training procedure is as follows:[1]

**Definition 1.1** (Learning from adversarial perturbations (later redefined) (Ilyas et al., 2019)). Let $\mathcal{D} := \{(\boldsymbol{x}_n, y_n)\}_{n=1}^N$ be a training dataset, where $\boldsymbol{x}_n$ denotes an input (e.g., an image) and $y_n$ denotes the corresponding label. Let $f$ be a classifier trained on $\mathcal{D}$. For each $n$, an adversarial example $\boldsymbol{x}_n^{\mathrm{adv}}$ is produced by imposing an adversarial perturbation on $\boldsymbol{x}_n$ to increase the probability for a target label $y_n^{\mathrm{adv}} \neq y_n$ given by $f$, constructing $\mathcal{D}^{\mathrm{adv}} := \{(\boldsymbol{x}_n^{\mathrm{adv}}, y_n^{\mathrm{adv}})\}_{n=1}^N$. Training a classifier from scratch on $\mathcal{D}^{\mathrm{adv}}$ is called *learning from adversarial perturbations*.

Notably, a training sample $\boldsymbol{x}^{\mathrm{adv}}$ appears almost identical to $\boldsymbol{x}$ to humans, but is always labeled as a different class $y^{\mathrm{adv}} \neq y$ (e.g., an adversarially perturbed, yet semantically unchanged, horse image with a cat label). Counterintuitively, networks can learn to classify correctly labeled test samples (e.g., a cat image with a cat label) from such mislabeled adversarial samples.

The unexpected success of training on mislabeled adversarial examples suggests that adversarial perturbations may contain label-aligned features. For example, frog adversarial images labeled as horses have perturbations that appear as noises to humans but contain horse features. This feature

---

[1]This is neither adversarial training nor training with (partially) noisy labels (cf. Appendix A).

hypothesis not only explains the counterintuitive generalization of learning from perturbations, but also sheds light on several puzzling phenomena of adversarial examples, such as why adversarial examples can fool classifiers and transfer among them (cf. Section 2).

While the feature hypothesis is considered a potential explanation for adversarial perturbations, a theoretical understanding of this hypothesis and its empirical foundation, learning from perturbations, remains limited. For example, it is still unknown how adversarial perturbations contain class features. The similarity in decision boundaries between the network trained on clean and mislabeled adversarial samples is also unknown.

In this study, we provide the first theoretical validation of the learnability from adversarial perturbations. Using recent results on the decision boundary of a one-hidden-layer neural network trained on mutually orthogonal samples (Frei et al., 2023), we show that perturbations contain sufficient class features for generalization. In addition, we demonstrate that the decision boundary when learning from perturbations is consistent with that from natural samples except for specific regions under mild conditions. While Ilyas et al. (2019) empirically considered only the case where $L_2$-constrained perturbations were superimposed on natural data, our theory covers broader settings. We reveal that even sparse perturbations, perturbations of a few pixels, contain class features and enable generalization. Moreover, we show that the alignment of decision boundaries derived from adversarial perturbations and natural samples becomes stronger when perturbations are superimposed on random noises. The main contributions are summarized as follows:

- We theoretically justified the feature hypothesis and learning from perturbations with one-hidden-layer neural networks trained on mutually orthogonal training samples.
- We showed that various adversarial perturbations including sparse perturbations can be represented by the weighted sum of benign training samples. This suggests that perturbations, yet apparently uninterpretable, contain sufficient class features for generalization.
- We demonstrated that classifiers learning from mislabeled adversarial samples produce consistent predictions with those from clean samples under mild conditions. Moreover, classifiers trained on random noises with perturbations provide accurate predictions for natural data even though the classifiers do not see any natural data during training.

## 2 RELATED WORK

Ilyas et al. (2019) first claimed that an adversarial perturbation contains class features, called non-robust features. These features are highly predictive and generalizable, yet brittle and incomprehensible to humans. This idea is supported by neural networks that learn from perturbations (cf. Definition 1.1) achieving good test accuracies on standard datasets (Ilyas et al., 2019). The hypothesis that perturbations contain features elucidates several phenomena of adversarial examples (Szegedy et al., 2014). For example, models misclassify adversarial examples because they react to features in perturbations. Transferability (Szegedy et al., 2014; Goodfellow et al., 2015) is also explained: different models respond to the same features in perturbations. Moreover, adversarially robust models focus on robust (semantic) features and ignore highly predictive non-robust (non-semantic) features, providing insights into the trade-off between robustness and accuracy (Su et al., 2018; Tsipras et al., 2019; Raghunathan et al., 2019; 2020; Yang et al., 2020), the human-aligned image gradients of robust models (Engstrom et al., 2019b; Kaur et al., 2019; Santurkar et al., 2019; Tsipras et al., 2019; Augustin et al., 2020), and the enhanced transfer learning ability of robust models (Aggarwal et al., 2020; Salman et al., 2020; Deng et al., 2021; Utrera et al., 2021).

Subsequent studies deepened the discussion of non-robust features. While Ilyas et al. (2019) considered robust and non-robust features separately, Springer et al. (2021b) claimed their potential entanglement. Some studies have attempted to separate robust and non-robust features using the information bottleneck (Kim et al., 2021) and neural tangent kernel (Tsilivis & Kempe, 2022). Engstrom et al. (2019a) provided a broad discussion about robust neural style transfer and feature leakage. Other studies have used the feature hypothesis to generate highly transferable adversarial examples (Springer et al., 2021a;b;c), understand the behavior of batch normalization in adversarial training (Benz et al., 2021), and degrade the robustness of adversarially trained models (Tao et al., 2022). However, the nature of adversarial perturbations as class features and the theoretical explanation for the counterintuitive success of learning from perturbations remain unclear.

In this study, we justify learning from perturbations, which is an essential foundation of the feature hypothesis. Our results support the above studies based on the feature hypothesis. We do not consider whether adversarial perturbations are robust or non-robust. We discuss the nature of perturbations as class features and why classifiers can obtain generalization ability from perturbations.

## 3 PRELIMINARY

### 3.1 SETTINGS

**Notations.** For a positive integer $n \in \mathbb{N}$, let $[n] := \{1, \ldots, n\}$. For a vector $\boldsymbol{x} \in \mathbb{R}^d$, we denote the Euclidean norm by $\|\boldsymbol{x}\|$. We use $\Omega(\cdot)$, $\Theta(\cdot)$, and $\mathcal{O}(\cdot)$ for the standard big Omega, big Theta, and big O notation. To hide polylogarithmic factors, we use $\tilde{\Omega}(\cdot)$, $\tilde{\Theta}(\cdot)$, and $\tilde{\mathcal{O}}(\cdot)$.

**Network.** Our network settings follow Frei et al. (2023). Let $f : \mathbb{R}^d \to \mathbb{R}$ be a one-hidden-layer neural network. The number of hidden neurons is even and is denoted by $m$. We assume that the hidden layer is trainable and the last layer is frozen to constant weights $\boldsymbol{a} \in \mathbb{R}^m$. The first half elements of $\boldsymbol{a}$ are $1/\sqrt{m}$ and the latter half are $-1/\sqrt{m}$. Let $\boldsymbol{W} := (\boldsymbol{w}_1, \ldots, \boldsymbol{w}_m)^\top \in \mathbb{R}^{m \times d}$ be the weights of the hidden layer. Let $\phi(z) := \max(z, \gamma z)$ be the element-wise leaky ReLU for a constant $\gamma \in (0, 1)$. Namely, $f(\boldsymbol{x}) := \boldsymbol{a}^\top \phi(\boldsymbol{W}\boldsymbol{x})$. The assumption that the positive and negative values of $\boldsymbol{a}$ are equal is introduced for notational simplicity and is fundamentally unnecessary. In Appendix G, we derive some results without this assumption.

**Training.** Let $\mathcal{D} := \{(\boldsymbol{x}_n, y_n)\}_{n=1}^N \subset \mathbb{R}^d \times \{\pm 1\}$ be a training dataset. With the exponential loss $\ell(z) = \exp(-z)$ or logistic loss $\ell(z) = \ln(1 + \exp(-z))$, a loss over $\mathcal{D}$ is defined as $\mathcal{L}(\boldsymbol{W}; \mathcal{D}) := \sum_{n=1}^N \ell(y_n f(\boldsymbol{x}_n; \boldsymbol{W}))/N$. The network parameters are updated by gradient flow, gradient descent with an infinitesimal step size, as $\mathrm{d}\boldsymbol{W}(t)/\mathrm{d}t = -\nabla_{\boldsymbol{W}} \mathcal{L}(\boldsymbol{W}(t); \mathcal{D})$, where $t \geq 0$ is a continuous training step. Finally, we summarize the training setting.

**Setting 3.1** (Training). Consider training a one-hidden-layer neural network $f$ on a dataset $\mathcal{D}$. The network parameter $\boldsymbol{W}$ is updated by minimizing the exponential or logistic loss over the dataset, $\mathcal{L}(\boldsymbol{W}; \mathcal{D})$, using gradient flow. The training runs for a sufficiently long time, i.e., $t \to \infty$.

**Learning from Adversarial Perturbations.** We formalize and extend Definition 1.1.

**Definition 3.2** (Learning from adversarial perturbations). Let $\mathcal{D} := \{(\boldsymbol{x}_n, y_n)\}_{n=1}^N \subset \mathbb{R}^d \times \{\pm 1\}$ be a training dataset. Let $f : \mathbb{R}^d \to \mathbb{R}$ be a one-hidden-layer neural network trained on $\mathcal{D}$ following Setting 3.1. Let $N^{\mathrm{adv}} \in \mathbb{N}$ be the number of samples with perturbations. For each $n \in [N^{\mathrm{adv}}]$, a perturbed sample $\boldsymbol{x}_n^{\mathrm{adv}} \in \mathbb{R}^d$ targeting a new label $y_n^{\mathrm{adv}} \in \{\pm 1\}$ is produced as either:

(a) (Perturbation on natural sample) $N^{\mathrm{adv}} = N$ and $\boldsymbol{x}_n^{\mathrm{adv}} := \boldsymbol{x}_n + \boldsymbol{\eta}_n$.
(b) (Perturbation on uniform noise) $\boldsymbol{x}_n^{\mathrm{adv}} := \boldsymbol{X}_n + \boldsymbol{\eta}_n$, where $\boldsymbol{X}_n$ is sampled from the uniform distribution $U([-1, 1]^d)$. For any $n \neq k$, $\boldsymbol{X}_n$ and $\boldsymbol{X}_k$ are independent. The target label $y_n^{\mathrm{adv}}$ is randomly sampled from $\{\pm 1\}$.

The perturbation $\boldsymbol{\eta}_n$ is given by an adversarial attack. Training a classifier from scratch on a new dataset $\{(\boldsymbol{x}_n^{\mathrm{adv}}, y_n^{\mathrm{adv}})\}_{n=1}^{N^{\mathrm{adv}}}$ following Setting 3.1 is called *learning from adversarial perturbations*.

This definition has two differences from Definition 1.1. First, we added a noise data case. In this scenario, we can justify learning from perturbations under milder conditions than in the natural sample scenario. Second, we removed the restriction of $y_n^{\mathrm{adv}} \neq y_n$ to consider broader cases. In addition to the uniform noise case, we examine the sub-Gaussian noise case in Appendix F.

### 3.2 DECISION BOUNDARY OF ONE-HIDDEN-LAYER NEURAL NETWORK

To understand learning from perturbations, we employ the following result on the implicit bias of gradient flow (Frei et al., 2023).

**Theorem 3.3** (Rearranged from Frei et al. (2023)). *Let $\{(\boldsymbol{x}_n, y_n)\}_{n=1}^N \subset \mathbb{R}^d \times \{\pm 1\}$ be a training dataset. Let $R_{\max} := \max_n \|\boldsymbol{x}_n\|$, $R_{\min} := \min_n \|\boldsymbol{x}_n\|$, and $p_{\max} := \max_{n \neq k} |\langle \boldsymbol{x}_n, \boldsymbol{x}_k \rangle|$.*

*A one-hidden-layer neural network $f : \mathbb{R}^d \to \mathbb{R}$ is trained on the dataset with Setting 3.1. If $\gamma^3 R_{\min}^4/(3N R_{\max}^2) \geq p_{\max}$, then $\mathrm{sgn}(f(z)) = \mathrm{sgn}(f^{\mathrm{bdy}}(z))$ holds with $t \to \infty$, where $f^{\mathrm{bdy}}(z) := \sum_{n=1}^{N} \lambda_n y_n \langle x_n, z \rangle$ and $\lambda_n \in \left(\frac{1}{2R_{\max}^2}, \frac{3}{2\gamma^2 R_{\min}^2}\right)$ for every $n \in [N]$.*

[Appendix B](#) provides a more detailed background. This theorem claims that the binary decision of $f(z)$ equals that of the linear function $f^{\mathrm{bdy}}(z)$; namely, $f(z)$ has a linear decision boundary. This theorem requires training samples to be nearly orthogonal, which is a common property of high-dimensional data. Although this theorem is not related to learning from perturbations, we utilize it to easily observe the decision boundary of learning from perturbations as follows:

**Corollary 3.4** (Decision Boundary when learning from perturbations). *Let $\{(x_n^{\mathrm{adv}}, y_n^{\mathrm{adv}})\}_{n=1}^{N^{\mathrm{adv}}}$ be a mislabeled training dataset with adversarial perturbations (cf. Definition 3.2). Let $R_{\max}^{\mathrm{adv}} := \max_n \|x_n^{\mathrm{adv}}\|$, $R_{\min}^{\mathrm{adv}} := \min_n \|x_n^{\mathrm{adv}}\|$, and $p_{\max}^{\mathrm{adv}} := \max_{n \neq k} |\langle x_n^{\mathrm{adv}}, x_k^{\mathrm{adv}} \rangle|$. A one-hidden-layer neural network $f$ is trained on the dataset with Setting 3.1. If $\gamma^3 R_{\min}^{\mathrm{adv}\,4}/(3N^{\mathrm{adv}} R_{\max}^{\mathrm{adv}\,2}) \geq p_{\max}^{\mathrm{adv}}$, then $\mathrm{sgn}(f(z)) = \mathrm{sgn}(f_{\mathrm{adv}}^{\mathrm{bdy}}(z))$ holds with $t \to \infty$, where $f_{\mathrm{adv}}^{\mathrm{bdy}}(z) := \sum_{n=1}^{N^{\mathrm{adv}}} \lambda_n^{\mathrm{adv}} y_n^{\mathrm{adv}} \langle x_n^{\mathrm{adv}}, z \rangle$ and $\lambda_n^{\mathrm{adv}} \in \left(\frac{1}{2R_{\max}^{\mathrm{adv}\,2}}, \frac{3}{2\gamma^2 R_{\min}^{\mathrm{adv}\,2}}\right)$ for every $n \in [N]$.*

## 4 THEORETICAL RESULTS

### 4.1 PERTURBATION AS CLASS FEATURES

Recall that a one-hidden-layer network trained on orthogonal samples with Setting 3.1 has a linear decision boundary (cf. Theorem 3.3). We focus on adversarial attacks on this boundary rather than the network itself, called geometry-inspired attacks (Moosavi-Dezfooli et al., 2016; Croce & Hein, 2020), which simplify notation.[2] Let $\epsilon > 0$ be the perturbation constraint. A geometry-inspired adversarial example $x_n^{\mathrm{adv}}$ maximizes $y_n^{\mathrm{adv}} f^{\mathrm{bdy}}(x_n^{\mathrm{adv}})$ under $\|x_n^{\mathrm{adv}} - x_n\| \leq \epsilon$ as follows:[3]

$$x_n^{\mathrm{adv}} := x_n + \eta_n, \qquad \eta_n := \epsilon y_n^{\mathrm{adv}} \frac{\nabla_{x_n} f^{\mathrm{bdy}}(x_n)}{\|\nabla_{x_n} f^{\mathrm{bdy}}(x_n)\|} = \epsilon y_n^{\mathrm{adv}} \frac{\sum_{k=1}^{N} \lambda_k y_k x_k}{\|\sum_{k=1}^{N} \lambda_k y_k x_k\|}. \tag{1}$$

Without loss of generality, we consider a single step of the gradient calculation because multiple steps also produce the same perturbation form due to the linearity of $f^{\mathrm{bdy}}$. The perturbation $\eta_n$ is represented as a weighted sum of the training samples. Since the training samples $\{x_n\}_{n=1}^{N}$ are nearly orthogonal (i.e., $x_n$ and $x_k$ do not negate each other for $n \neq k$), the perturbation contains class features of the training samples. This observation supports the hypothesis that adversarial perturbations contain class features, enabling generalization from them.

### 4.2 LEARNING FROM ADVERSARIAL PERTURBATIONS ON NATURAL SAMPLES

We consider learning from geometry-inspired perturbations on natural samples. Appendix C provides the proofs of the theorems. Using Corollary 3.4, the decision boundary can be derived as:

**Theorem 4.1** (Decision boundary when learning from geometry-inspired perturbations on natural samples). *Let $f$ be a one-hidden-layer neural network trained on geometry-inspired perturbations on natural samples (cf. Eq. (1) and Definition 3.2(a)) with Setting 3.1. If $N > C^2/R_{\min}^2$ and*

$$\frac{\gamma^3 (R_{\min}^2 - 2\frac{C}{\sqrt{N}}\epsilon + \epsilon^2)^2}{3N(R_{\max}^2 + 2\frac{C}{\sqrt{N}}\epsilon + \epsilon^2)} - 2\frac{C}{\sqrt{N}}\epsilon - \epsilon^2 \geq p_{\max} \tag{2}$$

---

[2]In Appendix E, we discuss geometry-inspired $L_0$ and $L_\infty$ attacks and a gradient-based $L_2$ attack that targets the network itself, such as fast gradient sign method (Goodfellow et al., 2015) and projected gradient descent (Madry et al., 2018).

[3]Geometry-inspired perturbations Eq. (1) can be defined only if $\lambda_n$ can be determined by Eq. (A11). If training samples satisfy $\gamma^3 R_{\min}^4/(3N R_{\max}^2) \geq p_{\max}$, Eq. (A11) is always consistent; thus, the definition is valid. Note that Eq. (A11) is consistent for some sample sets that do not satisfy $\gamma^3 R_{\min}^4/(3N R_{\max}^2) \geq p_{\max}$ (cf. the proof of Lemma C.1).

*with* $C := \frac{3R_{\max}^4 + \gamma^3 R_{\min}^4}{\gamma^2 R_{\min}^3 \sqrt{1-\gamma}}$, *then, with* $t \to \infty$, *the decision boundary of* $f$ *is given by*

$$f_{\mathrm{adv}}^{\mathrm{bdy}}(\boldsymbol{z}) := \underbrace{\frac{\sum_{n=1}^N \lambda_n^{\mathrm{adv}} y_n^{\mathrm{adv}} \langle \boldsymbol{x}_n, \boldsymbol{z} \rangle}{\sum_{n=1}^N \lambda_n^{\mathrm{adv}}}}_{\text{Effect of learning from mislabeled natural samples}} + \underbrace{\epsilon \frac{f^{\mathrm{bdy}}(\boldsymbol{z})}{\| \sum_{n=1}^N \lambda_n y_n \boldsymbol{x}_n \|}}_{\text{Effect of learning from perturbations}} . \quad (3)$$

The decision boundary, Eq. (3), includes two components that explain the effects of mislabeled samples and geometry-inspired perturbations. The sign of the first term is determined by the sum of the weighted inner products, $\sum_{n=1}^N \lambda_n^{\mathrm{adv}} y_n^{\mathrm{adv}} \langle \boldsymbol{x}_n, \boldsymbol{z} \rangle$. Because $y_n^{\mathrm{adv}}$ is mislabeled, the sign (binary decision) of the first term is not always consistent with human perception. The sign of the second term depends only on the sign of the standard decision boundary $f^{\mathrm{bdy}}(\boldsymbol{z})$. When the magnitude of the second term is dominant, $\mathrm{sgn}(f_{\mathrm{adv}}^{\mathrm{bdy}}(\boldsymbol{z}))$ matches $\mathrm{sgn}(f^{\mathrm{bdy}}(\boldsymbol{z}))$. This suggests that although the dataset appears mislabeled to humans, the classifier can still provide a reasonable prediction. A more general version of Theorem 4.1, without assuming $N > C^2/R_{\min}^2$ is given in Theorem C.2.

**Perturbation Constraint.** The assumption, Ineq. (2), requires mutually orthogonal adversarial samples, which restricts the perturbation constraint to $\epsilon = \mathcal{O}(\sqrt{d/N})$. The perturbation constraint $\epsilon$ linearly increases the dominance of the perturbation effect. If we ignore the restriction, then $\mathrm{sgn}(f_{\mathrm{adv}}^{\mathrm{bdy}}(\boldsymbol{z})) = \mathrm{sgn}(f^{\mathrm{bdy}}(\boldsymbol{z}))$ holds for any $\boldsymbol{z}$ with $\epsilon \to \infty$. This aligns with the intuition that large perturbations restore the standard decision boundary.

**Consistent Growth of Two Effects.** Here, we provide a short summary of the limiting behavior of Eq. (3). A detailed analysis can be found in Proposition C.4. Let $g(N,d)$ and $h(N,d)$ be positive functions of the number of training samples $N$ and input dimension $d$. In addition, let $T_1(\boldsymbol{z})$ and $T_2(\boldsymbol{z})$ be the first and second terms of Eq. (3), respectively. Given the labels $y_n$ and $y_n^{\mathrm{adv}}$ freely selected from $\{\pm 1\}$, estimating the growth rate for $T_1(\boldsymbol{z})$ and $T_2(\boldsymbol{z})$ is challenging. Therefore, we assume that $|\sum_{n=1}^N \lambda_n y_n \langle \boldsymbol{x}_n, \boldsymbol{z} \rangle| = \Theta(g(N,d))$ if $\sum_{n=1}^N \lambda_n |\langle \boldsymbol{x}_n, \boldsymbol{z} \rangle| = \Theta(g(N,d))$ and instead estimate the growth rate of $\sum_{n=1}^N \lambda_n |\langle \boldsymbol{x}_n, \boldsymbol{z} \rangle|$ rather than $|\sum_{n=1}^N \lambda_n y_n \langle \boldsymbol{x}_n, \boldsymbol{z} \rangle|$. A similar assumption applies to $|\sum_{n=1}^N \lambda_n^{\mathrm{adv}} y_n^{\mathrm{adv}} \langle \boldsymbol{x}_n, \boldsymbol{z} \rangle|$. Note that these assumptions are removed in noise data scenarios. Interestingly, under these conditions, for any $\boldsymbol{z}$, $|T_1(\boldsymbol{z})| = \Theta(h(N,d)) \Leftrightarrow |T_2(\boldsymbol{z})| = \Theta(h(N,d))$ holds, indicating consistent growth of the two terms. For example, if a test sample is weakly correlated with all the training samples, e.g., $\boldsymbol{z} = \sum_{n=1}^N \Theta(1/\sqrt{N}) \boldsymbol{x}_n$, then $|T_1(\boldsymbol{z})| = \Theta(d/\sqrt{N})$ and $|T_2(\boldsymbol{z})| = \Theta(d/\sqrt{N})$. Note that the scaling factor $\Theta(1/\sqrt{N})$ ensures $\Theta(\|\boldsymbol{z}\|) = \sqrt{d}$. This consistent growth implies that the effect of mislabeled data, $T_1(\boldsymbol{z})$, is not dominant even with a large input dimension $d$ and sample size $N$. Thus, learning from perturbations is feasible for high-dimensional datasets with numerous samples.

**Random Label Learning.** Next, we consider the limiting behavior of $T_1(\boldsymbol{z})$ and $T_2(\boldsymbol{z})$ when $y_n^{\mathrm{adv}}$ is randomly sampled from $\{\pm 1\}$. A detailed analysis is provided in Proposition C.8. Intuitively, if $y_n^{\mathrm{adv}}$ randomly takes $\pm 1$ independently of the sample $\boldsymbol{x}_n$ and its original label $y_n$, the magnitude of the numerator of $T_1(\boldsymbol{z})$ does not increase significantly, while the denominator consistently increases as the sample size $N$ increases. Consequently, the growth rate of $|T_1(\boldsymbol{z})|$ is lower than that of $|T_2(\boldsymbol{z})|$. Following this reasoning, we can demonstrate that $|T_2(\boldsymbol{z})|$ surpasses $|T_1(\boldsymbol{z})|$ except for a specific $\boldsymbol{z}$. For example, if $\boldsymbol{z}$ is weakly correlated with all the training samples, e.g., $\boldsymbol{z} = \sum_{n=1}^N \Theta(1/\sqrt{N}) \boldsymbol{x}_n$, then $|T_1(\boldsymbol{z})| = \mathcal{O}(d/N)$ and $|T_2(\boldsymbol{z})| = \Theta(d/\sqrt{N})$. With enough training samples, $|T_2(\boldsymbol{z})| > |T_1(\boldsymbol{z})|$ and $\mathrm{sgn}(f_{\mathrm{adv}}^{\mathrm{bdy}}(\boldsymbol{z})) = \mathrm{sgn}(f^{\mathrm{bdy}}(\boldsymbol{z}))$ hold. That is, classifiers trained on an apparently mislabeled dataset produce reasonable decisions for samples that are weakly correlated with the training samples. In contrast, if a test sample has a strong correlation with particular training samples, e.g., $\boldsymbol{z} = \Theta(1) \boldsymbol{x}_1$, the consistent growth of $|T_1(\boldsymbol{z})| = \mathcal{O}(d/N)$ and $|T_2(\boldsymbol{z})| = \Theta(d/N)$ persists. These results can be summarized and generalized as follows:

**Theorem 4.2** (Consistent decision of learning from geometry-inspired perturbations on natural samples). *Suppose that* Ineq. (2) *holds. Assume* $\|\boldsymbol{x}_n\| = \Theta(\sqrt{d})$ *for any* $n \in [N]$ *and* $\|\boldsymbol{z}\| = \Theta(\sqrt{d})$. *Suppose that* $y_n^{\mathrm{adv}}$ *is randomly sampled from* $\{\pm 1\}$ *for each* $n$. *Assume* $|\sum_{n=1}^N \lambda_n y_n \langle \boldsymbol{x}_n, \boldsymbol{z} \rangle| = \Theta(g(N,d))$ *if* $\sum_{n=1}^N \lambda_n |\langle \boldsymbol{x}_n, \boldsymbol{z} \rangle| = \Theta(g(N,d))$, *where* $g$ *is a positive*

*function of $N$ and $d$. If there is no $n$ such that $|\langle \boldsymbol{x}_n, \boldsymbol{z} \rangle| = \Theta(d)$ or $\sum_{n:|\langle \boldsymbol{x}_n, \boldsymbol{z} \rangle| \neq \Theta(d)} |\langle \boldsymbol{x}_n, \boldsymbol{z} \rangle| = \mathcal{O}(d)$ does not hold, with $N, d \to \infty$, then $\mathrm{sgn}(f_{\mathrm{adv}}^{\mathrm{bdy}}(\boldsymbol{z})) = \mathrm{sgn}(f^{\mathrm{bdy}}(\boldsymbol{z}))$ holds with probability at least 99.99%.*

This theorem suggests that classifiers trained on an apparently mislabeled dataset can produce decisions consistent with standard classifiers, except for specific inputs that satisfy the following two conditions: (A) there exists $n$ such that $|\langle \boldsymbol{x}_n, \boldsymbol{z} \rangle| = \Theta(d)$, and (B) $\sum_{n:|\langle \boldsymbol{x}_n, \boldsymbol{z} \rangle| \neq \Theta(d)} |\langle \boldsymbol{x}_n, \boldsymbol{z} \rangle| = \mathcal{O}(d)$ holds. Such exceptional inputs could be, for example, $\boldsymbol{z} = \boldsymbol{x}_1$, $\boldsymbol{x}_1 + \boldsymbol{x}_2 + \boldsymbol{x}_3$, and $\boldsymbol{x}_1 + \mathcal{O}(1/N)\mathbf{1}$, where $\mathbf{1}$ denotes an all-ones vector. Condition (A) represents a strong correlation with a few samples. Note that a strong correlation with many samples is invalid due to the orthogonality of $\{\boldsymbol{x}_n\}_{n=1}^N$ and $\|\boldsymbol{z}\| = \Theta(\sqrt{d})$ (cf. Lemma C.5). Condition (B) indicates that $\boldsymbol{z}$ has no weak correlation with many samples. For such $\boldsymbol{z}$, the impact of learning from mislabeled samples, $T_1(\boldsymbol{z})$, becomes dominant, and the decisions are not always aligned. Essentially, for inputs that do not strongly correlate with a few samples or weakly correlate with many samples, the network decisions align with those of a standard network. Because test datasets typically exclude samples similar to the training samples, a network learning from perturbations is expected to produce reasonable predictions for many test samples. This confirms the high test accuracy of learning from perturbations (Ilyas et al., 2019).

**Others.** In Appendix E, we derive similar results for other perturbation forms. In Appendix G, we establish Theorem 4.1 without the assumption on the last layer of a network. Moreover, Theorem 4.2 explains the success of learning from perturbations using random labels. In Appendix H, we attempt to delve into a flipped label scenario (i.e., $y_n^{\mathrm{adv}} = -y_n$) through an empirically supported assumption that standard classifiers focus on non-robust features (Etmann et al., 2019; Tsipras et al., 2019; Zhang & Zhu, 2019; Chalasani et al., 2020).

## 4.3 LEARNING FROM ADVERSARIAL PERTURBATIONS ON UNIFORM NOISES

In this section, we consider a noise data scenario in which adversarial perturbations are superimposed on uniform noises. The proofs of the theorems can be found in Appendix D. This scenario provides two advantages. First, this scenario prevents the unintentional leakage of useful features. For example, a frog image labeled as a horse may contain horses in the background. Second, this scenario can justify learning from perturbations without assuming $|\sum_{n=1}^{N^{\mathrm{adv}}} \lambda_n y_n \langle \boldsymbol{x}_n, \boldsymbol{z} \rangle| = \Theta(g(N, d))$ if $\sum_{n=1}^{N^{\mathrm{adv}}} \lambda_n |\langle \boldsymbol{x}_n, \boldsymbol{z} \rangle| = \Theta(g(N, d))$. Similarly to Theorem 4.1, we can derive the following decision boundary in the noise data scenario:[4]

**Theorem 4.3** (Decision boundary when learning from geometry-inspired perturbations on uniform noises). *Assume $\gamma^3 R_{\min}^4 / (3N R_{\max}^2) \geq p_{\max}$. Let $f$ be a one-hidden-layer neural network trained on geometry-inspired perturbations on natural data (cf. Eq. (1) and Definition 3.2(b)) with Setting 3.1. For any $n \neq k$, if*

$$\frac{d}{3} - \frac{\sqrt{Cd}}{2} \leq \|\boldsymbol{X}_n\|^2 \leq \frac{d}{3} + \frac{\sqrt{Cd}}{2}, \quad |\langle \boldsymbol{X}_n, \boldsymbol{X}_k \rangle| \leq \sqrt{2Cd}, \quad |\langle \boldsymbol{X}_n, \boldsymbol{\eta}/\epsilon \rangle| \leq \sqrt{2C}, \quad (4)$$

$$\frac{\gamma^3 (2d - 3\sqrt{Cd} - 12\sqrt{2C}\epsilon + 6\epsilon^2)^2}{18 N^{\mathrm{adv}} (2d + 3\sqrt{Cd} + 12\sqrt{2C}\epsilon + 6\epsilon^2)} \geq \sqrt{2Cd} + 2\sqrt{2C}\epsilon + \epsilon^2 \quad (5)$$

*with $C := \ln 1000 N^{\mathrm{adv}}$, then, with $t \to \infty$, the decision boundary of $f$ is given by:*

$$f_{\mathrm{adv}}^{\mathrm{bdy}}(\boldsymbol{z}) := \underbrace{\frac{\sum_{n=1}^{N^{\mathrm{adv}}} \lambda_n^{\mathrm{adv}} y_n^{\mathrm{adv}} \langle \boldsymbol{X}_n, \boldsymbol{z} \rangle}{\sum_{n=1}^{N^{\mathrm{adv}}} \lambda_n^{\mathrm{adv}}}}_{\text{Effect of learning from uniform noises}} + \underbrace{\epsilon \frac{f^{\mathrm{bdy}}(\boldsymbol{z})}{\|\sum_{n=1}^N \lambda_n y_n \boldsymbol{x}_n\|}}_{\text{Effect of learning from perturbations}}. \quad (6)$$

---

[4]In Theorem 4.3, we assume $\gamma^3 R_{\min}^4 / (3N R_{\max}^2) \geq p_{\max}$ to define geometry-inspired perturbations. To derive the decision boundary Eq. (6), we require only Ineqs. (4) and (5) and need not assume the orthogonality of natural training samples.

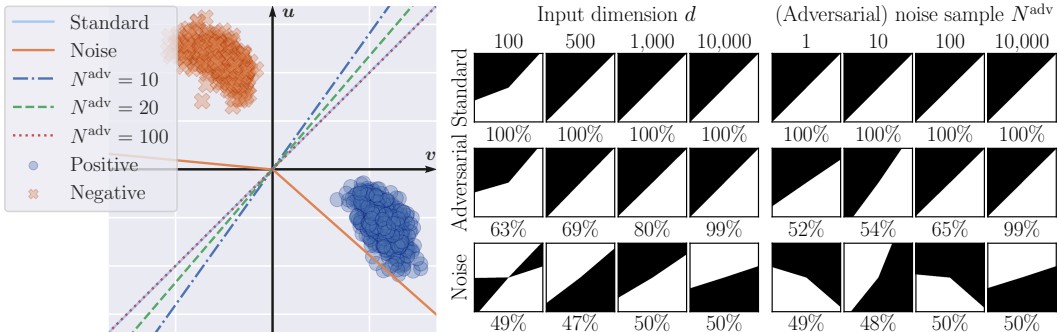

Figure 1: Decision boundaries of classifiers trained on multidimensional artificial datasets. The axis vectors $v$ and $u$ are defined in Theorem B.1. **Left:** Boundaries from standard data and noises with and without adversarial perturbations ($d = 10,000$). The blue circles and orange crosses indicate standard data projections onto this plane. **Right:** Boundaries across varying input dimensions (fix $N^{\text{adv}} = 10,000$) and number of noise samples (fix $d = 10,000$). First row: results from standard data; second and third rows: results from noises with and without adversarial perturbations, respectively. Percentages indicate the classification accuracy for the standard data.

Similar to Eq. (3), the decision boundary, Eq. (6), consists of two terms, representing the effects of uniform noises and geometry-inspired perturbations. Although we assume Ineq. (4), each inequality holds with probability at least 99.8% (cf. Lemma D.1). Similarly to Ineq. (2), the assumption, Ineq. (5), requires the training adversarial samples to be nearly orthogonal and restricts the perturbation constraint to $\epsilon = \tilde{\mathcal{O}}(\sqrt{d/N^{\text{adv}}})$ if $d > N^{{\text{adv}}^2}$. Next, we examine the growth rate of the two terms in Eq. (6) and the alignment between the decision boundaries.

**Theorem 4.4** (Consistent decision of learning from geometry-inspired perturbations on uniform noises). *Assume $\gamma^3 R_{\min}^4/(3NR_{\max}^2) \geq p_{\max}$. Suppose that Ineqs. (4) and (5) hold. Assume $\|z\| = \Theta(\sqrt{d})$, $|f^{\text{bdy}}(z)| = \Omega(1)$, and $d > N^{{\text{adv}}^2}$. Then, the following equations hold with probability at least 99.99%:*

$$\left| \frac{\sum_{n=1}^{N^{\text{adv}}} \lambda_n^{\text{adv}} y_n^{\text{adv}} \langle X_n, z \rangle}{\sum_{n=1}^{N^{\text{adv}}} \lambda_n^{\text{adv}}} \right| = \mathcal{O}\left( \frac{\sqrt{d}}{N^{\text{adv}}} \right), \qquad \epsilon \frac{|f^{\text{bdy}}(z)|}{\| \sum_{n=1}^{N} \lambda_n y_n x_n \|} = \tilde{\Omega}\left( \frac{d}{\sqrt{N^{\text{adv}} N}} \right). \quad (7)$$

*In addition, if $d$ and $N^{\text{adv}}$ are sufficiently large and $N^{\text{adv}} \geq N$ holds, then for any $z \in \mathbb{R}^d$, $\text{sgn}(f_{\text{adv}}^{\text{bdy}}(z)) = \text{sgn}(f^{\text{bdy}}(z))$ holds with probability at least 99.99%.*

This theorem indicates a strong alignment between the decision boundaries derived from natural samples and adversarial perturbations on uniform noises. Recall that in the natural sample scenario, consistent decisions are obtained, except for samples that are strongly correlated with a few training samples and not weakly correlated with many training samples (cf. Theorem 4.2). In contrast, Theorem 4.4 claims that consistent decisions can be obtained for any input with high probability. A large input dimension $d$ and number of adversarial samples $N^{\text{adv}}$ make the alignment stronger with the speed of $\tilde{\Omega}(\sqrt{N^{\text{adv}}d})$ at least.

The assumption $|f^{\text{bdy}}(z)| = \Omega(1)$ is mild due to its definition $f^{\text{bdy}}(z) = \sum_{n=1}^{N} \lambda_n y_n \langle x_n, z \rangle$. We introduce this assumption because the order of $f^{\text{bdy}}(z)$ cannot be determined owing to the uncertainty of $y_n$. Note that we assume $|\sum_{n=1}^{N^{\text{adv}}} \lambda_n y_n \langle x_n, z \rangle| = \Theta(g(N, d))$ if $\sum_{n=1}^{N^{\text{adv}}} \lambda_n |\langle x_n, z \rangle| = \Theta(g(N, d))$ in the natural sample scenario. Interestingly, even though we underestimate the order of $f^{\text{bdy}}(z)$, the effect of learning from perturbations still grows faster than that from uniform noises.

The theorem fails in at most 0.01% of cases where the randomly generated $X_n$ and the input $z$ are strongly correlated. For example, the theorem does not hold if $X_n$ is identical to $z$. However, these cases are rare if $d$ is sufficiently large.

In addition, as a corollary of Theorem 4.4, we can derive the following theorem:

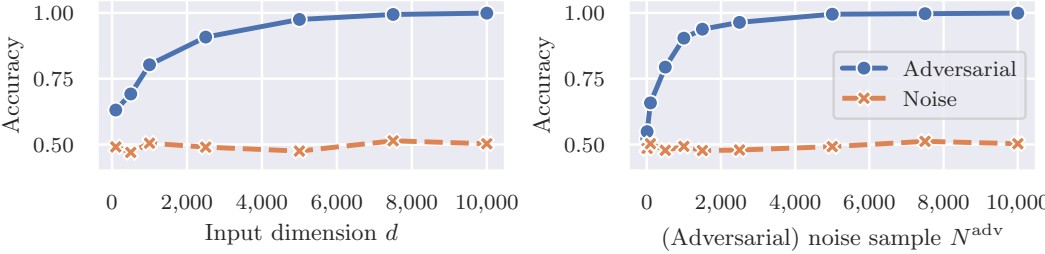

Figure 2: Accuracy of classifiers trained on uniform noises with or without adversarial perturbations for standard data in artificial dataset. The blue solid and orange dashed lines represent the results from noises with and without perturbations (i.e., pure noises), respectively. We fix $N^{\text{adv}} = 10,000$ on the left and $d = 10,000$ on the right.

---

**Corollary 4.5** (Complete classification for natural training samples when learning from geometry-inspired perturbations on uniform noises). *Assume $\gamma^3 R_{\min}^4/(3NR_{\max}^2) \geq p_{\max}$. Suppose that Ineqs. (4) and (5) hold. If $d$ and $N^{\text{adv}}$ are sufficiently large and $d > N^{\text{adv}^2} \geq \sqrt{N}$ holds, then a one-hidden-layer neural network trained on geometry-inspired perturbations on uniform noises with Setting 3.1 can completely classify the natural dataset $\{(\boldsymbol{x}_n, y_n)\}_{n=1}^{N}$ with probability at least 99.99%.*

---

This corollary claims that a classifier learning from perturbations on uniform noises can accurately classify the natural training samples even though the classifier does not see any natural data during training. This result highlights abundant class features in adversarial perturbations and justifies learning from them.

## 5 EXPERIMENTAL RESULTS

In this section, we empirically verify our theoretical results. Detailed experimental settings and additional results for other norms ($L_0$ and $L_\infty$) and Gaussian noises can be found in Appendix I.

### 5.1 ARTIFICIAL DATASET

In this section, we validate Theorem 4.4 and Corollary 4.5 using one-hidden-layer neural networks on artificial datasets. The standard dataset $\mathcal{D} := \{(\boldsymbol{x}_n, y_n)\}_{n=1}^{N}$ consists of $\boldsymbol{x}_n$ and $y_n$ sampled from $U([-1, 1]^d)$ and $U(\{\pm 1\})$, respectively. Our theorems only require the orthogonality of training samples; thus, using uniform noises as training samples poses no problem. The noise data $\{\boldsymbol{X}_n\}_{n=1}^{N^{\text{adv}}}$, on which perturbations were superimposed, were also uniformly distributed, i.e., $\boldsymbol{X}_n \sim U([-1, 1]^d)$. Using a network trained on $\mathcal{D}$ and $L_2$-constrained attack, we generated the adversarial dataset $\mathcal{D}^{\text{adv}} := \{(\boldsymbol{x}_n^{\text{adv}}, y_n^{\text{adv}})\}_{n=1}^{N^{\text{adv}}}$ with random target labels $y_n^{\text{adv}} \in \{\pm 1\}$. Visualizations of $\boldsymbol{x}_n$, $\boldsymbol{X}_n$, and $\boldsymbol{x}_n^{\text{adv}}$ can be found in Fig. A3.

In Fig. 1, the decision boundaries for each scenario are shown in a two-dimensional space spanned by the vectors $\boldsymbol{v} \in \mathbb{R}^d$ and $\boldsymbol{u} \in \mathbb{R}^d$ (cf. Theorem B.1). Theoretically, $\boldsymbol{v}$ and $\boldsymbol{u}$ draw the standard classifier's boundary diagonally from the top right to bottom left.[5] The left panel shows the boundaries of the networks trained on various datasets with $d = 10,000$. Although the boundary derived from pure noises (orange) differed significantly from the standard boundary (light blue), the boundary derived from adversarial perturbations (blue, green, and red for $N^{\text{adv}} = 10, 20$, and $100$, respectively) became more aligned with the standard boundary as the number of samples increased. The right panel shows the decision boundaries across various input dimensions and numbers of noise samples. While the boundaries from noises deviated markedly from the standard, those from perturbations closely mirrored the standard. As the input dimension and adversarial noise samples increased, the alignment became stronger. This boundary alignment is consistent with Theorem 4.4.

---
[5]The standard boundary deviates from the theoretical prediction when $d = 100$ because Theorem 3.3 is difficult to hold in low-dimensional data.

Table 1: Accuracy in each scenario. "R" denotes random selection of adversarial target labels, while "D" denotes deterministic selection. In the noise data scenario, $y_n^{\text{adv}}$ is always chosen randomly from ten labels. Accuracies above 15% are highlighted in bold. The underlined scenarios were also considered in Ilyas et al. (2019).

| | On natural samples | | | | | | On noise data | | |
|---|---|---|---|---|---|---|---|---|---|
| | $L_0$ (R) | $L_0$ (D) | $L_2$ (R) | $L_2$ (D) | $L_\infty$ (R) | $L_\infty$ (D) | $L_0$ | $L_2$ | $L_\infty$ |
| MNIST | **28.4** | 0.71 | **92.9** | **38.4** | **89.1** | 10.3 | **33.2** | **27.9** | **31.9** |
| FMNIST | 10.5 | 1.31 | **54.8** | **25.1** | **61.4** | **22.9** | **26.2** | **30.2** | **26.2** |
| CIFAR-10 | **54.9** | **16.8** | **77.1** | **42.8** | **77.2** | **51.5** | 9.87 | 10.2 | 9.74 |

In Fig. 2, we illustrate the accuracy of classifiers learning from perturbations on uniform noises for standard data. The accuracy improved as the input dimensions or number of adversarial samples increased. Given enough of these values, the classifiers could achieve near-perfect classification even though they had not seen any standard data. This counterintuitive success of learning from perturbations aligns with Corollary 4.5.

## 5.2 MNIST/FASHION-MNIST/CIFAR-10

Table 1 shows accuracy in each scenario for MNIST (Deng, 2012), Fashion-MNIST (Xiao et al., 2017), and CIFAR-10 (Krizhevsky, 2009). We used a six-layer convolutional neural network for MNIST and Fashion-MNIST and WideResNet (Zagoruyko & Komodakis, 2016) for CIFAR-10. Examples of adversarial images are shown in Figs. A16 to A18. In the table, "R" indicates that a target label was randomly chosen from the nine labels that differ from an original label and "D" indicates that a target label was deterministically chosen as the next sequential label after an original label. Random selection eliminates feature-label correlations, clarifying the learning impact from perturbations. Under deterministic selection, an anti-correlation between features and labels exists, and high test accuracy emphasizes that perturbations contain label-aligned features.

Table 1 indicates that classifiers learning from perturbations can achieve high test accuracy, beyond the cases in Ilyas et al. (2019). Remarkably, even $L_0$ perturbations, which appear to lack natural data structures, enable network generalization. These results support our theory that even sparse perturbations contain class features. Furthermore, successful learning in the noise data scenario is not limited to the artificial datasets in Section 5.1 but also extends to natural data distributions such as MNIST and Fashion-MNIST, supporting the validity of Theorem 4.3.

Consider counterexamples. For Fashion-MNIST, learning from $L_0$ perturbations was successful in the noise data scenario, but not in the natural sample scenario. This may be due to the nature of Fashion-MNIST, where a few pixels may not sufficiently overwrite the inherent features of objects spanning large regions of an image. For CIFAR-10, the noise data scenarios were challenging, possibly because of the domain gap between noise data and natural images, and the classifier's inability to extract generalizable features for natural data from noises.

## 6 CONCLUSION AND LIMITATION

We provided the first theoretical justification for learning from adversarial perturbations for one-hidden-layer networks trained on mutually orthogonal samples. We revealed that various perturbations, even sparse perturbations, contain sufficient class features for generalization. Moreover, we demonstrated that in natural sample and noise data scenarios, networks learning from perturbations produce decisions consistent with those of normally trained networks, except for specific inputs.

Our major limitations are the assumptions of a simple model, i.e., a one-hidden-layer neural network, and orthogonal training samples. In particular, the orthogonality assumptions, Ineqs. (2) and (5), restrict the perturbation constraint $\epsilon$ to $\mathcal{O}(\sqrt{d/N})$. However, in practice, $\epsilon$ is typically set to $\mathcal{O}(\sqrt{d})$. Relaxing these conditions enhances the applicability of our theorems. Nevertheless, our research provides the first fundamental insight and justification of learning from perturbations, which supports the feature hypothesis and various puzzling phenomena of adversarial examples.

ACKNOWLEDGMENTS

We would like to thank Huishuai Zhang for useful discussions. S. Kumano was supported by JSPS KAKENHI Grant Number JP23KJ0789, by JST, ACT-X Grant Number JPMJAX23C7, JAPAN, and by Microsoft Research Asia. H. Kera was supported by JSPS KAKENHI Grant Number JP22K17962. T. Yamasaki was supported by JSPS KAKENHI Grant Number JP22H03640 and Institute for AI and Beyond of The University of Tokyo.

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

# A    LEARNING FROM ADVERSARIAL PERTURBATIONS

In this study, we provide theoretical insights into learning from adversarial perturbations (cf. Definitions 1.1 and 3.2). Here, we clarify our research focus in comparison with adversarial training and training with noisy labels, and delve into the implications of learning from perturbations.

## A.1    COMPARISON WITH ADVERSARIAL TRAINING

Learning from perturbations, which aims to verify the feature hypothesis that perturbations contain class features, is not related to adversarial training, which aims to train classifiers to be robust against adversarial attacks. Learning from perturbations does not focus on (adversarial) robustness. Their concepts and procedures are summarized as follows:

**Learning from adversarial perturbations** (Ilyas et al., 2019): Given a dataset $\mathcal{D} := \{(\boldsymbol{x}_n, y_n)\}$ and a classifier $f$ trained on $\mathcal{D}$, we create adversarial examples $\boldsymbol{x}_n^{\mathrm{adv}}$ such that $f$'s prediction becomes $y_n^{\mathrm{adv}} \neq y_n$, i.e., $f(\boldsymbol{x}_n) = y_n$ and $f(\boldsymbol{x}_n^{\mathrm{adv}}) = y_n^{\mathrm{adv}}$, thereby forming a new dataset $\mathcal{D}^{\mathrm{adv}} := \{(\boldsymbol{x}_n^{\mathrm{adv}}, y_n^{\mathrm{adv}})\}$. These adversarial examples $\boldsymbol{x}_n^{\mathrm{adv}}$ appear indistinguishable from natural images $\boldsymbol{x}$ to the human eye, making $\mathcal{D}^{\mathrm{adv}}$ seemingly mislabeled. However, a classifier $g$ trained from scratch on $\mathcal{D}^{\mathrm{adv}}$ surprisingly yields high test accuracy on standard datasets that are correctly labeled from a human perspective. This counterintuitive generalization suggests that adversarial perturbations, while appearing as noises, contain label-aligned features.

**Adversarial training** (Madry et al., 2018): Given a dataset $\mathcal{D} := \{(\boldsymbol{x}_n, y_n)\}$ and an initialized classifier $f$, adversarial training updates $f$'s parameters to minimize losses over $\{(\boldsymbol{x}_n^{\mathrm{adv}}, y_n)\}$, which is regenerated at each training iteration. The trained classifier $f$ achieves high test accuracy on standard datasets under adversarial attacks.

Although both methods use adversarial examples, their goals and procedures are different. Our work attempts to provide a theoretical explanation for the success of learning from perturbations. This is a novel endeavor that differs from extensive studies on adversarial training. To the best of our knowledge, the reasons for the success of learning from adversarial perturbations have not been explained theoretically.

## A.2    COMPARISON WITH TRAINING WITH NOISY LABELS

Learning from perturbations is not training with noisy labels with adversarial examples. While in training with noisy labels, labels are partially mislabeled (e.g., 20% labels of whole data are mislabeled), in learning from perturbations, all labels are mislabeled. In this study, we do not focus on obtaining classifiers with high accuracy under mislabeled adversarial examples. Our study provides a theoretical explanation for why classifiers can obtain generalization ability to correctly labeled test samples from completely mislabeled adversarial examples.

## A.3    IMPLICATIONS OF LEARNING FROM ADVERSARIAL PERTURBATIONS

In this section, we introduce the implications of learning from adversarial perturbations using simple examples. The training, test, and adversarial datasets are defined as follows:

$$\mathcal{D} := \{(\mathrm{frog\ img}, \mathrm{frog}), (\mathrm{horse\ img}, \mathrm{horse}), (\mathrm{cat\ img}, \mathrm{cat})\}, \tag{A8}$$

$$\mathcal{D}^{\mathrm{test}} := \{(\mathrm{frog\ img}*, \mathrm{frog}), (\mathrm{horse\ img}*, \mathrm{horse}), (\mathrm{cat\ img}*, \mathrm{cat})\}, \tag{A9}$$

$$\mathcal{D}^{\mathrm{adv}} := \{(\mathrm{frog\ img}+, \mathrm{horse}), (\mathrm{horse\ img}+, \mathrm{cat}), (\mathrm{cat\ img}+, \mathrm{frog})\}. \tag{A10}$$

A trained classifier $f$ can predict the correct labels for natural images (e.g., $f(\mathrm{frog\ img}) = \mathrm{frog}$ and $f(\mathrm{frog\ img}*) = \mathrm{frog}$) but not for adversarial examples (e.g., $f(\mathrm{frog\ img}+) = \mathrm{horse}$). Note that frog img+ still appears to be a frog to humans. Counterintuitively, another classifier $g$ trained from scratch on $\mathcal{D}^{\mathrm{adv}}$ can correctly predict the classes of the natural test images in $\mathcal{D}^{\mathrm{test}}$ (e.g., $g(\mathrm{frog\ img}*) = \mathrm{frog}$).

Ilyas et al. (2019) hypothesized that adversarial perturbations contain imperceptible class features to humans. For example, frog img+ contains not only visible frog features but also invisible horse features.

Table A2: Example of adversarial dataset.

| Data index | Visible features | Invisible features | Label |
|---|---|---|---|
| 1 | Frog | Horse | Horse |
| 2 | Horse | Cat | Cat |
| 3 | Cat | Frog | Frog |
| 4 | Frog | Cat | Cat |
| 5 | Horse | Frog | Frog |
| 6 | Cat | Horse | Horse |
| . . . | . . . | . . . | . . . |

Consider training on the adversarial dataset $\mathcal{D}^{\mathrm{adv}}$ defined as Table A2. Through training, a classifier $g$ ignores visible features that are uncorrelated with labels and learns invisible features that are correlated with labels. Since natural test images contain invisible features of the corresponding classes (e.g., a frog image contains human-invisible frog features), this classifier can provide correct predictions for them. As a result, the classifier trained on a dataset that appears completely mislabeled to humans achieves high accuracy on natural test datasets.

# B DECISION BOUNDARY OF ONE-HIDDEN-LAYER NEURAL NETWORK

## B.1 STANDARD TRAINING

To formulate learning from adversarial perturbations, we use the theorems presented in Frei et al. (2023) (similar results are shown in Sarussi et al. (2021)), which addresses the implicit bias of one-hidden-layer neural networks under gradient flow with an exponential loss. This theorem does not directly address adversarial attacks, adversarial examples, or learning from perturbations. We leverage this because of the tractable form of a decision boundary. The main results of their study are summarized as follows:

**Theorem B.1** (Rearranged from Frei et al. (2023)). *Let* $\mathcal{D} := \{(\boldsymbol{x}_n, y_n)\}_{n=1}^N \subset \mathbb{R}^d \times \{\pm 1\}$ *be a training dataset. Let* $R_{\max} := \max_n \|\boldsymbol{x}_n\|$, $R_{\min} := \min_n \|\boldsymbol{x}_n\|$, *and* $p_{\max} := \max_{n \neq k} |\langle \boldsymbol{x}_n, \boldsymbol{x}_k \rangle|$. *A one-hidden-layer neural network* $f : \mathbb{R}^d \to \mathbb{R}$ *is trained on* $\mathcal{D}$ *with Setting 3.1. If* $\gamma^3 R_{\min}^4 / (3N R_{\max}^2) \geq p_{\max}$, *then gradient flow on* $f$ *converges to* $\lim_{t \to \infty} \frac{\boldsymbol{W}(t)}{\|\boldsymbol{W}(t)\|_F} = \frac{\boldsymbol{W}^{\mathrm{std}}}{\|\boldsymbol{W}^{\mathrm{std}}\|_F}$, *where* $\boldsymbol{W}^{\mathrm{std}} := (\boldsymbol{v}_1, \ldots, \boldsymbol{v}_{m/2}, \boldsymbol{u}_1, \ldots, \boldsymbol{u}_{m/2})^\top$ *satisfies*

$$\forall n \in [N] : y_n f(\boldsymbol{x}_n; \boldsymbol{W}^{\mathrm{std}}) = 1, \tag{A11}$$

$$\boldsymbol{v}_1 = \cdots = \boldsymbol{v}_{m/2} = \boldsymbol{v} := \frac{1}{\sqrt{m}} \sum_{n:y_n=+1} \lambda_n \boldsymbol{x}_n - \frac{\gamma}{\sqrt{m}} \sum_{n:y_n=-1} \lambda_n \boldsymbol{x}_n, \tag{A12}$$

$$\boldsymbol{u}_1 = \cdots = \boldsymbol{u}_{m/2} = \boldsymbol{u} := \frac{1}{\sqrt{m}} \sum_{n:y_n=-1} \lambda_n \boldsymbol{x}_n - \frac{\gamma}{\sqrt{m}} \sum_{n:y_n=+1} \lambda_n \boldsymbol{x}_n, \tag{A13}$$

*where* $\lambda_n \in \left( \frac{1}{2R_{\max}^2}, \frac{3}{2\gamma^2 R_{\min}^2} \right)$ *for every* $n \in [N]$. *The binary decision of* $f(\boldsymbol{z}; \boldsymbol{W}^{\mathrm{std}})$ *is also given by:*

$$\mathrm{sgn}\left(f(\boldsymbol{z}; \boldsymbol{W}^{\mathrm{std}})\right) = \mathrm{sgn}\left(f^{\mathrm{bdy}}(\boldsymbol{z})\right), \quad \text{where} \quad f^{\mathrm{bdy}}(\boldsymbol{z}) := \sum_{n=1}^N \lambda_n y_n \langle \boldsymbol{x}_n, \boldsymbol{z} \rangle. \tag{A14}$$

The theorem provides three insights: (i) Although there might be many possible directions $\boldsymbol{W}/\|\boldsymbol{W}\|_F$ that can accurately classify the training dataset, gradient flow consistently converges in direction to $\boldsymbol{W}^{\mathrm{std}}$ regardless of initial weight configurations. (ii) Given that $\boldsymbol{W}^{\mathrm{std}}$ consists of a maximum of two unique row vectors, its rank is constrained to two or less, highlighting the implicit bias of the gradient flow. (iii) The binary decision of $f(\boldsymbol{z}; \boldsymbol{W}^{\mathrm{std}})$ is the same as the sign of the linear

function $f^{\mathrm{bdy}}(\boldsymbol{z})$, indicating that $f(\boldsymbol{z}; \boldsymbol{W}^{\mathrm{std}})$ has a linear decision boundary. This theorem requires nearly orthogonal data, which is a typical characteristic of high-dimensional data.

Note that in Frei et al. (2023), the binary decision boundary is given by:

$$f^{\mathrm{bdyalt}}(\boldsymbol{z}) = \frac{\sqrt{m}}{2}\boldsymbol{v} - \frac{\sqrt{m}}{2}\boldsymbol{u}. \tag{A15}$$

To derive Eq. (A14), we rearrange the above equation as follows:

$$f^{\mathrm{bdyalt}}(\boldsymbol{z}) = \frac{\sqrt{m}}{2}\left( \frac{1}{\sqrt{m}} \sum_{n:y_n=+1} \lambda_n \boldsymbol{x}_n - \frac{\gamma}{\sqrt{m}} \sum_{n:y_n=-1} \lambda_n \boldsymbol{x}_n \right) \tag{A16}$$

$$- \frac{\sqrt{m}}{2}\left( \frac{1}{\sqrt{m}} \sum_{n:y_n=-1} \lambda_n \boldsymbol{x}_n - \frac{\gamma}{\sqrt{m}} \sum_{n:y_n=+1} \lambda_n \boldsymbol{x}_n \right) \tag{A17}$$

$$= \frac{1+\gamma}{2}\left( \sum_{n:y_n=+1} \lambda_n \boldsymbol{x}_n - \sum_{n:y_n=-1} \lambda_n \boldsymbol{x}_n \right) \tag{A18}$$

$$= \frac{1+\gamma}{2} \sum_{n=1}^{N} \lambda_n y_n \boldsymbol{x}_n. \tag{A19}$$

Thus,

$$\mathrm{sgn}\left( f(\boldsymbol{z}; \boldsymbol{W}^{\mathrm{std}}) \right) = \mathrm{sgn}\left( f^{\mathrm{bdyalt}}(\boldsymbol{z}) \right) = \mathrm{sgn}\left( \sum_{n=1}^{N} \lambda_n y_n \boldsymbol{x}_n \right). \tag{A20}$$

## B.2 LEARNING FROM ADVERSARIAL PERTURBATIONS

Theorem B.1 does not impose any assumptions on training dataset other than orthogonality. Thus, it can be adapted to a dataset with adversarial perturbations as follows:

**Corollary B.2** (Learning from adversarial perturbations). *Let $\mathcal{D}^{\mathrm{adv}} := \{(\boldsymbol{x}_n^{\mathrm{adv}}, y_n^{\mathrm{adv}})\}_{n=1}^{N^{\mathrm{adv}}} \subset \mathbb{R}^d \times \{\pm 1\}$ be a training dataset. Let $R_{\max}^{\mathrm{adv}} := \max_n \|\boldsymbol{x}_n^{\mathrm{adv}}\|$, $R_{\min}^{\mathrm{adv}} := \min_n \|\boldsymbol{x}_n^{\mathrm{adv}}\|$, and $p_{\max}^{\mathrm{adv}} := \max_{n \neq k} |\langle \boldsymbol{x}_n^{\mathrm{adv}}, \boldsymbol{x}_k^{\mathrm{adv}} \rangle|$. A one-hidden-layer neural network $f : \mathbb{R}^d \to \mathbb{R}$ is trained on the dataset with Setting 3.1. If $\gamma^3 R_{\min}^{\mathrm{adv}\,4}/(3N R_{\max}^{\mathrm{adv}\,2}) \geq p_{\max}^{\mathrm{adv}}$, then gradient flow on $f$ converges to $\lim_{t \to \infty} \frac{\boldsymbol{W}(t)}{\|\boldsymbol{W}(t)\|_F} = \frac{\boldsymbol{W}^{\mathrm{adv}}}{\|\boldsymbol{W}^{\mathrm{adv}}\|_F}$, where $\boldsymbol{W}^{\mathrm{adv}} := (\boldsymbol{v}_1^{\mathrm{adv}}, \ldots, \boldsymbol{v}_{m/2}^{\mathrm{adv}}, \boldsymbol{u}_1^{\mathrm{adv}}, \ldots, \boldsymbol{u}_{m/2}^{\mathrm{adv}})^\top$ satisfies*

$$\forall n \in [N] : y_n^{\mathrm{adv}} f(\boldsymbol{x}_n^{\mathrm{adv}}; \boldsymbol{W}^{\mathrm{adv}}) = 1, \tag{A21}$$

$$\boldsymbol{v}_1^{\mathrm{adv}} = \cdots = \boldsymbol{v}_{m/2}^{\mathrm{adv}} = \frac{1}{\sqrt{m}} \sum_{n:y_n^{\mathrm{adv}}=+1} \lambda_n^{\mathrm{adv}} \boldsymbol{x}_n^{\mathrm{adv}} - \frac{\gamma}{\sqrt{m}} \sum_{n:y_n^{\mathrm{adv}}=-1} \lambda_n^{\mathrm{adv}} \boldsymbol{x}_n^{\mathrm{adv}}, \tag{A22}$$

$$\boldsymbol{u}_1^{\mathrm{adv}} = \cdots = \boldsymbol{u}_{m/2}^{\mathrm{adv}} = \frac{1}{\sqrt{m}} \sum_{n:y_n^{\mathrm{adv}}=-1} \lambda_n^{\mathrm{adv}} \boldsymbol{x}_n^{\mathrm{adv}} - \frac{\gamma}{\sqrt{m}} \sum_{n:y_n^{\mathrm{adv}}=+1} \lambda_n^{\mathrm{adv}} \boldsymbol{x}_n^{\mathrm{adv}}, \tag{A23}$$

*where $\lambda_n^{\mathrm{adv}} \in \left( \frac{1}{2R_{\max}^{\mathrm{adv}\,2}}, \frac{3}{2\gamma^2 R_{\min}^{\mathrm{adv}\,2}} \right)$ for every $n \in [N]$. The binary decision of $f(\boldsymbol{z}; \boldsymbol{W}^{\mathrm{adv}})$ is also given by:*

$$\mathrm{sgn}\left( f(\boldsymbol{z}; \boldsymbol{W}^{\mathrm{adv}}) \right) = \mathrm{sgn}\left( f_{\mathrm{adv}}^{\mathrm{bdy}}(\boldsymbol{z}) \right), \quad \text{where} \quad f_{\mathrm{adv}}^{\mathrm{bdy}}(\boldsymbol{z}) := \sum_{n=1}^{N} \lambda_n^{\mathrm{adv}} y_n^{\mathrm{adv}} \langle \boldsymbol{x}_n^{\mathrm{adv}}, \boldsymbol{z} \rangle. \tag{A24}$$

This theorem establishes the foundation for learning from adversarial perturbations. The orthogonality assumption, model weights, and decision boundary depend on a definition of adversarial perturbations.

## C PROOFS OF THEOREMS IN SECTION 4.2

### C.1 DECISION BOUNDARY OF LEARNING FROM GEOMETRY-INSPIRED PERTURBATIONS ON NATURAL SAMPLES

In this section, we derive the decision boundary when learning from geometry-inspired perturbations on natural samples, Theorem C.2. Theorem 4.1 is a special case of Theorem C.2 with the assumption of many training samples. Lemma C.1 shows an orthogonality condition of training samples with geometry-inspired perturbations, which is required to derive the decision boundary using Corollary 3.4.

**Lemma C.1** (Orthogonality condition for learning from geometry-inspired perturbations on natural samples)**.** *Consider the geometry-inspired perturbations defined in Eq. (1). Let*

$$C := \frac{3R_{\max}^4 + \gamma^3 R_{\min}^4}{\gamma^2 R_{\min}^3 \sqrt{1-\gamma}}. \tag{A25}$$

*Suppose that the following inequalities hold:*

$$
\begin{cases}
\frac{\gamma^3 (R_{\min}-\epsilon)^4}{3N(R_{\max}+\epsilon)^2} - 2\epsilon R_{\max} - \epsilon^2 \geq p_{\max} & \left( N \leq \frac{C^2}{R_{\max}^2} \right) \\
\frac{\gamma^3 (R_{\min}-\epsilon)^4}{3N(R_{\max}^2 + 2\frac{C}{\sqrt{N}}\epsilon + \epsilon^2)} - 2\frac{C}{\sqrt{N}}\epsilon - \epsilon^2 \geq p_{\max} & \left( \frac{C^2}{R_{\max}^2} < N \leq \frac{C^2}{R_{\min}^2} \right) \\
\frac{\gamma^3 (R_{\min}^2 - 2\frac{C}{\sqrt{N}}\epsilon + \epsilon^2)^2}{3N(R_{\max}^2 + 2\frac{C}{\sqrt{N}}\epsilon + \epsilon^2)} - 2\frac{C}{\sqrt{N}}\epsilon - \epsilon^2 \geq p_{\max} & \left( N > \frac{C^2}{R_{\min}^2} \right)
\end{cases}
\tag{A26}
$$

*Then, the following inequality holds for any $\{(\boldsymbol{x}_n, y_n)\}_{n=1}^N$ and $\{y_n^{\mathrm{adv}}\}_{n=1}^N$:*

$$\frac{\gamma^3 R_{\min}^{\mathrm{adv}\,4}}{3N R_{\max}^{\mathrm{adv}\,2}} \geq p_{\max}^{\mathrm{adv}}. \tag{A27}$$

*Proof.* The proof flow is as follows:

1. Ineq. (A27) does not hold for some $\{(\boldsymbol{x}_n, y_n)\}_{n=1}^N$ and $\{y_n^{\mathrm{adv}}\}_{n=1}^N$ if $\epsilon > R_{\min}$.

2. Ineq. (A27) does not hold for some $\{(\boldsymbol{x}_n, y_n)\}_{n=1}^N$ and $\{y_n^{\mathrm{adv}}\}_{n=1}^N$ if $p_{\max} > \frac{\gamma^3 R_{\min}^4}{3N R_{\max}^2}$.

3. Ineq. (A27) holds for any $\{(\boldsymbol{x}_n, y_n)\}_{n=1}^N$ and $\{y_n^{\mathrm{adv}}\}_{n=1}^N$ if Ineq. (A26), $\epsilon \leq R_{\min}$, and $p_{\max} \leq \frac{\gamma^3 R_{\min}^4}{3N R_{\max}^2}$ hold.

4. Ineq. (A26) includes $\epsilon \leq R_{\min}$.

5. Ineq. (A26) includes $p_{\max} \leq \frac{\gamma^3 R_{\min}^4}{3N R_{\max}^2}$.

With $\boldsymbol{q} := \sum_{n=1}^N \lambda_n y_n \boldsymbol{x}_n$, we can represent the geometry-inspired adversarial example as follows:

$$\boldsymbol{x}_n^{\mathrm{adv}} := \boldsymbol{x}_n + \epsilon y_n^{\mathrm{adv}} \frac{\boldsymbol{q}}{\|\boldsymbol{q}\|}. \tag{A28}$$

**1.** Assume $\epsilon > R_{\min}$. Let $l := \arg\min_n \|\boldsymbol{x}_n\|$. We show that Ineq. (A27) does not hold if $y_n = y_k = y_n^{\mathrm{adv}} = y_k^{\mathrm{adv}}$, $y_l^{\mathrm{adv}} := -\operatorname{sgn}(\langle \boldsymbol{x}_l, \boldsymbol{q} \rangle) = -y_l$, and $p_{\max} = 0$.[6] A lower bound of the maximum inner product is

$$p_{\max}^{\mathrm{adv}} \geq \langle \boldsymbol{x}_n^{\mathrm{adv}}, \boldsymbol{x}_k^{\mathrm{adv}} \rangle = \epsilon y_n^{\mathrm{adv}} \left\langle \boldsymbol{x}_k, \frac{\boldsymbol{q}}{\|\boldsymbol{q}\|} \right\rangle + \epsilon y_k^{\mathrm{adv}} \left\langle \boldsymbol{x}_n, \frac{\boldsymbol{q}}{\|\boldsymbol{q}\|} \right\rangle + \epsilon^2 y_n^{\mathrm{adv}} y_k^{\mathrm{adv}} \geq \epsilon^2. \tag{A29}$$

---

[6]For example, $\boldsymbol{x}_1 := (2,0,0,0)$, $\boldsymbol{x}_2 := (0,2,0,0)$, $\boldsymbol{x}_3 := (0,0,1,0)$, $\boldsymbol{x}_4 := (0,0,0,2)$, $y_1 := 1$, $y_2 := 1$, $y_3 := 1$, $y_4 := -1$, $y_1^{\mathrm{adv}} := 1$, $y_2^{\mathrm{adv}} := 1$, $y_3^{\mathrm{adv}} := -1$, $y_4^{\mathrm{adv}} := -1$, $n = 1$, $k = 2$, and $l = 3$.

Note that

$$\mathrm{sgn}\left(\left\langle \boldsymbol{x}_n, \frac{\boldsymbol{q}}{\|\boldsymbol{q}\|}\right\rangle\right) = \mathrm{sgn}(\lambda_n y_n \|\boldsymbol{x}_n\|) = y_n. \tag{A30}$$

An upper bound of the minimum norm is

$$R_{\min}^{\mathrm{adv}\,2} \leq \|\boldsymbol{x}_l^{\mathrm{adv}}\|^2 = R_{\min}^2 - 2\epsilon\left|\left\langle \boldsymbol{x}_l, \frac{\boldsymbol{q}}{\|\boldsymbol{q}\|}\right\rangle\right| + \epsilon^2 \leq R_{\min}^2 + \epsilon^2. \tag{A31}$$

We can rearrange Ineq. (A27) using the above two bounds as follows:

$$\frac{\gamma^3 R_{\min}^{\mathrm{adv}\,4}}{3N R_{\max}^{\mathrm{adv}\,2}} - p_{\max}^{\mathrm{adv}} \leq \frac{R_{\min}^{\mathrm{adv}\,2}}{3} - p_{\max}^{\mathrm{adv}} \leq \frac{R_{\min}^2 + \epsilon^2}{3} - \epsilon^2 < -\frac{\epsilon^2}{3} < 0. \tag{A32}$$

**2.** Assume $p_{\max} > \gamma^3 R_{\min}^4 / 3N R_{\max}^2$. Note that the decision boundary of a classifier trained on such samples does not always converge to $f^{\mathrm{bdy}}(\boldsymbol{z}) := \sum_{n=1}^N \lambda_n y_n \langle \boldsymbol{x}_n, \boldsymbol{z}\rangle$ since the assumption of Theorem C.2 is not satisfied. However, for the definition of geometry-inspired perturbations, it is irrelevant whether the decision boundary converges to $f^{\mathrm{bdy}}$. We can define geometry-inspired perturbations as long as $\lambda_n$ is (uniquely) determined by Eq. (A11).

Let $x_1 := 1$, $x_2 := -1$, $y_1 := 1$, and $y_2 := -1$, which satisfy $p_{\max} > \gamma^3 R_{\min}^4 / 3N R_{\max}^2$. As defined in Theorem C.2,

$$v := \frac{\lambda_1 + \gamma\lambda_2}{\sqrt{m}}, \qquad\qquad u := -\frac{\lambda_2 + \gamma\lambda_1}{\sqrt{m}}. \tag{A33}$$

Thus,

$$f(x) = \frac{\phi((\lambda_1 + \gamma\lambda_2)x)}{2} - \frac{\phi(-(\lambda_2 + \gamma\lambda_1)x)}{2}. \tag{A34}$$

As defined in Theorem C.2, $\lambda_1$ and $\lambda_2$ satisfy the following simultaneous equations:

$$\begin{cases} \frac{\phi(\lambda_1 + \gamma\lambda_2)}{2} - \frac{\phi(-(\lambda_2 + \gamma\lambda_1))}{2} = 1 \\ \frac{\phi(-(\lambda_1 + \gamma\lambda_2))}{2} - \frac{\phi(\lambda_2 + \gamma\lambda_1)}{2} = -1 \end{cases}. \tag{A35}$$

Solving this,

$$\lambda_1 = \lambda_2 = 2\left(\frac{1-\gamma}{1-\gamma^2}\right)^2. \tag{A36}$$

Note that these satisfy $\lambda_1, \lambda_2 \in (1/2R_{\max}^2, 3/2\gamma^2 R_{\min}^2)$. Let $y_1^{\mathrm{adv}} := -1$ and $y_2^{\mathrm{adv}} := 1$. Then, $q/\|q\| = 1$, $x_1^{\mathrm{adv}} := 1 - \epsilon$, $x_2^{\mathrm{adv}} := -1 + \epsilon$, $R_{\min}^{\mathrm{adv}} = R_{\max}^{\mathrm{adv}} = |1 - \epsilon|$, and $p_{\max}^{\mathrm{adv}} = (1 - \epsilon)^2$. In this situation, Ineq. (A27) does not hold for any $\epsilon > 0$.

**3.** Assume $\epsilon \leq R_{\min}$ and $p_{\max} \leq \gamma^3 R_{\min}^4 / 3N R_{\max}^2$. We write the lower and upper bounds of $\lambda_n$ as $\lambda_{\min} := 1/2R_{\max}^2$ and $\lambda_{\max} := 3/2\gamma^2 R_{\min}^2$, respectively (cf. Theorem B.1).

*(Preliminary)* A lower bound of the norm of $\boldsymbol{q}$ is

$$\|\boldsymbol{q}\| = \sqrt{\sum_{n=1}^N \lambda_n \left(\lambda_n \|\boldsymbol{x}_n\|^2 + \sum_{k\neq n} \lambda_k y_n y_k \langle \boldsymbol{x}_n, \boldsymbol{x}_k\rangle\right)} \tag{A37}$$

$$\geq \sqrt{N\lambda_{\min}(\lambda_{\min} R_{\min}^2 - N\lambda_{\max} p_{\max})} \tag{A38}$$

$$= \frac{R_{\min}\sqrt{(1-\gamma)N}}{2R_{\max}^2}. \tag{A39}$$

An upper bound of the inner product between $\boldsymbol{x}_n$ and $\boldsymbol{q}$ is

$$\langle \boldsymbol{x}_n, \boldsymbol{q}\rangle = \sum_{k=1}^N \lambda_k y_k \langle \boldsymbol{x}_n, \boldsymbol{x}_k\rangle \leq \lambda_{\max}(R_{\max}^2 + N p_{\max}) = \frac{3R_{\max}^2}{2\gamma^2 R_{\min}^2} + \frac{\gamma R_{\min}^2}{2R_{\max}^2}. \tag{A40}$$

A naive upper bound of the inner product between $\boldsymbol{x}_n$ and $\boldsymbol{q}/\|\boldsymbol{q}\|$ is

$$\left\langle \boldsymbol{x}_n, \frac{\boldsymbol{q}}{\|\boldsymbol{q}\|} \right\rangle \leq R_{\max}. \tag{A41}$$

That can be also obtained as follows:

$$\left\langle \boldsymbol{x}_n, \frac{\boldsymbol{q}}{\|\boldsymbol{q}\|} \right\rangle \leq \frac{3R_{\max}^4 + \gamma^3 R_{\min}^4}{\gamma^2 R_{\min}^3 \sqrt{(1-\gamma)N}} =: \frac{C}{\sqrt{N}}. \tag{A42}$$

Note that

$$\begin{cases} \frac{C}{\sqrt{N}} \geq R_{\max} & \left(N \leq \frac{C^2}{R_{\max}^2}\right) \\ R_{\min} \leq \frac{C}{\sqrt{N}} < R_{\max} & \left(\frac{C^2}{R_{\max}^2} < N \leq \frac{C^2}{R_{\min}^2}\right) \\ \frac{C}{\sqrt{N}} < R_{\min} & \left(N > \frac{C^2}{R_{\min}^2}\right) \end{cases}. \tag{A43}$$

Thus,

$$\left\langle \boldsymbol{x}_n, \frac{\boldsymbol{q}}{\|\boldsymbol{q}\|} \right\rangle \leq \begin{cases} R_{\max} & \left(N \leq \frac{C^2}{R_{\max}^2}\right) \\ \frac{C}{\sqrt{N}} & (\text{otherwise}) \end{cases}. \tag{A44}$$

*(Upper bound of inner product)* An upper bound of the inner product is

$$\langle \boldsymbol{x}_n^{\text{adv}}, \boldsymbol{x}_k^{\text{adv}} \rangle \leq p_{\max} + 2\epsilon \left\langle \boldsymbol{x}_n, \frac{\boldsymbol{q}}{\|\boldsymbol{q}\|} \right\rangle + \epsilon^2 \tag{A45}$$

$$\leq \begin{cases} p_{\max} + 2\epsilon R_{\max} + \epsilon^2 & \left(N \leq \frac{C^2}{R_{\max}^2}\right) \\ p_{\max} + 2\frac{C}{\sqrt{N}}\epsilon + \epsilon^2 & (\text{otherwise}) \end{cases}. \tag{A46}$$

*(Lower and upper bounds of norm)* The norm of the geometry-inspired adversarial example is

$$\left\|\boldsymbol{x}_n^{\text{adv}}\right\| = \sqrt{\|\boldsymbol{x}_n\|^2 + 2\epsilon y_n^{\text{adv}} \left\langle \boldsymbol{x}_n, \frac{\boldsymbol{q}}{\|\boldsymbol{q}\|} \right\rangle + \epsilon^2}. \tag{A47}$$

Under $\epsilon \leq R_{\min}$, trivial lower and upper bounds of the above norm are

$$R_{\min} - \epsilon \leq \left\|\boldsymbol{x}_n^{\text{adv}}\right\| \leq R_{\max} + \epsilon. \tag{A48}$$

Now, we have the following three lower bounds of the norm of $\boldsymbol{x}_n$: (i) $\sqrt{R_{\min}^2 - 2\epsilon R_{\max} + \epsilon^2}$ for $N \leq \frac{C^2}{R_{\max}^2}$. (ii) $\sqrt{R_{\min}^2 - 2\frac{C}{\sqrt{N}}\epsilon + \epsilon^2}$ for $N > \frac{C^2}{R_{\max}^2}$. (iii) $R_{\min} - \epsilon$ for $\epsilon \leq R_{\min}$. Since $(R_{\min} - \epsilon)^2 - (R_{\min}^2 - 2\epsilon R_{\max} + \epsilon^2) \geq 0$, (iii) is always tighter than (i). In addition, since $(R_{\min} - \epsilon)^2 - (R_{\min}^2 - 2\frac{C}{\sqrt{N}}\epsilon + \epsilon^2) \geq 0$ under $\frac{C^2}{R_{\max}^2} < N \leq \frac{C^2}{R_{\min}^2}$, (iii) is always tighter than (ii). Thus, under $\epsilon \leq R_{\min}$,

$$\left\|\boldsymbol{x}_n^{\text{adv}}\right\| \geq \begin{cases} R_{\min} - \epsilon & \left(N \leq \frac{C^2}{R_{\min}^2}\right) \\ \sqrt{R_{\min}^2 - 2\frac{C}{\sqrt{N}}\epsilon + \epsilon^2} & (\text{otherwise}) \end{cases}. \tag{A49}$$

An upper bound of the norm is

$$\left\|\boldsymbol{x}_n^{\text{adv}}\right\| \leq \begin{cases} R_{\max} + \epsilon & \left(N \leq \frac{C^2}{R_{\max}^2}\right) \\ \sqrt{R_{\max}^2 + 2\frac{C}{\sqrt{N}}\epsilon + \epsilon^2} & (\text{otherwise}) \end{cases}. \tag{A50}$$

*(Orthogonality condition)* Using the above bounds, we can derive Ineq. (A26) where Ineq. (A27) always holds for any $\{(\boldsymbol{x}_n, y_n)\}_{n=1}^N$ and $\{y_n^{\text{adv}}\}_{n=1}^N$.

**4.** We prove that Ineq. (A26) implies $\epsilon \leq R_{\min}$. A common upper bound of the left term of Ineq. (A26) is $\gamma^3(R_{\min}^2 + \epsilon^2)/3N - \epsilon^2$. This bound monotonically decreases with $\epsilon$ and is below zero at $\epsilon = R_{\min}$. Thus, Ineq. (A26) does not hold for $\epsilon > R_{\min}$.

**5.** Here, we prove that Ineq. (A26) implies $p_{\max} \leq \gamma^3 R_{\min}^4 / 3N R_{\max}^2$. From the above discussion, we assume $\epsilon \leq R_{\min}$. In this case, $R_{\min} - \epsilon \leq R_{\min}$, $R_{\max} \leq R_{\max} + \epsilon$, $R_{\max}^2 \leq R_{\max}^2 + 2\frac{C}{\sqrt{N}}\epsilon + \epsilon^2$, $p_{\max} \leq p_{\max} + 2\epsilon \max + \epsilon^2$, and $p_{\max} \leq p_{\max} + 2\frac{C}{\sqrt{N}}\epsilon + \epsilon^2$ are trivial. Thus, the first two inequalities include $p_{\max} \leq \gamma^3 R_{\min}^4 / 3N R_{\max}^2$. Then, we consider the following inequality:

$$\frac{\gamma^3 R_{\min}^4}{3N R_{\max}^2} \geq \frac{\gamma^3 (R_{\min}^2 - 2\frac{C}{\sqrt{N}}\epsilon + \epsilon^2)^2}{3N(R_{\max}^2 + 2\frac{C}{\sqrt{N}}\epsilon + \epsilon^2)} - 2\frac{C}{\sqrt{N}}\epsilon - \epsilon^2. \tag{A51}$$

With $A := 2\frac{C}{\sqrt{N}}\epsilon + \epsilon^2 (\geq 0)$,

$$\frac{\gamma^3 R_{\min}^4}{3N R_{\max}^2} \geq \frac{\gamma^3 (R_{\min}^2 + A)^2}{3N(R_{\max}^2 + A)} - A \geq \frac{\gamma^3 (R_{\min}^2 - 2\frac{C}{\sqrt{N}}\epsilon + \epsilon^2)^2}{3N(R_{\max}^2 + A)} - A. \tag{A52}$$

Rearranging this,

$$\gamma^3 R_{\min}^4 + (3N R_{\max}^2 - 2\gamma^3 R_{\min}^2)R_{\max}^2 + (3N - \gamma^3)R_{\max}^2 A > 0. \tag{A53}$$

Thus, the above inequality holds. Finally, the claim is established. $\square$

We have represented an upper bound of $\langle \boldsymbol{x}_n, \boldsymbol{q}/\|\boldsymbol{q}\| \rangle$ as $C/\sqrt{N}$. Alternatively, we can use $p_{\max}$ to represent an upper bound of $\langle \boldsymbol{x}_n, \boldsymbol{q}/\|\boldsymbol{q}\| \rangle$ as follows:

$$\left\langle \boldsymbol{x}_n, \frac{\boldsymbol{q}}{\|\boldsymbol{q}\|} \right\rangle \leq \frac{3R_{\max}^2(R_{\max}^2 + N p_{\max})}{\gamma R_{\min}\sqrt{N(\gamma^2 R_{\min}^4 - 3N R_{\max}^2 p_{\max})}} =: C'. \tag{A54}$$

Using this bound, for a sufficiently large $N$, we can obtain a similar result as follows:

$$\frac{\gamma^3 (R_{\min}^2 - 2C'\epsilon + \epsilon^2)^2}{3N(R_{\max}^2 + 2C'\epsilon + \epsilon^2)} - 2C'\epsilon - \epsilon^2 \geq p_{\max}. \tag{A55}$$

As Ineq. (A54) is tighter than Ineq. (A42), Ineq. (A55) is tighter than Ineq. (A26). However, to interpret the restriction on $p_{\max}$, we employ Ineq. (A26), which contains $p_{\max}$ only in the right term. The left and right terms of Ineq. (A55) include $p_{\max}$, and it is more complex to determine the constraint on $p_{\max}$.

**Theorem C.2** (Decision boundary when learning from geometry-inspired perturbations on natural samples). *Let $f$ be a one-hidden-layer neural network trained on geometry-inspired perturbations on natural samples (cf. Eq. (1) and Definition 3.2(a)) with Setting 3.1. If Ineq. (A26) holds, then, with $t \to \infty$, the decision boundary of $f$ is given by Eq. (3).*

*Proof.* By Lemma C.1, if Ineq. (A26) holds, we can use Corollary 3.4. The decision boundary is

$$\mathrm{sgn}(f(\boldsymbol{z}; \boldsymbol{W}^{\mathrm{adv}})) = \mathrm{sgn}\left( \sum_{n=1}^N \lambda_n^{\mathrm{adv}} y_n^{\mathrm{adv}} \langle \boldsymbol{x}_n^{\mathrm{adv}}, \boldsymbol{z} \rangle \right) \tag{A56}$$

$$= \mathrm{sgn}\left( \sum_{n=1}^N \lambda_n^{\mathrm{adv}} y_n^{\mathrm{adv}} \langle \boldsymbol{x}_n, \boldsymbol{z} \rangle + \sum_{n=1}^N \lambda_n^{\mathrm{adv}} y_n^{\mathrm{adv}} \epsilon y_n^{\mathrm{adv}} \left\langle \frac{\boldsymbol{q}}{\|\boldsymbol{q}\|}, \boldsymbol{z} \right\rangle \right) \tag{A57}$$

$$= \mathrm{sgn}\left( \sum_{n=1}^N \lambda_n^{\mathrm{adv}} y_n^{\mathrm{adv}} \langle \boldsymbol{x}_n, \boldsymbol{z} \rangle + \left( \sum_{n=1}^N \lambda_n^{\mathrm{adv}} \right) \epsilon \left\langle \frac{\boldsymbol{q}}{\|\boldsymbol{q}\|}, \boldsymbol{z} \right\rangle \right) \tag{A58}$$

$$= \mathrm{sgn}\left( \frac{\sum_{n=1}^N \lambda_n^{\mathrm{adv}} y_n^{\mathrm{adv}} \langle \boldsymbol{x}_n, \boldsymbol{z} \rangle}{\sum_{n=1}^N \lambda_n^{\mathrm{adv}}} + \epsilon \frac{\langle \boldsymbol{q}, \boldsymbol{z} \rangle}{\|\boldsymbol{q}\|} \right) \tag{A59}$$

$$= \mathrm{sgn}\left( \frac{\sum_{n=1}^N \lambda_n^{\mathrm{adv}} y_n^{\mathrm{adv}} \langle \boldsymbol{x}_n, \boldsymbol{z} \rangle}{\sum_{n=1}^N \lambda_n^{\mathrm{adv}}} + \epsilon \frac{f^{\mathrm{bdy}}(\boldsymbol{z})}{\|\boldsymbol{q}\|} \right). \tag{A60}$$

$\square$

**Theorem 4.1** (Decision boundary when learning from geometry-inspired perturbations on natural samples)**.** *Let $f$ be a one-hidden-layer neural network trained on geometry-inspired perturbations on natural samples (cf. Eq. (1) and Definition 3.2(a)) with Setting 3.1. If $N > C^2/R_{\min}^2$ and*

$$\frac{\gamma^3(R_{\min}^2 - 2\frac{C}{\sqrt{N}}\epsilon + \epsilon^2)^2}{3N(R_{\max}^2 + 2\frac{C}{\sqrt{N}}\epsilon + \epsilon^2)} - 2\frac{C}{\sqrt{N}}\epsilon - \epsilon^2 \geq p_{\max} \tag{2}$$

*with $C := \frac{3R_{\max}^4 + \gamma^3 R_{\min}^4}{\gamma^2 R_{\min}^3 \sqrt{1-\gamma}}$, then, with $t \to \infty$, the decision boundary of $f$ is given by*

$$f_{\mathrm{adv}}^{\mathrm{bdy}}(\boldsymbol{z}) := \underbrace{\frac{\sum_{n=1}^N \lambda_n^{\mathrm{adv}} y_n^{\mathrm{adv}} \langle \boldsymbol{x}_n, \boldsymbol{z} \rangle}{\sum_{n=1}^N \lambda_n^{\mathrm{adv}}}}_{\text{Effect of learning from mislabeled natural samples}} + \underbrace{\epsilon \frac{f^{\mathrm{bdy}}(\boldsymbol{z})}{\| \sum_{n=1}^N \lambda_n y_n \boldsymbol{x}_n \|}}_{\text{Effect of learning from perturbations}}. \tag{3}$$

*Proof.* This is the special case of Theorem C.2 for $N > C^2/R_{\min}^2$. $\qquad\square$

## C.2   Limiting Behavior of Learning from Geometry-Inspired Perturbations on Natural Samples with Deterministic Labels

In this section, we consider learning from geometry-inspired perturbations on natural samples with deterministic adversarial labels. In Proposition C.4, we show that the effects of perturbations and mislabeled natural samples grow at the same speed with respect to an input dimension and the number of training samples, suggesting that learning from perturbations is feasible on a high-dimensional dataset with many samples. To prove Proposition C.4, we first prepare Lemma C.3.

**Lemma C.3** (Order of norm of weighted sum of training data)**.** *Assume $\gamma^3 R_{\min}^4 / 3N R_{\max}^2 \geq p_{\max}$ and $\|\boldsymbol{x}_n\| = \Theta(\sqrt{d})$ for any $n \in [N]$. Then,*

$$\left\| \sum_{n=1}^N \lambda_n y_n \boldsymbol{x}_n \right\| = \Theta\left( \sqrt{\frac{N}{d}} \right). \tag{A61}$$

*Proof.* By definition in Theorem 3.3, $\lambda_n = \Theta(1/d)$. By the assumption, $p_{\max} = \mathcal{O}(d/N)$. A lower bound is

$$\left\| \sum_{n=1}^N \lambda_n y_n \boldsymbol{x}_n \right\| = \sqrt{\sum_{n=1}^N \lambda_n \left( \lambda_n \|\boldsymbol{x}_n\|^2 + \sum_{k \neq n} \lambda_k y_n y_k \langle \boldsymbol{x}_n, \boldsymbol{x}_k \rangle \right)} \tag{A62}$$

$$\geq \sqrt{\sum_{n=1}^N \lambda_n \left( \lambda_n \|\boldsymbol{x}_n\|^2 - \sum_{k \neq n} \lambda_k p_{\max} \right)} \tag{A63}$$

$$= \Omega\left( \sqrt{\frac{N}{d}} \right). \tag{A64}$$

Similarly, an upper bound is $\mathcal{O}(\sqrt{N/d})$. Note that the radicand of the lower bound is positive because the following inequality holds:

$$\lambda_n \|\boldsymbol{x}_n\|^2 - \sum_{k \neq n} \lambda_k p_{\max} \geq \frac{R_{\min}^2}{2R_{\max}^2} - \frac{\gamma R_{\min}^2}{2R_{\max}^2} \geq \frac{(1-\gamma)R_{\min}^2}{2R_{\max}^2} > 0. \tag{A65}$$

$\square$

**Proposition C.4** (Limiting behavior of learning from geometry-inspired perturbations on natural samples (deterministic label))**.** *Suppose that Ineq. (2) holds. Assume $\|\boldsymbol{x}_n\| = \Theta(\sqrt{d})$ for any*

$n \in [N]$ *and* $\|\boldsymbol{z}\| = \Theta(\sqrt{d})$. *Assume*

$$\sum_{n=1}^{N} \lambda_n |\langle \boldsymbol{x}_n, \boldsymbol{z} \rangle| = \Theta(g_1(N, d)) \Rightarrow \left| \sum_{n=1}^{N} \lambda_n y_n \langle \boldsymbol{x}_n, \boldsymbol{z} \rangle \right| = \Theta(g_1(N, d)), \tag{A66}$$

$$\sum_{n=1}^{N} \lambda_n |\langle \boldsymbol{x}_n, \boldsymbol{z} \rangle| = \Theta(g_2(N, d)) \Rightarrow \left| \sum_{n=1}^{N} \lambda_n y_n \langle \boldsymbol{x}_n, \boldsymbol{z} \rangle \right| = \Theta(g_2(N, d)). \tag{A67}$$

*where $g_1$ and $g_2$ are positive functions of $N$ and $d$. Then, the following statements hold:*

(a) *For any $\boldsymbol{z}$, $|T_1(\boldsymbol{z})| = \Theta(g_3(N, d)) \Leftrightarrow |T_2(\boldsymbol{z})| = \Theta(g_3(N, d))$, where $g_3$ is a positive function of $N$ and $d$.*

(b) *For $z = \sum_{n=1}^{N} \Theta(1/\sqrt{N}) \boldsymbol{x}_n$, $|T_1(z)| = \Theta(d/\sqrt{N})$ and $|T_2(z)| = \Theta(d/\sqrt{N})$.*

(c) *For $z = \Theta(1) \boldsymbol{x}_1$, $|T_1(z)| = \Theta(d/N)$ and $|T_2(z)| = \Theta(d/N)$.*

*Proof.* Under Ineq. (2), $\epsilon = \mathcal{O}(\sqrt{d/N})$. Because we can set $\epsilon$ freely under Ineq. (2), we consider $\epsilon = \Theta(\sqrt{d/N})$ which maximizes the effect of learning from perturbations.

**(a)** By $\lambda_n = \Theta(1/d)$ and $\lambda_n^{\text{adv}} = \Theta(1/d)$ (cf. Theorem 3.3 and Corollary 3.4),

$$\sum_{n=1}^{N} \lambda_n |\langle \boldsymbol{x}_n, \boldsymbol{z} \rangle| = \Theta(g(N, d)) \Leftrightarrow \sum_{n=1}^{N} \lambda_n^{\text{adv}} |\langle \boldsymbol{x}_n, \boldsymbol{z} \rangle| = \Theta(g(N, d)). \tag{A68}$$

Under the assumption,

$$\sum_{n=1}^{N} \lambda_n |\langle \boldsymbol{x}_n, \boldsymbol{z} \rangle| = \Theta(g(N, d)), \quad \sum_{n=1}^{N} \lambda_n^{\text{adv}} |\langle \boldsymbol{x}_n, \boldsymbol{z} \rangle| = \Theta(g(N, d)) \tag{A69}$$

$$\Rightarrow \left| \sum_{n=1}^{N} \lambda_n y_n \langle \boldsymbol{x}_n, \boldsymbol{z} \rangle \right| = \Theta(g(N, d)), \quad \left| \sum_{n=1}^{N} \lambda_n^{\text{adv}} y_n^{\text{adv}} \langle \boldsymbol{x}_n, \boldsymbol{z} \rangle \right| = \Theta(g(N, d)). \tag{A70}$$

By $\| \sum_{n=1}^{N} \lambda_n y_n \boldsymbol{x}_n \| = \Theta(\sqrt{N/d})$ (cf. Lemma C.3)

$$|T_1(\boldsymbol{z})| = \frac{\Theta(g(N, d))}{\Theta\left(\frac{N}{d}\right)} = \Theta\left(\frac{dg(N, d)}{N}\right), \tag{A71}$$

$$|T_2(\boldsymbol{z})| = \Theta\left(\sqrt{\frac{d}{N}}\right) \frac{\Theta(g(N, d))}{\Theta\left(\sqrt{\frac{N}{d}}\right)} = \Theta\left(\frac{dg(N, d)}{N}\right). \tag{A72}$$

**(b)** Since $\sum_{n=1}^{N} \lambda_n |\langle \boldsymbol{x}_n, \boldsymbol{z} \rangle| = \Theta(\sqrt{N})$ and $\sum_{n=1}^{N} \lambda_n^{\text{adv}} |\langle \boldsymbol{x}_n, \boldsymbol{z} \rangle| = \Theta(\sqrt{N})$, $T_1(\boldsymbol{z}) = \Theta(d/\sqrt{N})$ and $T_2(\boldsymbol{z}) = \Theta(d/\sqrt{N})$.

**(c)** Since $\sum_{n=1}^{N} \lambda_n |\langle \boldsymbol{x}_n, \boldsymbol{z} \rangle| = \Theta(1)$ and $\sum_{n=1}^{N} \lambda_n^{\text{adv}} |\langle \boldsymbol{x}_n, \boldsymbol{z} \rangle| = \Theta(1)$, $T_1(\boldsymbol{z}) = \Theta(d/N)$ and $T_2(\boldsymbol{z}) = \Theta(d/N)$. □

### C.3 LIMITING BEHAVIOR OF LEARNING FROM GEOMETRY-INSPIRED PERTURBATIONS ON NATURAL SAMPLES WITH RANDOM LABELS

In this section, we consider learning from geometry-inspired perturbations on natural samples with random adversarial labels $y_n^{\text{adv}} \sim U(\pm 1)$. To prove the key theorem, Proposition C.8, we prepare Lemmas C.6 and C.7. In addition, we consider Lemma C.5 to prove Lemma C.6. Finally, based on Proposition C.8, we demonstrate the matching of decision boundaries between learning from standard samples and adversarial perturbations, Theorem 4.2.

First, we show that assumptions $\|\boldsymbol{z}\| = \Theta(\sqrt{d})$ and $|\langle \boldsymbol{x}_n, \boldsymbol{x}_k \rangle| = \mathcal{O}(d/N)$ restrict the correlation between $\boldsymbol{z}$ and $\{\boldsymbol{x}_n\}_{n=1}^{N}$. In other words, $\boldsymbol{z}$ cannot be strongly correlated with many training samples. This lemma is used to prove Lemma C.6.

**Lemma C.5** (Restriction on correlation). *For any $n, k \in [N], n \neq k$, assume $\|\boldsymbol{x}_n\| = \Theta(\sqrt{d})$, $\|\boldsymbol{z}\| = \Theta(\sqrt{d})$, and $|\langle \boldsymbol{x}_n, \boldsymbol{x}_k \rangle| = \mathcal{O}(d/N)$. Then, there exist at most $\mathcal{O}(\min(N^{-2\alpha}, N))$ instances of $n$ that satisfy $|\langle \boldsymbol{x}_n, \boldsymbol{z} \rangle| = \Theta(N^\alpha d^\beta)$ with $\alpha \leq 0$ and $\beta \leq 1$.*

*Proof.* For each $n$, let $\psi_n := \mathrm{sgn}(\langle \boldsymbol{x}_n, \boldsymbol{z} \rangle)$. For $\alpha \leq 0$ and $\beta \leq 1$, denote the number of samples such that $|\langle \boldsymbol{x}_n, \boldsymbol{z} \rangle| = \Theta(N^\alpha d^\beta)$ holds by $[N]_{\alpha,\beta} := \{n \in [N] : |\langle \boldsymbol{x}_n, \boldsymbol{z} \rangle| = \Theta(N^\alpha d^\beta)\}$. We define $\delta \leq 1$ such that $|[N]_{\alpha,\beta}| = \Theta(N^\delta)$. Then,

$$\sum_{n \in [N]_{\alpha,\beta}} |\langle \boldsymbol{x}_n, \boldsymbol{z} \rangle| = \sum_{n \in [N]_{\alpha,\beta}} \langle \psi_n \boldsymbol{x}_n, \boldsymbol{z} \rangle = \sum_{n \in [N]_{\alpha,\beta}} \Theta(N^\alpha d^\beta) = \Theta(N^{\alpha+\delta} d^\beta). \tag{A73}$$

By the Cauchy–Schwarz inequality,

$$\sum_{n \in [N]_{\alpha,\beta}} \langle \psi_n \boldsymbol{x}_n, \boldsymbol{z} \rangle = \left\langle \sum_{n \in [N]_{\alpha,\beta}} \psi_n \boldsymbol{x}_n, \boldsymbol{z} \right\rangle \leq \left\| \sum_{n \in [N]_{\alpha,\beta}} \psi_n \boldsymbol{x}_n \right\| \|\boldsymbol{z}\|. \tag{A74}$$

Note that

$$\left\| \sum_{n \in [N]_{\alpha,\beta}} \psi_n \boldsymbol{x}_n \right\| = \sqrt{\sum_{n \in [N]_{\alpha,\beta}} \|\boldsymbol{x}_n\|^2 + \sum_{n \in [N]_{\alpha,\beta}} \sum_{k \neq n} \psi_n \psi_k \langle \boldsymbol{x}_n, \boldsymbol{x}_k \rangle} \tag{A75}$$

$$= \sqrt{\Theta(N^\delta d) \pm \Theta(N^{2\delta}) \mathcal{O}\left(\frac{d}{N}\right)} \tag{A76}$$

$$= \Theta(\sqrt{N^\delta d}). \tag{A77}$$

Thus, $\sum_{n \in [N]_{\alpha,\beta}} \langle \psi_n \boldsymbol{x}_n, \boldsymbol{z} \rangle = \mathcal{O}(\sqrt{N^\delta d})$. Comparing this with Eq. (A73), $\alpha + \delta \leq \delta/2 \Leftrightarrow \delta \leq -2\alpha$. $\qquad \square$

Then we compare the growth rates of $\sum_{n=1}^{N} |\langle \boldsymbol{x}_n, \boldsymbol{z} \rangle|$ and $\sum_{n=1}^{N} \langle \boldsymbol{x}_n, \boldsymbol{z} \rangle^2$ to evaluate the growth rates of $|T_1(\boldsymbol{z})|$ and $|T_2(\boldsymbol{z})|$ in Proposition C.8.

**Lemma C.6** (Comparison between sums of absolute and squared inner products). *For any $n, k \in [N], n \neq k$, assume $\|\boldsymbol{x}_n\| = \Theta(\sqrt{d})$, $\|\boldsymbol{z}\| = \Theta(\sqrt{d})$, and $|\langle \boldsymbol{x}_n, \boldsymbol{x}_k \rangle| = \mathcal{O}(d/N)$. Then, the growth rates of $\sum_{n=1}^{N} |\langle \boldsymbol{x}_n, \boldsymbol{z} \rangle|$ and $\Theta(1/d) \sum_{n=1}^{N} \langle \boldsymbol{x}_n, \boldsymbol{z} \rangle^2$ are the same if and only if $|\langle \boldsymbol{x}_n, \boldsymbol{z} \rangle| = 0$ for every $n$, or there exists $n$ such that $|\langle \boldsymbol{x}_n, \boldsymbol{z} \rangle| = \Theta(d)$ and $\sum_{n: |\langle \boldsymbol{x}_n, \boldsymbol{z} \rangle| \neq \Theta(d)} |\langle \boldsymbol{x}_n, \boldsymbol{z} \rangle| = \mathcal{O}(d)$. Otherwise, $\sum_{n=1}^{N} |\langle \boldsymbol{x}_n, \boldsymbol{z} \rangle|$ grows faster than $\Theta(1/d) \sum_{n=1}^{N} \langle \boldsymbol{x}_n, \boldsymbol{z} \rangle^2$.*

*Proof.* First, we summarize the content of this proof as follows:

(A) If $|\langle \boldsymbol{x}_n, \boldsymbol{z} \rangle| = 0$ for every $n$, then $\sum_{n=1}^{N} |\langle \boldsymbol{x}_n, \boldsymbol{z} \rangle| = \Theta(1/d) \sum_{n=1}^{N} \langle \boldsymbol{x}_n, \boldsymbol{z} \rangle^2 = 0$.

(B) Assume that there exists $n$ such that $|\langle \boldsymbol{x}_n, \boldsymbol{z} \rangle| > 0$.

    (B-a) If there is no $n$ such that $|\langle \boldsymbol{x}_n, \boldsymbol{z} \rangle| = \Theta(d)$, then $\sum_{n=1}^{N} |\langle \boldsymbol{x}_n, \boldsymbol{z} \rangle|$ grows faster than $\Theta(1/d) \sum_{n=1}^{N} \langle \boldsymbol{x}_n, \boldsymbol{z} \rangle^2$.

    (B-b) Assume that there exists $n$ such that $|\langle \boldsymbol{x}_n, \boldsymbol{z} \rangle| = \Theta(d)$.

        (B-b-I) If $\sum_{n: |\langle \boldsymbol{x}_n, \boldsymbol{z} \rangle| \neq \Theta(d)} |\langle \boldsymbol{x}_n, \boldsymbol{z} \rangle| > \Omega(d)$, then $\sum_{n=1}^{N} |\langle \boldsymbol{x}_n, \boldsymbol{z} \rangle|$ grows faster than $\Theta(1/d) \sum_{n=1}^{N} \langle \boldsymbol{x}_n, \boldsymbol{z} \rangle^2$.

        (B-b-II) If $\sum_{n: |\langle \boldsymbol{x}_n, \boldsymbol{z} \rangle| \neq \Theta(d)} |\langle \boldsymbol{x}_n, \boldsymbol{z} \rangle| = \mathcal{O}(d)$, then $\sum_{n=1}^{N} |\langle \boldsymbol{x}_n, \boldsymbol{z} \rangle| = \Theta(d)$ and $\Theta(1/d) \sum_{n=1}^{N} \langle \boldsymbol{x}_n, \boldsymbol{z} \rangle^2 = \Theta(d)$.

We use $[N]_{\alpha,\beta}$ in the proof of Lemma C.5. Denote the number of elements in $[N]_{\alpha,\beta}$ by $C(\alpha,\beta) := |[N]_{\alpha,\beta}|$. Denote the set of $(\alpha,\beta)$ by $S := \{(\alpha,\beta) : C(\alpha,\beta) > 0\}$. We can write $\sum_{n=1}^{N} |\langle \boldsymbol{x}_n, \boldsymbol{z} \rangle|$ and $\Theta(1/d) \sum_{n=1}^{N} \langle \boldsymbol{x}_n, \boldsymbol{z} \rangle^2$ as follows:

$$\sum_{n=1}^{N} |\langle \boldsymbol{x}_n, \boldsymbol{z} \rangle| = \sum_{(\alpha,\beta) \in S} C(\alpha,\beta) \Theta(N^\alpha d^\beta), \tag{A78}$$

$$\Theta\left(\frac{1}{d}\right) \sum_{n=1}^{N} \langle \boldsymbol{x}_n, \boldsymbol{z} \rangle^2 = \sum_{(\alpha,\beta) \in S} C(\alpha,\beta) \Theta(N^{2\alpha} d^{2\beta-1}). \tag{A79}$$

**(A)** This is trivial.

**(B-a)** Assume that there exists $n$ such that $|\langle \boldsymbol{x}_n, \boldsymbol{z} \rangle| > 0$. Because $N^\alpha d^\beta$ grows faster than or equal to $N^{2\alpha} d^{2\beta-1}$ for $\alpha \leq 0$ and $\beta \leq 1$, $\sum_{n=1}^{N} |\langle \boldsymbol{x}_n, \boldsymbol{z} \rangle|$ grows faster than or equal to $\Theta(1/d) \sum_{n=1}^{N} \langle \boldsymbol{x}_n, \boldsymbol{z} \rangle^2$. The growth rates of $N^\alpha d^\beta$ and $N^{2\alpha} d^{2\beta-1}$ are consistent if and only if $\alpha = 0$ and $\beta = 1$. Thus, if there is no $n$ such that $|\langle \boldsymbol{x}_n, \boldsymbol{z} \rangle| = \Theta(d)$, i.e., $C(0,1) > 0$, then $\sum_{n=1}^{N} |\langle \boldsymbol{x}_n, \boldsymbol{z} \rangle|$ grows faster than $\Theta(1/d) \sum_{n=1}^{N} \langle \boldsymbol{x}_n, \boldsymbol{z} \rangle^2$.

**(B-b-I and -II)** Assume that there exists $n$ such that $|\langle \boldsymbol{x}_n, \boldsymbol{z} \rangle| = \Theta(d)$. By Lemma C.5, $C(0,1) = \Theta(1)$. Let $S' := S \setminus \{(0,1)\}$. The above equations can be rearranged as follows:

$$\sum_{n=1}^{N} |\langle \boldsymbol{x}_n, \boldsymbol{z} \rangle| = \Theta(d) + \sum_{(\alpha,\beta) \in S'} C(\alpha,\beta) \Theta(N^\alpha d^\beta), \tag{A80}$$

$$\Theta\left(\frac{1}{d}\right) \sum_{n=1}^{N} \langle \boldsymbol{x}_n, \boldsymbol{z} \rangle^2 = \Theta(d) + \sum_{(\alpha,\beta) \in S'} C(\alpha,\beta) \Theta(N^{2\alpha} d^{2\beta-1}). \tag{A81}$$

Since $N^\alpha d^\beta$ grows faster than $N^{2\alpha} d^{2\beta-1}$ for $\alpha < 0$ and $\beta < 1$, $\sum_{(\alpha,\beta) \in S'} C(\alpha,\beta) \Theta(N^\alpha d^\beta)$ grows faster than $\sum_{(\alpha,\beta) \in S'} C(\alpha,\beta) \Theta(N^{2\alpha} d^{2\beta-1})$. If $\sum_{(\alpha,\beta) \in S'} C(\alpha,\beta) \Theta(N^\alpha d^\beta)$ determines the growth rate of $\sum_{n=1}^{N} |\langle \boldsymbol{x}_n, \boldsymbol{z} \rangle|$, i.e., $\sum_{(\alpha,\beta) \in S'} C(\alpha,\beta) \Theta(N^\alpha d^\beta) > \Omega(d)$, then $\sum_{n=1}^{N} |\langle \boldsymbol{x}_n, \boldsymbol{z} \rangle|$ grows faster than $\Theta(1/d) \sum_{n=1}^{N} \langle \boldsymbol{x}_n, \boldsymbol{z} \rangle^2$. In contrast, if $\sum_{(\alpha,\beta) \in S'} C(\alpha,\beta) \Theta(N^\alpha d^\beta)$ does not change the growth rate of $\sum_{n=1}^{N} |\langle \boldsymbol{x}_n, \boldsymbol{z} \rangle|$, i.e., $\sum_{(\alpha,\beta) \in S'} C(\alpha,\beta) \Theta(N^\alpha d^\beta) = \mathcal{O}(d)$, then $\sum_{n=1}^{N} |\langle \boldsymbol{x}_n, \boldsymbol{z} \rangle| = \Theta(d)$ and $\Theta(1/d) \sum_{n=1}^{N} \langle \boldsymbol{x}_n, \boldsymbol{z} \rangle^2 = \Theta(d)$; namely, their growth rates are the same. □

Then we prepare a concentration inequality to evaluate an upper bound of $T_1(\boldsymbol{z})$ in Proposition C.8.

**Lemma C.7** (Concentration inequality). *Let $\{x_n\}_{n=1}^{N}$ be $N \in \mathbb{N}$ independent random variables. Assume that $\boldsymbol{x}_n$ is sampled from $[a_n, b_n]$ and $\mathbb{E}[\boldsymbol{x}_n] = 0$ for each $n \in [N]$. Then, for $t > 0$,*

$$\mathbb{P}\left[\left|\sum_{n=1}^{N} \boldsymbol{x}_n\right| \geq t\right] \leq 2 \exp\left(\frac{1}{8} \sum_{n=1}^{N} (b_n - a_n)^2 - t\right). \tag{A82}$$

*Proof.* By Markov's inequality, for $t > 0$,

$$\mathbb{P}\left[\sum_{n=1}^{N} \boldsymbol{x}_n \geq t\right] \leq \frac{\mathbb{E}\left[\exp\left(\sum_{n=1}^{N} \boldsymbol{x}_n\right)\right]}{e^t} = \frac{\prod_{n=1}^{N} \mathbb{E}[\exp(x_n)]}{e^t}. \tag{A83}$$

By Hoeffding's lemma,

$$\mathbb{P}\left[\sum_{n=1}^{N} \boldsymbol{x}_n \geq t\right] \leq \frac{\prod_{n=1}^{N} \exp\left((b_n - a_n)^2/8\right)}{e^t} = \exp\left(\frac{1}{8} \sum_{n=1}^{N} (b_n - a_n)^2 - t\right). \tag{A84}$$

We can derive the same inequality for $\mathbb{P}[-\sum_{n=1}^{N} \boldsymbol{x}_n \geq t]$. □

While the concentration inequality, Lemma C.7, is weaker than Hoeffding's inequality, we use it for a simple proof of Proposition C.8. The proof of Proposition C.8 requires us to consider the probability $\mathbb{P}[|\sum_{n=1}^{N} \lambda_n^{\mathrm{adv}} y_n^{\mathrm{adv}} \langle \boldsymbol{x}_n, \boldsymbol{z} \rangle| > \sum_{n=1}^{N} \lambda_n |\langle \boldsymbol{x}_n, \boldsymbol{z} \rangle|]$. Using Lemma C.7, this can be represented as follows:

$$
\mathbb{P}\left[\left|\sum_{n=1}^{N} \lambda_n^{\mathrm{adv}} y_n^{\mathrm{adv}} \langle \boldsymbol{x}_n, \boldsymbol{z} \rangle\right| > \sum_{n=1}^{N} \lambda_n |\langle \boldsymbol{x}_n, \boldsymbol{z} \rangle|\right]
$$
$$
\leq 2 \exp\left(\frac{1}{2} \sum_{n=1}^{N} \lambda_n^{\mathrm{adv}\,2} \langle \boldsymbol{x}_n, \boldsymbol{z} \rangle^2 - \sum_{n=1}^{N} \lambda_n |\langle \boldsymbol{x}_n, \boldsymbol{z} \rangle|\right). \tag{A85}
$$

Using Hoeffding's inequality, it can also be represented as follows:

$$
\mathbb{P}\left[\left|\sum_{n=1}^{N} \lambda_n^{\mathrm{adv}} y_n^{\mathrm{adv}} \langle \boldsymbol{x}_n, \boldsymbol{z} \rangle\right| > \sum_{n=1}^{N} \lambda_n |\langle \boldsymbol{x}_n, \boldsymbol{z} \rangle|\right] \leq 2 \exp\left(-\frac{(\sum_{n=1}^{N} \lambda_n |\langle \boldsymbol{x}_n, \boldsymbol{z} \rangle|)^2}{2 \sum_{n=1}^{N} \lambda_n^{\mathrm{adv}\,2} \langle \boldsymbol{x}_n, \boldsymbol{z} \rangle^2}\right). \tag{A86}
$$

In the former case, the growth rates of $\sum_{n=1}^{N} \lambda_n^{\mathrm{adv}\,2} \langle \boldsymbol{x}_n, \boldsymbol{z} \rangle^2$ and $\sum_{n=1}^{N} \lambda_n |\langle \boldsymbol{x}_n, \boldsymbol{z} \rangle|$ are the main concern (cf. Lemma C.6). In the latter case, the focus is on the growth rates of $\sum_{n=1}^{N} \lambda_n^{\mathrm{adv}\,2} \langle \boldsymbol{x}_n, \boldsymbol{z} \rangle^2$ and $(\sum_{n=1}^{N} \lambda_n |\langle \boldsymbol{x}_n, \boldsymbol{z} \rangle|)^2$, which present a more complex scenario than the former. Thus, we use Lemma C.7 to prove Proposition C.8.

Proposition C.8 describes the limiting behavior of the two components of the decision boundary: the effect of learning from mislabeled natural samples $T_1(\boldsymbol{z})$ and from perturbations $T_2(\boldsymbol{z})$.

**Proposition C.8** (Limiting behavior of learning from geometry-inspired perturbations on natural samples (random label)). *Suppose that Ineq. (2) holds. Assume $\|\boldsymbol{x}_n\| = \Theta(\sqrt{d})$ for any $n \in [N]$ and $\|\boldsymbol{z}\| = \Theta(\sqrt{d})$. Suppose that $y_n^{\mathrm{adv}}$ is randomly sampled from $\{\pm 1\}$ for each $n$. Assume*

$$
\sum_{n=1}^{N} \lambda_n |\langle \boldsymbol{x}_n, \boldsymbol{z} \rangle| = \Theta(g(N, d)) \Rightarrow \left|\sum_{n=1}^{N} \lambda_n y_n \langle \boldsymbol{x}_n, \boldsymbol{z} \rangle\right| = \Theta(g(N, d)). \tag{A87}
$$

*where $g$ is a positive function of $N$ and $d$. Then, the following statements hold with probability at least 99.99%:*

(a) *Assume that there exists $n$ such that $|\langle \boldsymbol{x}_n, \boldsymbol{z} \rangle| > 0$. If there is no $n$ such that $|\langle \boldsymbol{x}_n, \boldsymbol{z} \rangle| = \Theta(d)$ or $\sum_{n:|\langle \boldsymbol{x}_n, \boldsymbol{z} \rangle| \neq \Theta(d)} |\langle \boldsymbol{x}_n, \boldsymbol{z} \rangle| = \mathcal{O}(d)$ does not hold, with $N, d \to \infty$, then $|T_2(z)| > |T_1(z)|$.*
(b) *For $z = \sum_{n=1}^{N} \Theta(1/\sqrt{N}) \boldsymbol{x}_n$, $|T_1(z)| = \mathcal{O}(d/N)$ and $|T_2(z)| = \Theta(d/\sqrt{N})$.*
(c) *For $z = \Theta(1)\boldsymbol{x}_1$, $|T_1(z)| = \mathcal{O}(d/N)$ and $|T_2(z)| = \Theta(d/N)$.*

*Proof.* Similarly to the proof of Proposition C.4, if $\sum_{n=1}^{N} \lambda_n |\langle \boldsymbol{x}_n, \boldsymbol{z} \rangle| = \Theta(g(N, d))$,

$$
|T_1(\boldsymbol{z})| = \Theta\left(\frac{d}{N}\right) \left|\sum_{n=1}^{N} \lambda_n^{\mathrm{adv}} y_n^{\mathrm{adv}} \langle \boldsymbol{x}_n, \boldsymbol{z} \rangle\right|, \qquad |T_2(\boldsymbol{z})| = \Theta\left(\frac{dg(N, d)}{N}\right). \tag{A88}
$$

**(a)** By Lemma C.7,

$$
\mathbb{P}\left[\left|\sum_{n=1}^{N} \lambda_n^{\mathrm{adv}} y_n^{\mathrm{adv}} \langle \boldsymbol{x}_n, \boldsymbol{z} \rangle\right| > t\right] \leq 2 \exp\left(\frac{1}{2} \sum_{n=1}^{N} \lambda_n^{\mathrm{adv}\,2} \langle \boldsymbol{x}_n, \boldsymbol{z} \rangle^2 - t\right). \tag{A89}
$$

Thus, $|\sum_{n=1}^{N} \lambda_n^{\mathrm{adv}} y_n^{\mathrm{adv}} \langle \boldsymbol{x}_n, \boldsymbol{z} \rangle| = \mathcal{O}(h(N, d))$ if $\sum_{n=1}^{N} \lambda_n^{\mathrm{adv}\,2} \langle \boldsymbol{x}_n, \boldsymbol{z} \rangle^2 = \mathcal{O}(h(N, d))$, where $h(N, d)$ is a positive function of $N$ and $d$, with sufficiently high probability. By Lemma C.6, if there is no $n$ such that $|\langle \boldsymbol{x}_n, \boldsymbol{z} \rangle| = \Theta(d)$ or $\sum_{n:|\langle \boldsymbol{x}_n, \boldsymbol{z} \rangle| \neq \Theta(d)} |\langle \boldsymbol{x}_n, \boldsymbol{z} \rangle| = \mathcal{O}(d)$ does not hold, $g(N, d)$ grows faster than $h(N, d)$. Thus, $|T_2(z)|$ grows faster than $|T_1(z)|$; namely, if $N, d \to \infty$, $|T_2(z)|$ becomes larger than $|T_1(z)|$.

**(b)** The proof of $|T_2(z)| = \Theta(d/\sqrt{N})$ can be found in the proof of Proposition C.4. For $z = \sum_{n=1}^{N} \Theta(1/\sqrt{N})\boldsymbol{x}_n$, $\sum_{n=1}^{N} \lambda_n^{\mathrm{adv}^2} \langle \boldsymbol{x}_n, \boldsymbol{z} \rangle^2 = \mathcal{O}(1)$. Thus, $h(N,d) = \mathcal{O}(1)$, and $|T_1(z)| = \mathcal{O}(d/N)$.

**(c)** The proof of $|T_2(z)| = \Theta(d/N)$ can be found in the proof of Proposition C.4. For $z = \Theta(1)\boldsymbol{x}_1$, $\sum_{n=1}^{N} \lambda_n^{\mathrm{adv}^2} \langle \boldsymbol{x}_n, \boldsymbol{z} \rangle^2 = \mathcal{O}(1)$. Thus, $h(N,d) = \mathcal{O}(1)$, and $|T_1(z)| = \mathcal{O}(d/N)$. $\qquad\square$

In Proposition C.8, we show that if $N, d \to \infty$, the effect of learning from perturbations $|T_2(\boldsymbol{z})|$ exceeds that from mislabeled samples $|T_1(\boldsymbol{z})|$ except for specific inputs. Consequently, we can justify learning from perturbations on natural samples as follows:

---

**Theorem 4.2** (Consistent decision of learning from geometry-inspired perturbations on natural samples)**.** *Suppose that Ineq. (2) holds. Assume $\|\boldsymbol{x}_n\| = \Theta(\sqrt{d})$ for any $n \in [N]$ and $\|\boldsymbol{z}\| = \Theta(\sqrt{d})$. Suppose that $y_n^{\mathrm{adv}}$ is randomly sampled from $\{\pm 1\}$ for each $n$. Assume $|\sum_{n=1}^{N} \lambda_n y_n \langle \boldsymbol{x}_n, \boldsymbol{z} \rangle| = \Theta(g(N,d))$ if $\sum_{n=1}^{N} \lambda_n |\langle \boldsymbol{x}_n, \boldsymbol{z} \rangle| = \Theta(g(N,d))$, where $g$ is a positive function of $N$ and $d$. If there is no $n$ such that $|\langle \boldsymbol{x}_n, \boldsymbol{z} \rangle| = \Theta(d)$ or $\sum_{n:|\langle \boldsymbol{x}_n, \boldsymbol{z} \rangle| \neq \Theta(d)} |\langle \boldsymbol{x}_n, \boldsymbol{z} \rangle| = \mathcal{O}(d)$ does not hold, with $N, d \to \infty$, then $\mathrm{sgn}(f_{\mathrm{adv}}^{\mathrm{bdy}}(\boldsymbol{z})) = \mathrm{sgn}(f^{\mathrm{bdy}}(\boldsymbol{z}))$ holds with probability at least 99.99%.*

---

*Proof.* If $|\langle \boldsymbol{x}_n, \boldsymbol{z} \rangle| = 0$ for every $n$, then $f_{\mathrm{adv}}^{\mathrm{bdy}}(\boldsymbol{z}) = |T_1(\boldsymbol{z})| = |T_2(\boldsymbol{z})| = f^{\mathrm{bdy}}(\boldsymbol{z}) = 0$. Assume that there exists $n$ such that $|\langle \boldsymbol{x}_n, \boldsymbol{z} \rangle| > 0$. By Proposition C.8, if there is no $n$ such that $|\langle \boldsymbol{x}_n, \boldsymbol{z} \rangle| = \Theta(d)$ or $\sum_{n:|\langle \boldsymbol{x}_n, \boldsymbol{z} \rangle| \neq \Theta(d)} |\langle \boldsymbol{x}_n, \boldsymbol{z} \rangle| = \mathcal{O}(d)$, with $N, d \to \infty$, then $|T_2(z)| > |T_1(z)|$ with sufficiently high probability; thus, $\mathrm{sgn}(f_{\mathrm{adv}}^{\mathrm{bdy}}(\boldsymbol{z})) = \mathrm{sgn}(f^{\mathrm{bdy}}(\boldsymbol{z}))$. $\qquad\square$

## D  PROOFS OF THEOREMS IN SECTION 4.3

In this section, we prove Theorems 4.3 and 4.4 and Corollary 4.5. The proof flows of Theorems 4.3 and 4.4 follow Appendix C. In addition, we derive Corollary 4.5 as a natural consequence of Theorem 4.2.

First, we summarize the properties of uniform random variables, which are required to consider the orthogonality condition of perturbations on uniform noises.

---

**Lemma D.1** (Properties of uniform random vectors)**.** *Let $\{\boldsymbol{X}_n\}_{n=1}^{N} \subset [-1,1]^d$ be $N \in \mathbb{N}$ independent random variables sampled from the uniform distribution $U([-1,1]^d)$. Let $\boldsymbol{z} \in \mathbb{R}^d$ be a constant vector. Then, for a positive constant $t > 1/N$, the following inequalities hold:*

$$(a) \ \mathbb{P}\left[ \max_n \left| \|\boldsymbol{X}_n\|^2 - \frac{d}{3} \right| \leq \frac{\sqrt{d \ln tN}}{2} \right] \geq \left( 1 - \frac{2}{tN} \right)^N, \tag{A90}$$

$$(b) \ \mathbb{P}\left[ \max_n |\langle \boldsymbol{X}_n, \boldsymbol{X}_k \rangle| \leq \sqrt{2d \ln tN} \right] \geq \left( 1 - \frac{2}{tN} \right)^N, \tag{A91}$$

$$(c) \ \mathbb{P}\left[ \max_n |\langle \boldsymbol{X}_n, \boldsymbol{z} \rangle| \leq \sqrt{2 \ln tN} \|\boldsymbol{z}\| \right] \geq \left( 1 - \frac{2}{tN} \right)^N. \tag{A92}$$

---

*Proof.* Let $a > 0$ be a positive constant.

**(a)** By Hoeffding's inequality with $\mathbb{E}[X_{n,i}^2] = 1/3$,

$$\mathbb{P}\left[ \left| \|X_n\|^2 - \frac{d}{3} \right| \geq a \right] \leq 2 \exp\left( -\frac{2a^2}{d} \right). \tag{A93}$$

Thus,

$$\mathbb{P}\left[ \max_n \left| \|X_n\|^2 - \frac{d}{3} \right| \leq a \right] = \left( \mathbb{P}\left[ \left| \|X_n\|^2 - \frac{d}{3} \right| \leq a \right] \right)^N \tag{A94}$$

$$= \left(1 - \mathbb{P}\left[\left|\|X_n\|^2 - \frac{d}{3}\right| \geq a\right]\right)^N \tag{A95}$$

$$\geq \left(1 - 2\exp\left(-\frac{2a^2}{d}\right)\right)^N. \tag{A96}$$

For $a = \sqrt{d\ln(tN)/2}$ with $t > 1/N$,

$$\mathbb{P}\left[\max_n \left|\|X_n\|^2 - \frac{d}{3}\right| \leq \sqrt{d\ln tN}\right] \geq \left(1 - \frac{2}{tN}\right)^N. \tag{A97}$$

**(b)** Similarly to (i),

$$\mathbb{P}[|\langle \boldsymbol{X}_n, \boldsymbol{X}_k\rangle| \geq a] \leq 2\exp\left(-\frac{a^2}{2d}\right), \tag{A98}$$

$$\mathbb{P}\left[\max_n |\langle \boldsymbol{X}_n, \boldsymbol{X}_k\rangle| \leq a\right] \geq \left(1 - 2\exp\left(-\frac{a^2}{2d}\right)\right)^N, \tag{A99}$$

$$\mathbb{P}\left[\max_n |\langle \boldsymbol{X}_n, \boldsymbol{X}_k\rangle| \leq \sqrt{2d\ln tN}\right] \geq \left(1 - \frac{2}{tN}\right)^N. \tag{A100}$$

**(c)** Similarly to (i),

$$\mathbb{P}[|\langle \boldsymbol{X}_n, \boldsymbol{z}\rangle| \geq a] \leq 2\exp\left(-\frac{a^2}{2\|\boldsymbol{z}\|^2}\right), \tag{A101}$$

$$\mathbb{P}\left[\max_n |\langle \boldsymbol{X}_n, \boldsymbol{z}\rangle| \leq a\right] \geq \left(1 - 2\exp\left(-\frac{a^2}{2\|\boldsymbol{z}\|^2}\right)\right)^N, \tag{A102}$$

$$\mathbb{P}\left[\max_n |\langle \boldsymbol{X}_n, \boldsymbol{z}\rangle| \leq \sqrt{2\ln tN}\|\boldsymbol{z}\|\right] \geq \left(1 - \frac{2}{tN}\right)^N. \tag{A103}$$

$\square$

**Theorem 4.3** (Decision boundary when learning from geometry-inspired perturbations on uniform noises)**.** *Assume $\gamma^3 R_{\min}^4/(3NR_{\max}^2) \geq p_{\max}$. Let $f$ be a one-hidden-layer neural network trained on geometry-inspired perturbations on natural data (cf. Eq. (1) and Definition 3.2(b)) with Setting 3.1. For any $n \neq k$, if*

$$\frac{d}{3} - \frac{\sqrt{Cd}}{2} \leq \|\boldsymbol{X}_n\|^2 \leq \frac{d}{3} + \frac{\sqrt{Cd}}{2}, \quad |\langle \boldsymbol{X}_n, \boldsymbol{X}_k\rangle| \leq \sqrt{2Cd}, \quad |\langle \boldsymbol{X}_n, \boldsymbol{\eta}/\epsilon\rangle| \leq \sqrt{2C}, \tag{4}$$

$$\frac{\gamma^3(2d - 3\sqrt{Cd} - 12\sqrt{2C}\epsilon + 6\epsilon^2)^2}{18N^{\mathrm{adv}}(2d + 3\sqrt{Cd} + 12\sqrt{2C}\epsilon + 6\epsilon^2)} \geq \sqrt{2Cd} + 2\sqrt{2C}\epsilon + \epsilon^2 \tag{5}$$

*with $C := \ln 1000N^{\mathrm{adv}}$, then, with $t \to \infty$, the decision boundary of $f$ is given by:*

$$f_{\mathrm{adv}}^{\mathrm{bdy}}(\boldsymbol{z}) := \underbrace{\frac{\sum_{n=1}^{N^{\mathrm{adv}}} \lambda_n^{\mathrm{adv}} y_n^{\mathrm{adv}}\langle \boldsymbol{X}_n, \boldsymbol{z}\rangle}{\sum_{n=1}^{N^{\mathrm{adv}}} \lambda_n^{\mathrm{adv}}}}_{\text{Effect of learning from uniform noises}} + \underbrace{\epsilon\frac{f^{\mathrm{bdy}}(\boldsymbol{z})}{\|\sum_{n=1}^{N} \lambda_n y_n \boldsymbol{x}_n\|}}_{\text{Effect of learning from perturbations}}. \tag{6}$$

*Proof.* Let $\boldsymbol{q} := \sum_{n=1}^{N} \lambda_n y_n \boldsymbol{x}_n$. The norm and inner product are

$$\left\|\boldsymbol{x}_n^{\mathrm{adv}}\right\|^2 = \|\boldsymbol{X}_n\|^2 + 2\epsilon y_n^{\mathrm{adv}}\left\langle \boldsymbol{X}_n, \frac{\boldsymbol{q}}{\|\boldsymbol{q}\|}\right\rangle + \epsilon^2, \tag{A104}$$

$$\langle \boldsymbol{x}_n^{\mathrm{adv}}, \boldsymbol{x}_k^{\mathrm{adv}}\rangle = \langle \boldsymbol{X}_n, \boldsymbol{X}_k\rangle + \epsilon y_n^{\mathrm{adv}}\left\langle \boldsymbol{X}_k, \frac{\boldsymbol{q}}{\|\boldsymbol{q}\|}\right\rangle + \epsilon y_k^{\mathrm{adv}}\left\langle \boldsymbol{X}_n, \frac{\boldsymbol{q}}{\|\boldsymbol{q}\|}\right\rangle + \epsilon^2 y_n^{\mathrm{adv}} y_k^{\mathrm{adv}}. \tag{A105}$$

Thus, under Ineq. (4),

$$R_{\min}^{\mathrm{adv}\,2} \geq \frac{d}{3} - \frac{\sqrt{Cd}}{2} - 2\sqrt{2C}\epsilon + \epsilon^2, \qquad R_{\max}^{\mathrm{adv}\,2} \leq \frac{d}{3} + \frac{\sqrt{Cd}}{2} + 2\sqrt{2C}\epsilon + \epsilon^2, \qquad \text{(A106)}$$

$$p_{\max}^{\mathrm{adv}} \leq \sqrt{2Cd} + 2\sqrt{2C}\epsilon + \epsilon^2. \qquad \text{(A107)}$$

Using these bounds, we can rearrange $\gamma^3 R_{\min}^{\mathrm{adv}\,4}/3N^{\mathrm{adv}} R_{\max}^{\mathrm{adv}\,2}$ as follows:

$$\frac{\gamma^3 R_{\min}^{\mathrm{adv}\,4}}{3N^{\mathrm{adv}} R_{\max}^{\mathrm{adv}\,2}} \geq \frac{\gamma^3 (2d - 3\sqrt{Cd} - 12\sqrt{2C}\epsilon + 6\epsilon^2)^2}{18N^{\mathrm{adv}}(2d + 3\sqrt{Cd} + 12\sqrt{2C}\epsilon + 6\epsilon^2)}. \qquad \text{(A108)}$$

Thus, if Ineq. (5) holds, then $\gamma^3 R_{\min}^{\mathrm{adv}\,4}/3N^{\mathrm{adv}} R_{\max}^{\mathrm{adv}\,2} \geq p_{\max}^{\mathrm{adv}}$ holds, and we can use Corollary 3.4. The decision boundary can be derived similarly to Theorem C.2. $\qquad\square$

Lemma D.2 is used in the proof of Theorem 4.4.

**Lemma D.2** (Upper bound of sum of squared inner products). *If $\|\boldsymbol{x}_n\| = \Theta(\sqrt{d})$, $\|\boldsymbol{z}\| = \Theta(\sqrt{d})$, and $|\langle \boldsymbol{x}_n, \boldsymbol{x}_k \rangle| = \mathcal{O}(d/N)$ for any $n, k \in [N], n \neq k$, then $\sum_{n=1}^{N} \langle \boldsymbol{x}_n, \boldsymbol{z} \rangle^2 = \mathcal{O}(d^2)$.*

*Proof.* With $\psi_n := \mathrm{sgn}(\langle \boldsymbol{x}_n, \boldsymbol{z} \rangle)$, by the Cauchy–Schwarz inequality,

$$\sum_{n=1}^{N} |\langle \boldsymbol{x}_n, \boldsymbol{z} \rangle| = \sum_{n=1}^{N} \langle \psi_n \boldsymbol{x}_n, \boldsymbol{z} \rangle \qquad \text{(A109)}$$

$$\leq \left\| \sum_{n=1}^{N} \psi_n \boldsymbol{x}_n \right\| \|\boldsymbol{z}\| \qquad \text{(A110)}$$

$$= \sqrt{\sum_{n=1}^{N} \|\boldsymbol{x}_n\|^2 + \sum_{n=1}^{N}\sum_{k\neq n} \psi_n \psi_k \langle \boldsymbol{x}_n, \boldsymbol{x}_k \rangle} \|\boldsymbol{z}\| \qquad \text{(A111)}$$

$$= \mathcal{O}(\sqrt{N}d). \qquad \text{(A112)}$$

By the Cauchy–Schwarz inequality,

$$\sum_{n=1}^{N} \langle \boldsymbol{x}_n, \boldsymbol{z} \rangle^2 = \sum_{n=1}^{N} \langle \langle \boldsymbol{x}_n, \boldsymbol{z} \rangle \boldsymbol{x}_n, \boldsymbol{z} \rangle \qquad \text{(A113)}$$

$$= \left\langle \sum_{n=1}^{N} \langle \boldsymbol{x}_n, \boldsymbol{z} \rangle \boldsymbol{x}_n, \boldsymbol{z} \right\rangle \qquad \text{(A114)}$$

$$\leq \sqrt{\sum_{n=1}^{N} \langle \boldsymbol{x}_n, \boldsymbol{z} \rangle^2 \|\boldsymbol{x}_n\|^2 + \sum_{n=1}^{N}\sum_{k\neq n} \langle \boldsymbol{x}_n, \boldsymbol{z} \rangle \langle \boldsymbol{x}_k, \boldsymbol{z} \rangle \langle \boldsymbol{x}_n, \boldsymbol{x}_k \rangle} \|\boldsymbol{z}\|. \qquad \text{(A115)}$$

Now,

$$\sum_{n=1}^{N}\sum_{k\neq n} \langle \boldsymbol{x}_n, \boldsymbol{z} \rangle \langle \boldsymbol{x}_k, \boldsymbol{z} \rangle \langle \boldsymbol{x}_n, \boldsymbol{x}_k \rangle \leq \sum_{n=1}^{N}\sum_{k=1}^{N} |\langle \boldsymbol{x}_n, \boldsymbol{z} \rangle| |\langle \boldsymbol{x}_k, \boldsymbol{z} \rangle| |\langle \boldsymbol{x}_n, \boldsymbol{x}_k \rangle| \qquad \text{(A116)}$$

$$= \mathcal{O}\left(\frac{d}{N}\right) \left( \sum_{n=1}^{N} |\langle \boldsymbol{x}_n, \boldsymbol{z} \rangle| \right)^2 \qquad \text{(A117)}$$

$$= \mathcal{O}(d^3). \qquad \text{(A118)}$$

Thus,

$$\sum_{n=1}^{N} \langle \boldsymbol{x}_n, \boldsymbol{z} \rangle^2 = \sqrt{\Theta(d^2) \sum_{n=1}^{N} \langle \boldsymbol{x}_n, \boldsymbol{z} \rangle^2 + \mathcal{O}(d^4)}. \qquad \text{(A119)}$$

Let $\sum_{n=1}^{N}\langle \boldsymbol{x}_n, \boldsymbol{z}\rangle^2 = \mathcal{O}(N^\zeta d^2)$ for a constant $\zeta \in \mathbb{R}^d$. Using this,

$$\underbrace{\sum_{n=1}^{N}\langle \boldsymbol{x}_n, \boldsymbol{z}\rangle^2}_{\mathcal{O}(N^\zeta d^2)} = \mathcal{O}(\max(N^{\zeta/2}d^2, d^2)). \tag{A120}$$

If $\zeta > 0$, the left term grows faster than the right term, which contradicts the equation. Thus, $\zeta \leq 0$. □

---

**Theorem 4.4** (Consistent decision of learning from geometry-inspired perturbations on uniform noises). *Assume* $\gamma^3 R_{\min}^4/(3NR_{\max}^2) \geq p_{\max}$. *Suppose that Ineqs. (4) and (5) hold. Assume* $\|\boldsymbol{z}\| = \Theta(\sqrt{d})$, $|f^{\mathrm{bdy}}(\boldsymbol{z})| = \Omega(1)$, *and* $d > N^{\mathrm{adv}^2}$. *Then, the following equations hold with probability at least 99.99%:*

$$\left| \frac{\sum_{n=1}^{N^{\mathrm{adv}}} \lambda_n^{\mathrm{adv}} y_n^{\mathrm{adv}} \langle \boldsymbol{X}_n, \boldsymbol{z}\rangle}{\sum_{n=1}^{N^{\mathrm{adv}}} \lambda_n^{\mathrm{adv}}} \right| = \mathcal{O}\left(\frac{\sqrt{d}}{N^{\mathrm{adv}}}\right), \qquad \epsilon \frac{|f^{\mathrm{bdy}}(\boldsymbol{z})|}{\|\sum_{n=1}^{N} \lambda_n y_n \boldsymbol{x}_n\|} = \tilde{\Omega}\left(\frac{d}{\sqrt{N^{\mathrm{adv}} N}}\right). \tag{7}$$

*In addition, if $d$ and $N^{\mathrm{adv}}$ are sufficiently large and $N^{\mathrm{adv}} \geq N$ holds, then for any $\boldsymbol{z} \in \mathbb{R}^d$, $\mathrm{sgn}(f_{\mathrm{adv}}^{\mathrm{bdy}}(\boldsymbol{z})) = \mathrm{sgn}(f^{\mathrm{bdy}}(\boldsymbol{z}))$ holds with probability at least 99.99%.*

---

*Proof.* By the definition in Theorem 3.3 and Corollary 3.4, $\lambda_n = \Theta(1/d)$ and $\lambda_n^{\mathrm{adv}} = \Theta(1/d)$.

**Left term.** Consider the limiting behavior of $|\sum_{n=1}^{N^{\mathrm{adv}}} \lambda_n^{\mathrm{adv}} y_n^{\mathrm{adv}} \langle \boldsymbol{X}_n, \boldsymbol{z}\rangle|$. First, we provide an *incorrect* idea for clarity. By Hoeffding's inequality with respect to $\{y_n^{\mathrm{adv}}\}_{n=1}^{N^{\mathrm{adv}}}$ and $\{\boldsymbol{X}_n\}_{n=1}^{N^{\mathrm{adv}}}$, for $t > 0$,

$$\mathbb{P}\left[\left|\sum_{n=1}^{N^{\mathrm{adv}}} \lambda_n^{\mathrm{adv}} y_n^{\mathrm{adv}} \langle \boldsymbol{X}_n, \boldsymbol{z}\rangle\right| \geq t\right] \leq 2\exp\left(-\frac{t^2}{2\sum_{n=1}^{N^{\mathrm{adv}}} \sum_{i=1}^{d} \lambda_n^{\mathrm{adv}^2} z_i^2}\right) \tag{A121}$$

$$= 2\exp\left(-\frac{t^2}{2\sum_{n=1}^{N^{\mathrm{adv}}} \lambda_n^{\mathrm{adv}^2} \|\boldsymbol{z}\|^2}\right) \tag{A122}$$

$$= \mathcal{O}\left(\exp\left(-\frac{dt^2}{N^{\mathrm{adv}}}\right)\right). \tag{A123}$$

From this inequality, one might consider that $|\sum_{n=1}^{N^{\mathrm{adv}}} \lambda_n^{\mathrm{adv}} y_n^{\mathrm{adv}} \langle \boldsymbol{X}_n, \boldsymbol{z}\rangle| = \mathcal{O}(\sqrt{N^{\mathrm{adv}}/d})$ holds with sufficiently high probability. This is incorrect because the above rearrangement does not take into account the orthogonality assumption on $\{\boldsymbol{X}_n\}_{n=1}^{N^{\mathrm{adv}}}$, i.e., $|\langle \boldsymbol{X}_n, \boldsymbol{X}_k\rangle| = \mathcal{O}(d/N)$ for $n \neq k$. We then provide a correct estimation. By Hoeffding's inequality with respect to $\{y_n^{\mathrm{adv}}\}_{n=1}^{N^{\mathrm{adv}}}$,

$$\mathbb{P}\left[\left|\sum_{n=1}^{N^{\mathrm{adv}}} \lambda_n^{\mathrm{adv}} y_n^{\mathrm{adv}} \langle \boldsymbol{X}_n, \boldsymbol{z}\rangle\right| \geq t\right] \leq 2\exp\left(-\frac{t^2}{2\sum_{n=1}^{N^{\mathrm{adv}}} \lambda_n^{\mathrm{adv}^2} \langle \boldsymbol{X}_n, \boldsymbol{z}\rangle^2}\right). \tag{A124}$$

By Lemma D.2 with the assumptions, $\sum_{n=1}^{N^{\mathrm{adv}}} \lambda_n^{\mathrm{adv}^2} \langle \boldsymbol{X}_n, \boldsymbol{z}\rangle^2 = \mathcal{O}(1/d)$. Thus, we obtain $|\sum_{n=1}^{N^{\mathrm{adv}}} \lambda_n^{\mathrm{adv}} y_n^{\mathrm{adv}} \langle \boldsymbol{X}_n, \boldsymbol{z}\rangle| = \mathcal{O}(1/\sqrt{d})$ with sufficiently high probability. Because the growth rate of $|\sum_{n=1}^{N^{\mathrm{adv}}} \lambda_n^{\mathrm{adv}}|$ is $\Theta(N^{\mathrm{adv}}/d)$, the claim is established.

**Right term.** Under Ineq. (5) and $d > N^{\mathrm{adv}^2}$, $\epsilon = \tilde{\mathcal{O}}(d/N^{\mathrm{adv}})$. Since we can set $\epsilon$ freely under $\epsilon = \tilde{\mathcal{O}}(d/N^{\mathrm{adv}})$, we consider $\epsilon = \tilde{\Theta}(d/N^{\mathrm{adv}})$. By Lemma C.3 and the assumption $|f^{\mathrm{bdy}}(\boldsymbol{z})| = \Omega(1)$, the claim is established.

**Consistent decision.** This is trivial by the growth rate of two terms. □

**Corollary 4.5** (Complete classification for natural training samples when learning from geome-try-inspired perturbations on uniform noises). *Assume $\gamma^3 R_{\min}^4/(3N R_{\max}^2) \geq p_{\max}$. Suppose that Ineqs. (4) and (5) hold. If $d$ and $N^{\mathrm{adv}}$ are sufficiently large and $d > N^{\mathrm{adv}^2} \geq \sqrt{N}$ holds, then a one-hidden-layer neural network trained on geometry-inspired perturbations on uniform noises with Setting 3.1 can completely classify the natural dataset $\{(\boldsymbol{x}_n, y_n)\}_{n=1}^N$ with probability at least 99.99%.*

*Proof.* We can rearrange $f^{\mathrm{bdy}}(\boldsymbol{x}_n)$ as follows:

$$f^{\mathrm{bdy}}(\boldsymbol{x}_n) = \sum_{k=1}^N \lambda_k y_k \langle \boldsymbol{x}_k, \boldsymbol{x}_n \rangle = \lambda_n y_n \|\boldsymbol{x}_n\|^2 - \sum_{l \neq n} \lambda_l y_l \langle \boldsymbol{x}_l, \boldsymbol{x}_n \rangle. \tag{A125}$$

By Ineq. (A65), $|f^{\mathrm{bdy}}(\boldsymbol{x}_n)| = \Theta(1)$. By Theorem 4.4, the claim is established. $\square$

# E    OTHER PERTURBATIONS

In the main text, we consider learning from geometry-inspired $L_2$ perturbations, which simplify notation. In this section, we consider learning from geometry-inspired $L_0$ and $L_\infty$ and gradient-based $L_2$ perturbations.

## E.1    GEOMETRY-INSPIRED $L_0$ PERTURBATIONS

Let $d_\delta \in \mathbb{N}_{\leq d}$ be the number of modified pixels. An adversarial example is restricted to $\|\boldsymbol{x}_n^{\mathrm{adv}} - \boldsymbol{x}_n\|_0 \leq d_\delta$. Following one pixel attack (Su et al., 2019) and SparseFool (Modas et al., 2019), we do not constrain the distance between original and perturbed pixels. We define a geometry-inspired $L_0$ perturbation as follows:

$$\boldsymbol{\eta}_n := \epsilon y_n^{\mathrm{adv}} \frac{\boldsymbol{\nabla}_{\boldsymbol{x}_n} f^{\mathrm{bdy}}(\boldsymbol{x}_n) \odot \boldsymbol{M}_n}{\|\boldsymbol{\nabla}_{\boldsymbol{x}_n} f^{\mathrm{bdy}}(\boldsymbol{x}_n) \odot \boldsymbol{M}_n\|} = \epsilon y_n^{\mathrm{adv}} \frac{\sum_{k=1}^N \lambda_k y_k \boldsymbol{x}_k \odot \boldsymbol{M}_k}{\|\sum_{k=1}^N \lambda_k y_k \boldsymbol{x}_k \odot \boldsymbol{M}_k\|}, \tag{A126}$$

where $\odot$ denotes the Hadamard product and $\boldsymbol{M}_n \in \{0,1\}^d$ denotes a mask vector. Let $S_n$ be the set of the top-$d_\delta$ elements where $|\nabla_{x_{n,i}} f^{\mathrm{bdy}}(\boldsymbol{x}_n)|$ is the largest. The $i$-th element of $\boldsymbol{M}_n$ is set to one if $i$ is contained in $S_n$ and zero otherwise. Since $\boldsymbol{\nabla}_{\boldsymbol{x}_n} f^{\mathrm{bdy}}(\boldsymbol{x}_n)$ does not depend on $n$, $\boldsymbol{M}_n$ does not depend on $n$. Thus, we denote $\boldsymbol{M} := \boldsymbol{M}_1 = \cdots = \boldsymbol{M}_N$. Rearranging the above equation,

$$\boldsymbol{\eta}_n = \epsilon y_n^{\mathrm{adv}} \frac{\sum_{k=1}^N \lambda_k y_k \boldsymbol{x}_k \odot \boldsymbol{M}}{\|\sum_{k=1}^N \lambda_k y_k \boldsymbol{x}_k \odot \boldsymbol{M}\|}. \tag{A127}$$

Similar to Eq. (1), this perturbation is represented as the weighted sum of benign training samples, and thus contains class features. This result indicates that even sparse perturbations, which seem to lack natural data structures, enable networks to generalize.

Consider the decision boundary when learning from $L_0$ perturbations. To employ Corollary 3.4, we first construct an orthogonality condition for $L_0$ perturbations. As a preliminary, refer to the proof of Lemma C.1. The norm is

$$\|\boldsymbol{x}_n^{\mathrm{adv}}\|^2 = \|\boldsymbol{x}_n\|^2 + 2\epsilon y_n^{\mathrm{adv}} \frac{\sum_{k=1}^N \lambda_k y_k \langle \boldsymbol{x}_k \odot \boldsymbol{M}, \boldsymbol{x}_n \rangle}{\|\sum_{k=1}^N \lambda_k y_k \boldsymbol{x}_k \odot \boldsymbol{M}\|} + \epsilon^2 \tag{A128}$$

$$= \Theta(d) \pm \Theta\left(\sqrt{\frac{d_\delta}{N}}\right)\epsilon + \epsilon^2. \tag{A129}$$

Note that

$$\left\|\sum_{n=1}^N \lambda_n y_n \boldsymbol{x}_n \odot \boldsymbol{M}\right\|^2 = \sum_{n=1}^N \lambda_n^2 \|\boldsymbol{x}_n \odot \boldsymbol{M}\|^2 + \sum_{n=1}^N \sum_{k \neq n} \lambda_n \lambda_k y_n y_k \langle \boldsymbol{x}_n \odot \boldsymbol{M}, \boldsymbol{x}_k \odot \boldsymbol{M} \rangle \tag{A130}$$

$$=\Theta\left(\frac{Nd_\delta}{d^2}\right). \tag{A131}$$

In addition,

$$\sum_{k=1}^{N}\lambda_k y_k\langle\boldsymbol{x}_k\odot\boldsymbol{M},\boldsymbol{x}_n\rangle=\lambda_n y_n\langle\boldsymbol{x}_n\odot\boldsymbol{M},\boldsymbol{x}_n\rangle+\sum_{k\neq n}\lambda_k y_k\langle\boldsymbol{x}_k\odot\boldsymbol{M},\boldsymbol{x}_n\rangle=\Theta\left(\frac{d_\delta}{d}\right). \tag{A132}$$

The inner product is

$$\langle\boldsymbol{x}_n^{\mathrm{adv}},\boldsymbol{x}_k^{\mathrm{adv}}\rangle$$

$$=\langle\boldsymbol{x}_n,\boldsymbol{x}_k\rangle+\epsilon y_n^{\mathrm{adv}}\frac{\sum_{l=1}^{N}\lambda_l y_l\langle\boldsymbol{x}_l\odot\boldsymbol{M},\boldsymbol{x}_k\rangle}{\|\sum_{l=1}^{N}\lambda_l y_l\boldsymbol{x}_l\odot\boldsymbol{M}\|}+\epsilon y_k^{\mathrm{adv}}\frac{\sum_{l=1}^{N}\lambda_l y_l\langle\boldsymbol{x}_l\odot\boldsymbol{M},\boldsymbol{x}_n\rangle}{\|\sum_{l=1}^{N}\lambda_l y_l\boldsymbol{x}_l\odot\boldsymbol{M}\|}+\epsilon^2 \tag{A133}$$

$$=\mathcal{O}\left(\frac{d}{N}\right)\pm\Theta\left(\sqrt{\frac{d_\delta}{N}}\right)\epsilon+\epsilon^2. \tag{A134}$$

The orthogonality condition can be rearranged as follows:

$$\frac{(\Theta(d)-\Theta(\sqrt{d_\delta/N})\epsilon+\epsilon^2)^2}{\Theta(N)(\Theta(d)+\Theta(\sqrt{d_\delta/N})\epsilon+\epsilon^2)}-\mathcal{O}\left(\frac{d}{N}\right)-\Theta\left(\sqrt{\frac{d_\delta}{N}}\right)\epsilon-\epsilon^2\geq 0. \tag{A135}$$

Thus, $\epsilon$ is constrained to $\mathcal{O}(\sqrt{d/N})$. Moreover, similarly to Theorem 4.2, the following decision boundary can be derived using Corollary 3.4:

$$f_{\mathrm{adv}}^{\mathrm{bdy}}(\boldsymbol{z})=\frac{\sum_{n=1}^{N}\lambda_n^{\mathrm{adv}}y_n^{\mathrm{adv}}\langle\boldsymbol{x}_n,\boldsymbol{z}\rangle}{\sum_{n=1}^{N}\lambda_n^{\mathrm{adv}}}+\epsilon\frac{\sum_{n=1}^{N}\lambda_n y_n\langle\boldsymbol{x}_n\odot\boldsymbol{M},\boldsymbol{z}\rangle}{\|\sum_{n=1}^{N}\lambda_n y_n\boldsymbol{x}_n\odot\boldsymbol{M}\|}. \tag{A136}$$

Finally, under the condition of Theorem 4.2, we consider the limiting behavior of this boundary for $\boldsymbol{z}:=\sum_{n=1}^{N}\Theta(1/\sqrt{N})\boldsymbol{x}_n$. Since the left term of this boundary equals that of Eq. (3), the left term grows with $\mathcal{O}(d/N)$. Similarly to Lemma C.3, the right term grows with $\Theta(\sqrt{dd_\delta/N})$. If $d/d_\delta=\Theta(1)$, the right term grows faster than the left; namely, the effect of learning from perturbations dominates the classifier decision. Thus, learning from $L_0$ perturbations succeeds for samples that are weakly correlated with many training samples.

### E.2 GEOMETRY-INSPIRED $L_\infty$ PERTURBATIONS

Similar to fast gradient sign method (Goodfellow et al., 2015) and projected gradient descent (Madry et al., 2018), we define an $L_\infty$ adversarial perturbation as follows:

$$\boldsymbol{\eta}_n:=\epsilon y_n^{\mathrm{adv}}\operatorname{sgn}(\boldsymbol{\nabla}_{\boldsymbol{x}_n}f^{\mathrm{bdy}}(\boldsymbol{x}_n))=\epsilon y_n^{\mathrm{adv}}\operatorname{sgn}\left(\sum_{k=1}^{N}\lambda_k y_k\boldsymbol{x}_k\right). \tag{A137}$$

Assume $\langle\operatorname{sgn}(\sum_{k=1}^{N}\lambda_k y_k\boldsymbol{x}_k),\boldsymbol{x}_n\rangle=\Theta(\sqrt{d})$ for any $n$. Similarly to the above discussion, the orders of the norm and inner product are

$$\left\|\boldsymbol{x}_n^{\mathrm{adv}}\right\|^2=\Theta(d)\pm\Theta(\sqrt{d})\epsilon+\Theta(d)\epsilon^2,\quad\langle\boldsymbol{x}_n^{\mathrm{adv}},\boldsymbol{x}_k^{\mathrm{adv}}\rangle=\mathcal{O}\left(\frac{d}{N}\right)\pm\Theta(\sqrt{d})\epsilon+\Theta(d)\epsilon^2. \tag{A138}$$

Assuming $d/N=\Theta(1)$, the orthogonality condition requires $\epsilon=\mathcal{O}(1/\sqrt{d})$. The decision boundary is

$$f_{\mathrm{adv}}^{\mathrm{bdy}}(\boldsymbol{z}):=\frac{\sum_{n=1}^{N}\lambda_n^{\mathrm{adv}}y_n^{\mathrm{adv}}\langle\boldsymbol{x}_n,\boldsymbol{z}\rangle}{\sum_{n=1}^{N}\lambda_n^{\mathrm{adv}}}+\epsilon\left\langle\operatorname{sgn}\left(\sum_{k=1}^{N}\lambda_k y_k\boldsymbol{x}_k\right),\boldsymbol{z}\right\rangle. \tag{A139}$$

Consider the limiting behavior of this boundary for $\boldsymbol{z}:=\Theta(d/\sqrt{N})\sum_{k=1}^{N}\lambda_k y_k\boldsymbol{x}_k$. Similarly to the discussion above, the left and right terms grow with $\mathcal{O}(d/N)(=\mathcal{O}(1))$ and $\Theta(\sqrt{d})$, respectively. Note that $\langle\operatorname{sgn}(\sum_{k=1}^{N}\lambda_k y_k\boldsymbol{x}_k),\sum_{k=1}^{N}\lambda_k y_k\boldsymbol{x}_k\rangle=\Theta(\sqrt{N})$. Thus, learning from $L_\infty$ perturbations succeeds for samples that are weakly correlated with many training samples.

### E.3 Gradient-based $L_2$ Perturbations

In the main text, we consider geometry-inspired perturbations that target the decision boundary $f^{\text{bdy}}$ instead of the network $f$ itself. In this section, we consider $L_2$ gradient-based perturbations that target the network $f$ itself. Similar concepts are employed in fast gradient sign method (Goodfellow et al., 2015) and projected gradient descent (Madry et al., 2018).

**Attack on Network Directly.** First, we consider an attack on the network $f$. Since gradient flow on $f$ converges in direction to $\boldsymbol{W}^{\text{std}}$ (cf. Theorem B.1), with $t \to \infty$, we can represent the network as $f(\boldsymbol{z}; c\boldsymbol{W}^{\text{std}})$ with $c > 0$. Note that $\langle \boldsymbol{v}, \boldsymbol{x}_n \rangle > 0$ and $\langle \boldsymbol{u}, \boldsymbol{x}_n \rangle < 0$ if $y_n = 1$, and $\langle \boldsymbol{u}, \boldsymbol{x}_n \rangle > 0$ and $\langle \boldsymbol{v}, \boldsymbol{x}_n \rangle < 0$ if $y_n = -1$ (Frei et al., 2023). By Theorem B.1, if $y_n = 1$,

$$f(\boldsymbol{x}_n; c\boldsymbol{W}^{\text{std}}) = \sum_{k=1}^{m/2} \frac{1}{\sqrt{m}} \langle c\boldsymbol{v}_k, \boldsymbol{x}_n \rangle - \sum_{k=1}^{m/2} \frac{\gamma}{\sqrt{m}} \langle c\boldsymbol{u}_k, \boldsymbol{x}_n \rangle \tag{A140}$$

$$= \frac{\sqrt{m}}{2} \langle c\boldsymbol{v}, \boldsymbol{x}_n \rangle - \frac{\gamma\sqrt{m}}{2} \langle c\boldsymbol{u}, \boldsymbol{x}_n \rangle \tag{A141}$$

$$= \frac{c(1+\gamma^2)}{2} \sum_{k:y_k=+1} \lambda_k y_k \langle \boldsymbol{x}_k, \boldsymbol{x}_n \rangle - \frac{c\gamma}{2} \sum_{k:y_k=-1} \lambda_k y_k \langle \boldsymbol{x}_k, \boldsymbol{x}_n \rangle. \tag{A142}$$

Thus,

$$\frac{\boldsymbol{\nabla}_{\boldsymbol{x}_n} f(\boldsymbol{x}_n; c\boldsymbol{W}^{\text{std}})}{\|\boldsymbol{\nabla}_{\boldsymbol{x}_n} f(\boldsymbol{x}_n; c\boldsymbol{W}^{\text{std}})\|} = \frac{(1+\gamma^2) \sum_{k:y_k=+1} \lambda_k y_k \boldsymbol{x}_k - \gamma \sum_{k:y_k=-1} \lambda_k y_k \boldsymbol{x}_k}{\|(1+\gamma^2) \sum_{k:y_k=+1} \lambda_k y_k \boldsymbol{x}_k - \gamma \sum_{k:y_k=-1} \lambda_k y_k \boldsymbol{x}_k\|}. \tag{A143}$$

We denote this by $\boldsymbol{s}_+$. Similarly, for negatively labeled samples, we define $\boldsymbol{s}_-$. An adversarial example targeting the network can be represented as follows:

$$\boldsymbol{x}_n^{\text{adv}} := \boldsymbol{x}_n + \begin{cases} \epsilon y_n^{\text{adv}} \boldsymbol{s}_+ & (y_n = 1) \\ \epsilon y_n^{\text{adv}} \boldsymbol{s}_- & (y_n = -1) \end{cases}. \tag{A144}$$

To consider the orthogonality condition, we evaluate the order of $\langle \boldsymbol{s}_+, \boldsymbol{x}_n \rangle$. Similarly to Lemma C.3, the denominator of $\boldsymbol{s}_+$ grows with $\Theta(\sqrt{N/d})$. In addition, the inner product between the numerator and $\boldsymbol{x}_n$ grows with $\Theta(1)$ (cf. Lemma C.1). Thus, similar to Lemma C.1, the orthogonality condition requires $\epsilon = \mathcal{O}(\sqrt{d/N})$. By Corollary 3.4, the decision boundary is

$$\begin{aligned} &f_{\text{adv}}^{\text{bdy}}(\boldsymbol{z}) \\ &= \frac{\sum_{n=1}^N \lambda_n^{\text{adv}} y_n^{\text{adv}} \langle \boldsymbol{x}_n, \boldsymbol{z} \rangle}{\sum_{n=1}^N \lambda_n^{\text{adv}}} + \epsilon \frac{\sum_{n:y_n=+1} \lambda_n^{\text{adv}} \langle \boldsymbol{s}_+, \boldsymbol{z} \rangle + \sum_{n:y_n=-1} \lambda_n^{\text{adv}} \langle \boldsymbol{s}_-, \boldsymbol{z} \rangle}{\sum_{n=1}^N \lambda_n^{\text{adv}}}. \end{aligned} \tag{A145}$$

Consider the limiting behavior of this boundary for $\boldsymbol{z} = \sum_{n=1}^N \Theta(1/\sqrt{N}) \boldsymbol{x}_n$. Because the left term of this boundary equals Eq. (3), it grows with $\mathcal{O}(d/N)$. Since $\lambda^{\text{adv}} = \Theta(1/d)$ (cf. Corollary 3.4) and $\langle \boldsymbol{s}_+, \boldsymbol{z} \rangle = \sqrt{Nd}$, the right term grows with $\mathcal{O}(d/\sqrt{N})$. Thus, learning from $L_2$ perturbations that target the network itself also succeeds for samples that are weakly correlated with many training samples.

**Attack on Network with Loss Function.** Then, we consider an attack on the network $f$ with the exponential loss. The gradients of the exponential loss for $\boldsymbol{x}_n$ can be calculated as follows:

$$\nabla_{\boldsymbol{x}_n} \exp\big(-y_n f(\boldsymbol{x}_n; c\boldsymbol{W}^{\text{std}})\big) = -y_n \exp\big(-y_n f(\boldsymbol{x}_n; c\boldsymbol{W}^{\text{std}})\big) \nabla_{\boldsymbol{x}_n} f(\boldsymbol{x}_n; c\boldsymbol{W}^{\text{std}}) \tag{A146}$$

$$= -y_n \exp(-c) \nabla_{\boldsymbol{x}_n} f(\boldsymbol{x}_n; c\boldsymbol{W}^{\text{std}}). \tag{A147}$$

This indicates that the normalized gradients with the exponential loss are consistent with those of the network without a loss function. Thus, the case of the direct attack on the network can be applied to the case of the exponential loss. The same discussion applies to the logistic loss.

## F    Sub-Gaussian Noise Scenario

In this section, we consider learning from perturbations on sub-Gaussian noises. A sub-Gaussian random variable is defined as follows:

**Definition F.1** (Sub-Gaussian). *A random variable $X$ is called sub-Gaussian if there exists some $\sigma^2 > 0$ such that for every $s \in \mathbb{R}$, $X$ satisfies $\mathbb{E}[e^{s(X-\mathbb{E}[X])}] \leq e^{\sigma^2 s^2/2}$.*

We write a sub-Gaussian random variable as $X \sim \mathrm{subG}(\sigma^2)$. Sub-Gaussian random variables include Gaussian and any bounded random variables such as Rademacher (or symmetric Bernoulli) and uniform random variables. In addition, we define a sub-exponential random variable as follows:

**Definition F.2** (Sub-exponential). *A random variable $X$ is called sub-exponential if there exist some $\sigma^2 > 0$ and $b > 0$ such that for every $|s| \leq 1/b$, $X$ satisfies $\mathbb{E}[e^{s(X-\mathbb{E}[X])}] \leq e^{\sigma^2 s^2/2}$.*

We write a sub-exponential random variable as $X \sim \mathrm{subE}(\sigma^2, b)$. Sub-Gaussian and -exponential have the following properties:

**Lemma F.3** (Properties of sub-Gaussian and -exponential).

    *(a) If $X$ follows $\mathrm{subG}(\sigma^2)$, then $\mathbb{P}[|X - \mathbb{E}[X]| > t] \leq 2\exp(-t^2/2\sigma^2)$ holds for any $t > 0$.*

    *(b) If $X$ follows $\mathrm{subE}(\sigma^2, b)$, then $\mathbb{P}[|X - \mathbb{E}[X]| > t] \leq 2\exp(-\min\{t^2/\sigma^2, t/b\}/2)$ holds for any $t > 0$.*

    *(c) If $X_1$ and $X_2$ are independent and follow $\mathrm{subG}(\sigma_1^2)$ and $\mathrm{subG}(\sigma_2^2)$, respectively, then $\alpha_1 X_1 + \alpha_2 X_2$ follows $\mathrm{subG}(\alpha_1^2\sigma_1^2 + \alpha_2^2\sigma_2^2)$ for any $\alpha_1, \alpha_2 \in \mathbb{R}$.*

    *(d) If $X_1$ and $X_2$ are independent and follow $\mathrm{subE}(\sigma_1^2, b_1)$ and $\mathrm{subE}(\sigma_2^2, b_2)$, respectively, then $\alpha_1 X_1 + \alpha_2 X_2$ follows $\mathrm{subE}(\alpha_1^2\sigma_1^2 + \alpha_2^2\sigma_2^2, \max\{|\alpha_1|b_1, |\alpha_2|b_2\})$ for any $\alpha_1, \alpha_2 \in \mathbb{R}$.*

    *(e) If $X$ follows $\mathrm{subG}(\sigma^2)$, then $X^2$ follows $\mathrm{subE}(256\sigma^4, 16\sigma^2)$.*

    *(f) Suppose that (i) $X_1$ and $X_2$ are independent. (ii) $X_1$ and $X_2$ follow $\mathrm{subG}(\sigma_1^2)$ and $\mathrm{subG}(\sigma_2^2)$, respectively. (iii) $\mathbb{E}[X_1] = \mathbb{E}[X_2] = 0$. Then, $X_1 X_2$ follows $\mathrm{subE}(8\sigma_1^2\sigma_2^2, \sqrt{2}\sigma_1\sigma_2)$.*

*Proof.* For (a)–(e), refer to Duchi (2023); Rigollet & Hütter (2023). We consider (f). For any $s \in \mathbb{R}$,

$$\mathbb{E}_{X_1, X_2}[e^{sX_1X_2}] = \mathbb{E}_{X_2}[\mathbb{E}_{X_1}[e^{sX_1x_2} \mid X_2 = x_2]] \tag{A148}$$

$$\leq \mathbb{E}_{X_2}[e^{\sigma_1^2 X_2^2 s^2/2}] \tag{A149}$$

$$= \mathbb{E}_{X_2}\left[\sum_{n=0}^{\infty} \frac{(\sigma_1^2 X_2^2 s^2)^n}{n! 2^n}\right]. \tag{A150}$$

By $\mathbb{E}[X_2^{2n}] \leq 2(2\sigma_2^2)^n n!$ for any $n \in \mathbb{N}$ (Rigollet & Hütter, 2023),

$$\mathbb{E}[e^{sX_1X_2}] \leq 1 + 2\sum_{n=1}^{\infty}(\sigma_1^2\sigma_2^2 s^2)^n. \tag{A151}$$

For any $|s| \leq 1/\sqrt{2}\sigma_1\sigma_2$,

$$\mathbb{E}[e^{sX_1X_2}] \leq 1 + 4\sigma_1^2\sigma_2^2 s^2 \leq e^{8\sigma_1^2\sigma_2^2 s^2/2}. \tag{A152}$$

Thus, $X_1 X_2$ follows $\mathrm{subE}(8\sigma_1^2\sigma_2^2, \sqrt{2}\sigma_1\sigma_2)$. $\qquad\square$

Similarly to Lemma D.1, we consider the properties of random vectors with sub-Gaussian entries.

**Lemma F.4** (Properties of sub-Gaussian random vectors). *Let $\{\boldsymbol{X}_n\}_{n=1}^N \subset \mathbb{R}^d$ be $N \in \mathbb{N}$ independent random variables sampled from the sub-Gaussian distribution $\mathrm{subG}(1)$. Assume $\mathbb{E}[X_{n,i}] = 0$ for any $n \in [N]$ and $i \in [d]$. Let $\boldsymbol{z} \in \mathbb{R}^d$ be a constant vector. Then, the following statements hold:*

    *(a) If $d \geq 2\ln 1000N$,*

$$\mathbb{P}\left[\max_n \left|\|\boldsymbol{X}_n\|^2 - d\right| \leq 16\sqrt{2d\ln 1000N}\right] \geq \left(1 - \frac{1}{500N}\right)^N. \tag{A153}$$

*(b) If $d \geq \ln(1000N)/4$,*

$$\mathbb{P}\Big[\max_n |\langle \boldsymbol{X}_n, \boldsymbol{X}_k\rangle| \leq 2\sqrt{2d\ln 1000N}\Big] \geq \Big(1 - \frac{1}{500N}\Big)^N. \tag{A154}$$

*(c)*

$$\mathbb{P}\Big[\max_n |\langle \boldsymbol{X}_n, \boldsymbol{z}\rangle| \leq \sqrt{2\ln 1000N}\|\boldsymbol{z}\|\Big] \geq \Big(1 - \frac{1}{500N}\Big)^N. \tag{A155}$$

*Proof.* As a preliminary, refer to Lemma F.3 and the proof of Lemma D.1.

**(a)** $\|\boldsymbol{X}_n\|^2$ follows $\text{subE}(256d, 16)$ with mean $d$. For $t > 0$,

$$\frac{t^2}{256d} \leq \frac{t}{16} \Leftrightarrow t \leq 16d. \tag{A156}$$

If $d \geq 2\ln 1000N$,

$$\mathbb{P}\Big[\big|\|X_n\|^2 - d\big| \geq 16\sqrt{2d\ln 1000N}\Big] \leq \frac{1}{500N}. \tag{A157}$$

Thus,

$$\mathbb{P}\Big[\max_n \big|\|X_n\|^2 - d\big| \leq 16\sqrt{2d\ln 1000N}\Big] \geq \Big(1 - \frac{1}{500N}\Big)^N. \tag{A158}$$

**(b)** $\langle \boldsymbol{X}_n, \boldsymbol{X}_k\rangle$ follows $\text{subE}(8d, \sqrt{2})$ with zero mean. For $t > 0$,

$$\frac{t^2}{8d} \leq \frac{t}{\sqrt{2}} \Leftrightarrow t \leq 4\sqrt{2}d. \tag{A159}$$

If $d \geq \ln(1000N)/4$,

$$\mathbb{P}\Big[|\langle \boldsymbol{X}_n, \boldsymbol{X}_k\rangle| \geq 2\sqrt{2d\ln 1000N}\Big] \leq \frac{1}{500N}. \tag{A160}$$

**(c)** $\langle \boldsymbol{X}_n, \boldsymbol{z}\rangle$ follows $\text{subG}(\|\boldsymbol{z}\|^2)$ with zero mean. For $t = \sqrt{2\ln 1000N}\|\boldsymbol{z}\|$,

$$\mathbb{P}\Big[|\langle \boldsymbol{X}_n, \boldsymbol{z}\rangle| \geq \sqrt{2\ln 1000N}\|\boldsymbol{z}\|\Big] \leq \frac{1}{500N}. \tag{A161}$$

$\square$

By Lemma F.4, for $\|\boldsymbol{z}\| = 1$, the following inequalities hold with probability at least 99.8%:

$$\|\boldsymbol{X}_n\|^2 = \tilde{\Theta}(d), \qquad \langle \boldsymbol{X}_n, \boldsymbol{X}_k\rangle = \tilde{\mathcal{O}}(\sqrt{d}), \qquad \langle \boldsymbol{X}_n, \boldsymbol{z}\rangle = \tilde{\mathcal{O}}(1). \tag{A162}$$

These orders are consistent with those for uniform noises (cf. Lemma D.1). Thus, similarly to Theorems 4.3 and 4.4, if $\epsilon$ is constrained to $\epsilon = \tilde{\mathcal{O}}(\sqrt{d/N^{\text{adv}}})$, the decision boundary Eq. (6) and consistent decisions can be obtained for learning from perturbations on sub-Gaussian noises.

# G   THEOREM 4.1 WITHOUT ASSUMPTION OF LAST LAYER OF NETWORK

In this section, we derive Theorem 4.1 without the assumption that the positive and negative values of $\boldsymbol{a}$ are equal. Let $m_+$ and $m_-$ be the numbers of positive and negative neurons in the hidden layer, respectively. In this setting, Frei et al. (2023) provides the decision boundary as follows:

$$f^{\text{bdy}}(\boldsymbol{z}) = (m_+ + \gamma m_-) \sum_{n:y_n=+1} \lambda_n \boldsymbol{x}_n - (\gamma m_+ + m_-) \sum_{n:y_n=-1} \lambda_n \boldsymbol{x}_n. \tag{A163}$$

With $A_n := (m_+ + \gamma m_-)$ if $y_n = +1$ and $A_n := (\gamma m_+ + m_-)$ otherwise,

$$f^{\mathrm{bdy}}(\boldsymbol{z}) = \sum_{n=1}^{N} A_n \lambda_n y_n \boldsymbol{x}_n. \tag{A164}$$

Similar to Eq. (1), a perturbation is defined as follows:

$$\boldsymbol{\eta}_n := \epsilon y_n^{\mathrm{adv}} \frac{\sum_{k=1}^{N} A_k \lambda_k y_k \boldsymbol{x}_k}{\left\| \sum_{k=1}^{N} A_k \lambda_k y_k \boldsymbol{x}_k \right\|}. \tag{A165}$$

Since $A_n = \Theta(1)$, the growth rate of $\boldsymbol{\eta}_n$ with respect to $d$ and $N$ is the same as the original definition of $\boldsymbol{\eta}_n$. Thus, similarly to Theorem C.2, if $\epsilon = \mathcal{O}(\sqrt{d/N})$, the decision boundary is

$$f_{\mathrm{adv}}^{\mathrm{bdy}}(\boldsymbol{z}) := \frac{\sum_{n=1}^{N} A_n^{\mathrm{adv}} \lambda_n^{\mathrm{adv}} y_n^{\mathrm{adv}} \langle \boldsymbol{x}_n, \boldsymbol{z} \rangle}{\sum_{n=1}^{N} A_n^{\mathrm{adv}} \lambda_n^{\mathrm{adv}}} + \epsilon \frac{\sum_{n=1}^{N} A_n \lambda_n y_n \langle \boldsymbol{x}_n, \boldsymbol{z} \rangle}{\| \sum_{n=1}^{N} A_n \lambda_n y_n \boldsymbol{x}_n \|}, \tag{A166}$$

where $A_n^{\mathrm{adv}} := (m_+ + \gamma m_-)$ if $y_n^{\mathrm{adv}} = +1$ and $A_n^{\mathrm{adv}} := (\gamma m_+ + m_-)$ otherwise.

## H  FLIPPED LABEL LEARNING

In this section, we explain the success of learning from perturbations under the flipped label scenario, i.e., $y_n^{\mathrm{adv}} = -y_n$ for every $n \in [N]$, with assumptions about a data structure and learning bias. First, we assume that $\boldsymbol{x}_n$ consists of two features: robust and non-robust features, which is based on the hypothesis from Ilyas et al. (2019). Second, we suppose that standard trained classifiers focus only on non-robust features, which is empirically supported by Etmann et al. (2019); Tsipras et al. (2019); Zhang & Zhu (2019); Chalasani et al. (2020). In other words, we assume that the decision boundary of the classifier consists of the learning effect only from non-robust features. Formally, these assumptions are summarized as follows:

**Assumption H.1.** (a) For every $n \in [N]$, a natural sample $\boldsymbol{x}_n$ can be represented as $\boldsymbol{x}_n := \boldsymbol{x}_n^{\mathrm{rob}} + \boldsymbol{x}_n^{\mathrm{non}}$. (b) For $n \neq k$, $\langle \boldsymbol{x}_n^{\mathrm{rob}}, \boldsymbol{x}_k^{\mathrm{rob}} \rangle = \mathcal{O}(d/N)$, $\langle \boldsymbol{x}_n^{\mathrm{rob}}, \boldsymbol{x}_k^{\mathrm{non}} \rangle = \mathcal{O}(d/N)$, and $\langle \boldsymbol{x}_n^{\mathrm{non}}, \boldsymbol{x}_k^{\mathrm{non}} \rangle = \mathcal{O}(d/N)$. (c) Under the setting of Theorem 3.3, the decision boundary is given by $f^{\mathrm{bdy}}(\boldsymbol{z}) := \sum_{n=1}^{N} \lambda_n y_n \langle \boldsymbol{x}_n^{\mathrm{non}}, \boldsymbol{z} \rangle$.

Consider perturbations to the decision boundary $f^{\mathrm{bdy}}(\boldsymbol{z}) := \sum_{n=1}^{N} \lambda_n y_n \langle \boldsymbol{x}_n^{\mathrm{non}}, \boldsymbol{z} \rangle$. Similarly to Eq. (1), an adversarial example and perturbation are defined as follows:

$$\boldsymbol{x}_n^{\mathrm{adv}} := \boldsymbol{x}_n + \boldsymbol{\eta}_n, \qquad \boldsymbol{\eta}_n := \epsilon y_n^{\mathrm{adv}} \frac{\sum_{k=1}^{N} \lambda_k y_k \boldsymbol{x}_k^{\mathrm{non}}}{\| \sum_{k=1}^{N} \lambda_k y_k \boldsymbol{x}_k^{\mathrm{non}} \|}. \tag{A167}$$

Under these settings,

$$\left\| \boldsymbol{x}_n^{\mathrm{adv}} \right\|^2 = \Theta(d) \pm \Theta\left( \sqrt{\frac{d}{N}} \right) \epsilon + \epsilon^2, \qquad \langle \boldsymbol{x}_n^{\mathrm{adv}}, \boldsymbol{x}_k^{\mathrm{adv}} \rangle \leq p_{\mathrm{max}} + \Theta\left( \sqrt{\frac{d}{N}} \right) \epsilon + \epsilon^2. \tag{A168}$$

Similarly to Theorem C.2, if $\epsilon = \mathcal{O}(\sqrt{d/N})$, the decision boundary is

$$f_{\mathrm{adv}}^{\mathrm{bdy}}(\boldsymbol{z}) := \frac{\sum_{n=1}^{N} \lambda_n^{\mathrm{adv}} y_n^{\mathrm{adv}} \langle \boldsymbol{x}_n, \boldsymbol{z} \rangle}{\sum_{n=1}^{N} \lambda_n^{\mathrm{adv}}} + \epsilon \frac{f^{\mathrm{bdy}}(\boldsymbol{z})}{\| \sum_{n=1}^{N} \lambda_n y_n \boldsymbol{x}_n^{\mathrm{non}} \|} \tag{A169}$$

$$= \Theta\left( \frac{d}{N} \right) \left( f^{\mathrm{bdy}}(\boldsymbol{z}) - \sum_{n=1}^{N} \lambda_n^{\mathrm{adv}} y_n^{\mathrm{adv}} \langle \boldsymbol{x}_n, \boldsymbol{z} \rangle \right). \tag{A170}$$

For $\boldsymbol{z} = \sum_{n=1}^{N} \Theta(1/\sqrt{N}) \boldsymbol{x}_n^{\mathrm{non}}$, $|f^{\mathrm{bdy}}(\boldsymbol{z})| = \Theta(\sqrt{N})$ and $|\sum_{n=1}^{N} \lambda_n^{\mathrm{adv}} y_n^{\mathrm{adv}} \langle \boldsymbol{x}_n, \boldsymbol{z} \rangle| = \Theta(1)$. Thus, if $N, d \to \infty$, then $\mathrm{sgn}(f_{\mathrm{adv}}^{\mathrm{bdy}}(\boldsymbol{z})) = \mathrm{sgn}(f^{\mathrm{bdy}}(\boldsymbol{z}))$ holds. Finally, under Assumption H.1, we can explain why even a flipped learning scenario (corresponding deterministic label scenarios in Table 1) yields moderately high accuracy on natural test datasets.

# I  EXPERIMENTAL SETTINGS AND ADDITIONAL RESULTS

In this section, we present detailed experimental settings and additional empirical results. First, we introduce the settings that are consistent across experiments on artificial and real-world datasets.

We used an NVIDIA A100 GPU. A scheduler reduced a learning rate to 10% of its original value if a training loss did not decrease over 10 consecutive epochs. Uniform noises were sampled from $U([-1,1]^d)$.

As adversarial attacks, we employed projected gradient descent (Madry et al., 2018). The step sizes of projected gradient descent are 0.3 for $L_0$ attacks and $\epsilon/5$ for $L_2$ and $L_\infty$ attacks, respectively. The number of steps is 100 for $L_2$ and $L_\infty$ attacks.

In $L_0$ attacks, the number of modifiable pixels was limited, but the distance between altered and original pixels was not restricted. Sparse attacks employ the following procedures in each step: (i) select the pixel with the highest gradient magnitude from the masked region, (ii) remove the mask for the selected pixel, and (iii) update the perturbations for unmasked pixels. Thus, the number of steps corresponds to the maximum number of modified pixels. Note that the modified pixels shown in Figs. A16 to A18 may not always match the number of steps because we selected the adversarial example with the highest loss over all steps for the final output.

Although we primarily considered geometry-inspired perturbations (cf. Eq. (1)) in the theoretical discussion, we employ gradient-based perturbations (cf. Eq. (A144)) in the experiments. This is due to the practical difficulty of obtaining the decision boundary of a one-hidden-layer neural network (cf. Eq. (A14)) and the value of $\lambda_n$ (cf. Theorem B.1), making geometry-inspired perturbations infeasible in practice.

## I.1  ARTIFICIAL DATASET

The procedure for generating artificial data based on uniform noises can be found in Section 5.1. Similarly, a dataset based on the Gaussian noises $\mathcal{N}(0, \boldsymbol{I}_d)$ was created. Examples of standard, noise, and adversarial samples are shown in Fig. A3. The experimental settings followed the theoretical settings, Section 3.1. We used one-hidden-layer neural networks and stochastic gradient descent with a learning rate of 0.01, momentum of 0.9, and exponential loss. Considering $t \to \infty$, we set the epochs to 100,000. The vectors $\boldsymbol{v}$ and $\boldsymbol{u}$ used to illustrate the decision boundary are defined in Theorem 3.3. In practice, these values are incalculable due to the uncomputable nature of $\lambda_n$. Therefore, we approximated $v$ as the average of the first half of the weights in $\boldsymbol{W}$ and $\boldsymbol{u}$ as the average of the latter half.

The experimental results for artificial datasets based on uniform and Gaussian noises with $L_0$, $L_2$, and $L_\infty$ adversarial perturbations are provided in Figs. A4 to A15. We can confirm a strong alignment between the decision boundaries when learning from standard samples and perturbations (cf. Theorem 4.4). High classification accuracy for standard training data (cf. Corollary 4.5) was also observed across a variety of noise and perturbation forms.

## I.2  MNIST/FASHION-MNIST/CIFAR-10

Examples of standard and adversarial samples for each dataset are shown in Figs. A16 to A18. A six-layer convolutional neural network was used for MNIST and Fashion-MNIST. WideResNet-28-10 with a dropout ratio of 0.3 was used for CIFAR-10. The batch size was set to 128. While no data augmentation was applied to MNIST and Fashion-MNIST, random cropping and horizontal flipping were applied to CIFAR-10. We used stochastic gradient descent with Nesterov momentum of 0.9, weight decay of $5 \times 10^{-4}$, and cross-entropy loss. The initial learning rates can be found in Table A3. The perturbation constraint $\epsilon$ or number of modifiable pixels $d_\delta$ was set according to Table A4. We set the epochs to 100 for MNIST and 200 for Fashion-MNIST and CIFAR-10. However, in the experiments for Fig. A19, we set the epochs to 300, considering a large number of training samples.

For MNIST and Fashion-MNIST, Fig. A19 shows the accuracy of learning from perturbations with various adversarial sample sizes. In several cases, we observed a general increase in the accuracy as the number of samples increased. However, in some cases, while the accuracy did not decrease,

Table A3: Initial learning rates in each learning scenario.

| | On natural samples | | | | | | On noise data | | |
|---|---|---|---|---|---|---|---|---|---|
| | $L_0$ (R) | $L_0$ (D) | $L_2$ (R) | $L_2$ (D) | $L_\infty$ (R) | $L_\infty$ (D) | $L_0$ | $L_2$ | $L_\infty$ |
| MNIST | 0.1 | 0.1 | 0.01 | 0.01 | 0.01 | 0.01 | 0.1 | 0.1 | 0.1 |
| FMNIST | 0.1 | 0.1 | 0.01 | 0.01 | 0.01 | 0.01 | 0.1 | 0.1 | 0.1 |
| CIFAR-10 | 0.1 | 0.1 | 0.1 | 0.1 | 0.01 | 0.1 | 0.1 | 0.1 | 0.1 |

Table A4: Perturbation constraint $\epsilon$ or number of modifiable pixels $d_\delta$ for MNIST, Fashion-MNIST, and CIFAR-10.

| MNIST | | | Fashion-MNIST | | | CIFAR-10 | | |
|---|---|---|---|---|---|---|---|---|
| $L_0$ | $L_2$ | $L_\infty$ | $L_0$ | $L_2$ | $L_\infty$ | $L_0$ | $L_2$ | $L_\infty$ |
| 10 | 2.0 | 0.3 | 35 | 2.0 | 0.3 | 150 | 0.5 | 0.1 |

there was no significant improvement in the accuracy. These inconsistencies could potentially be resolved through extensive experiments and tuning of learning rates and architectures.

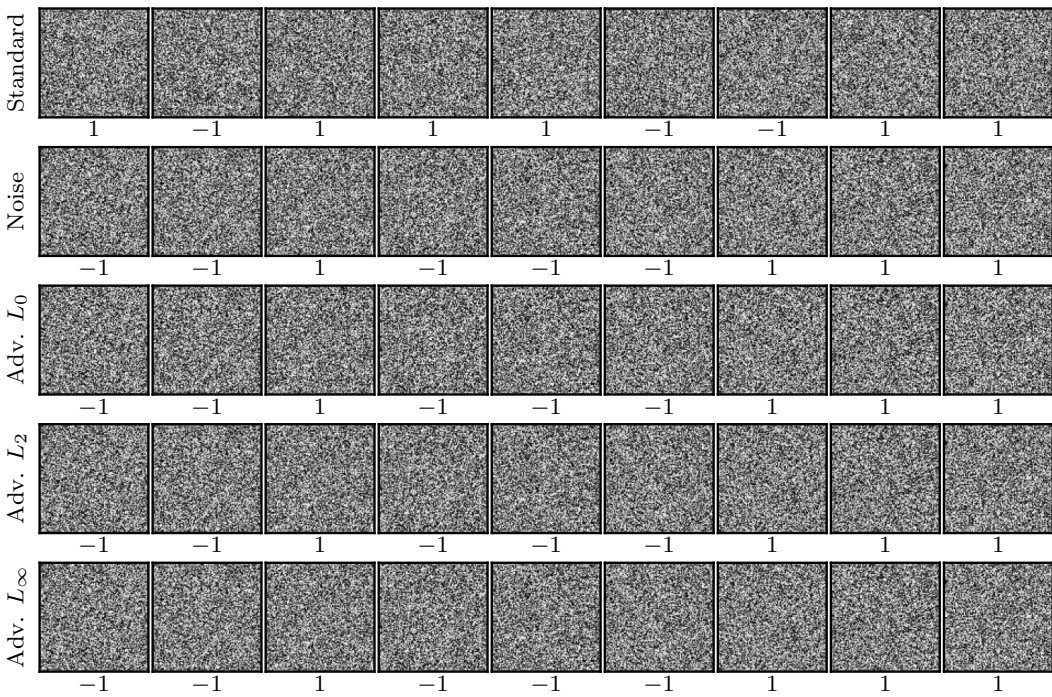

Figure A3: Artificial data based on uniform noises. Standard images were drawn from the uniform distribution $U([-1, 1]^d)$, and their corresponding labels from $U(\{\pm 1\})$. We treated them as *natural* images. Noise images are similarly drawn from $U([-1, 1]^d)$. Adversarial examples were generated to superimpose adversarial perturbations on the noise images to fool a classifier trained on the standard (but seemingly noisy) images. The labels below the adversarial examples indicate target labels that were randomly sampled from $\{\pm 1\}$. Those below the noise images were used for comparative experiments in training classifiers on these noises.

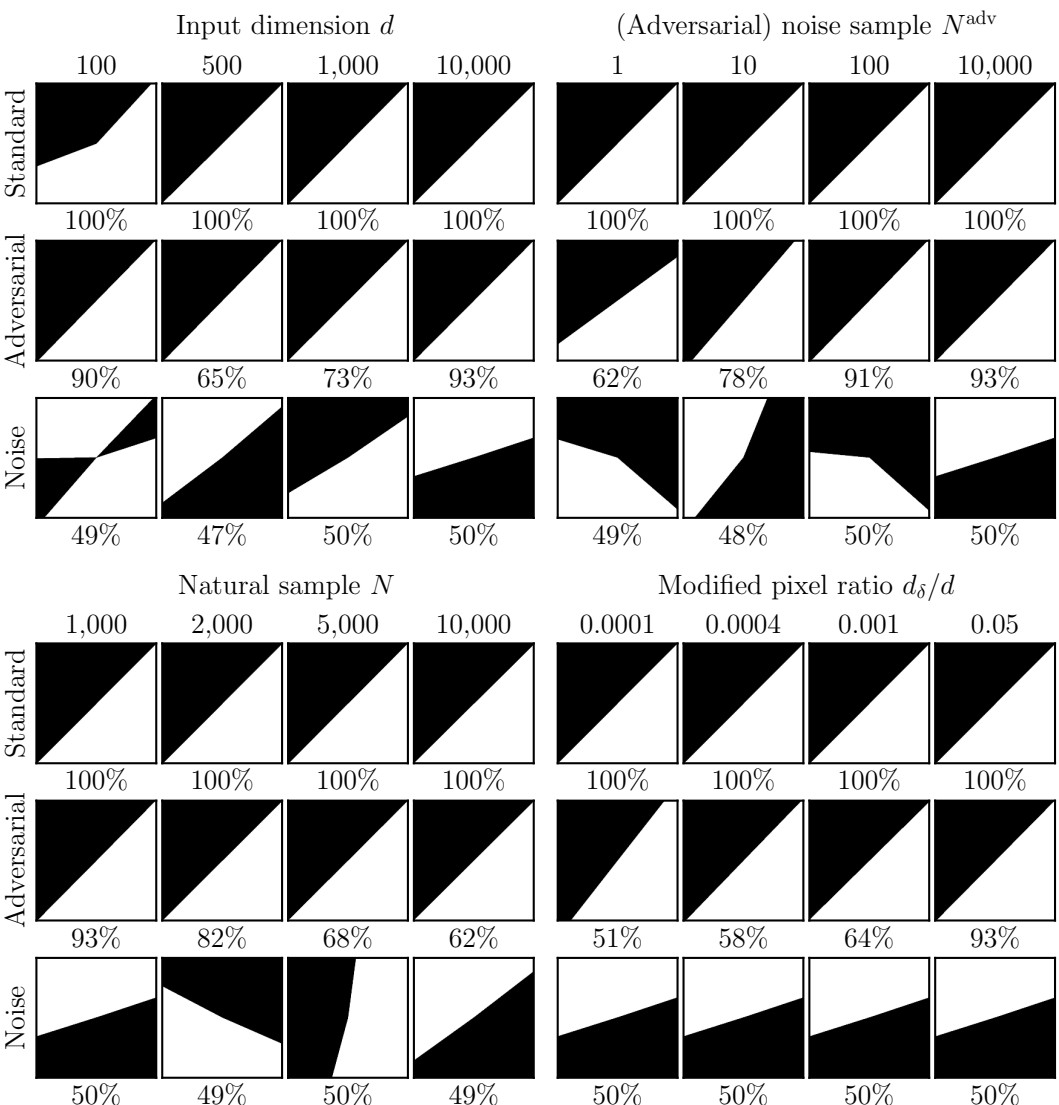

Figure A4: Decision boundaries of classifiers trained on artificial datasets based on uniform noises and $L_0$ adversarial perturbations. Each variable was varied based on $d = 10,000$, $N^{\text{adv}} = 10,000$, $N = 1000$, and $d_\delta/d = 0.05$. The description is the same as Fig. 1.

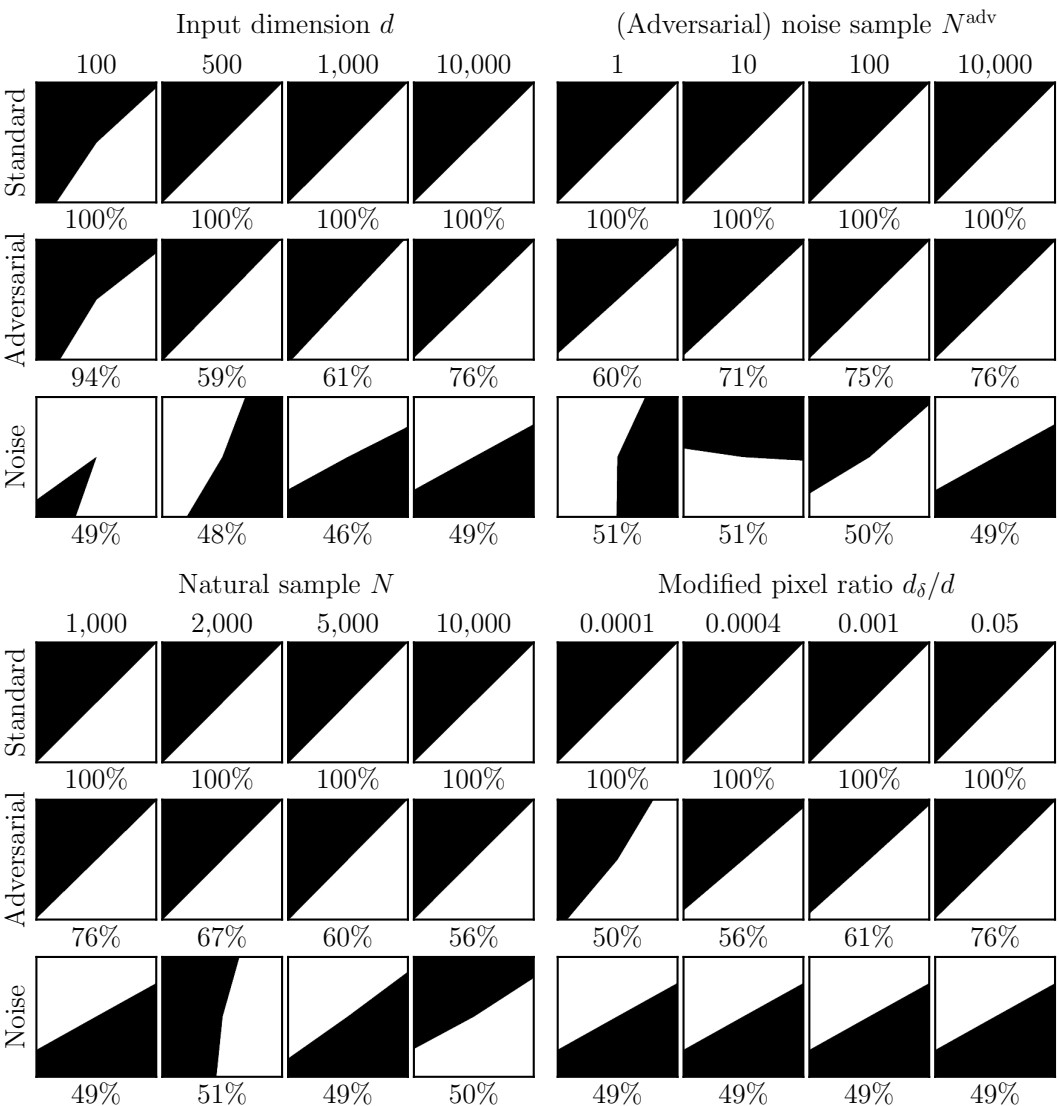

Figure A5: Decision boundaries of classifiers trained on artificial datasets based on Gauss noises and $L_0$ adversarial perturbations. The description is the same as Fig. A4.

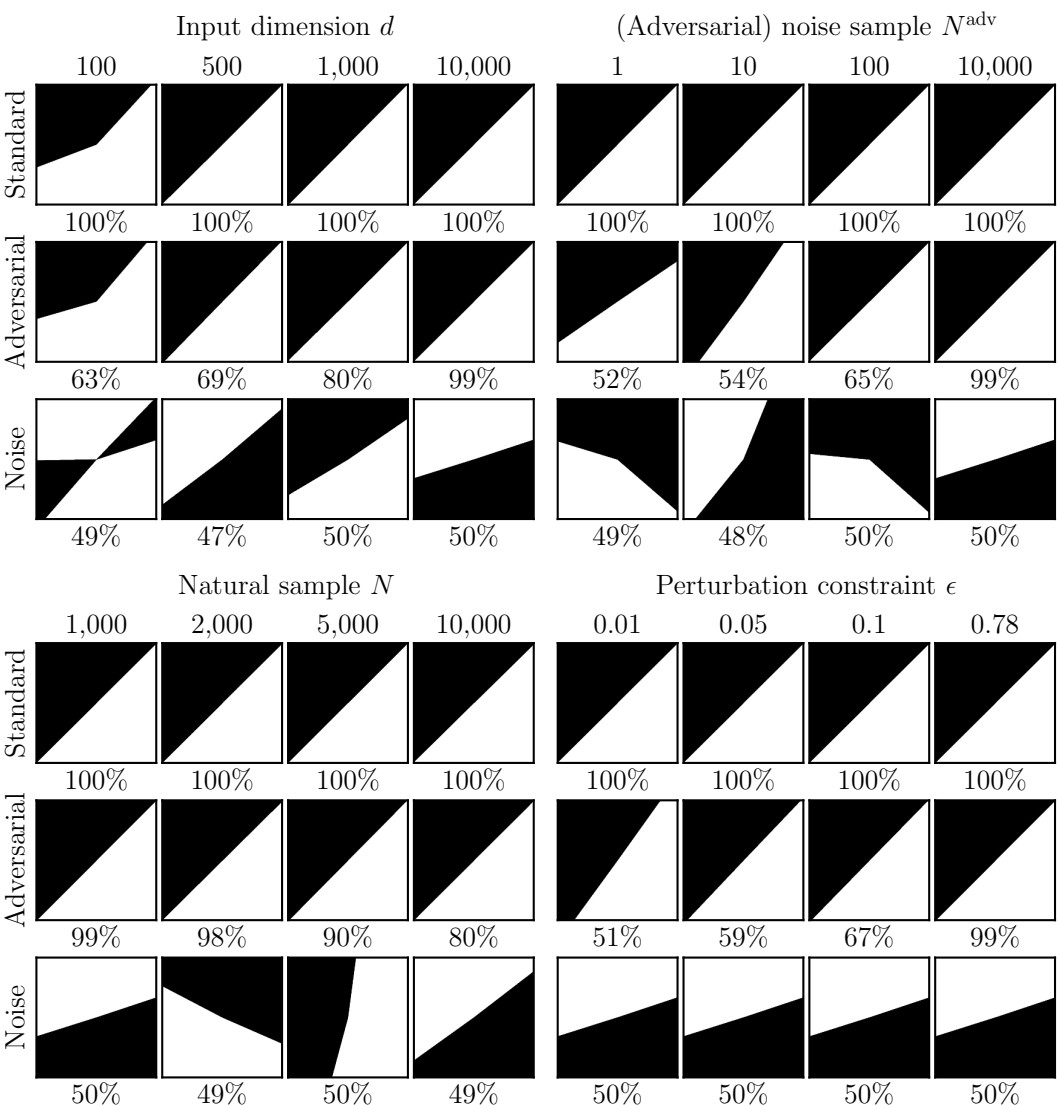

Figure A6: Decision boundaries of classifiers trained on artificial datasets based on uniform noises and $L_2$ adversarial perturbations. Each variable was varied based on $d = 10,000$, $N^{\mathrm{adv}} = 10,000$, $N = 1000$, and $\epsilon = 0.78$. The description is the same as Fig. 1.

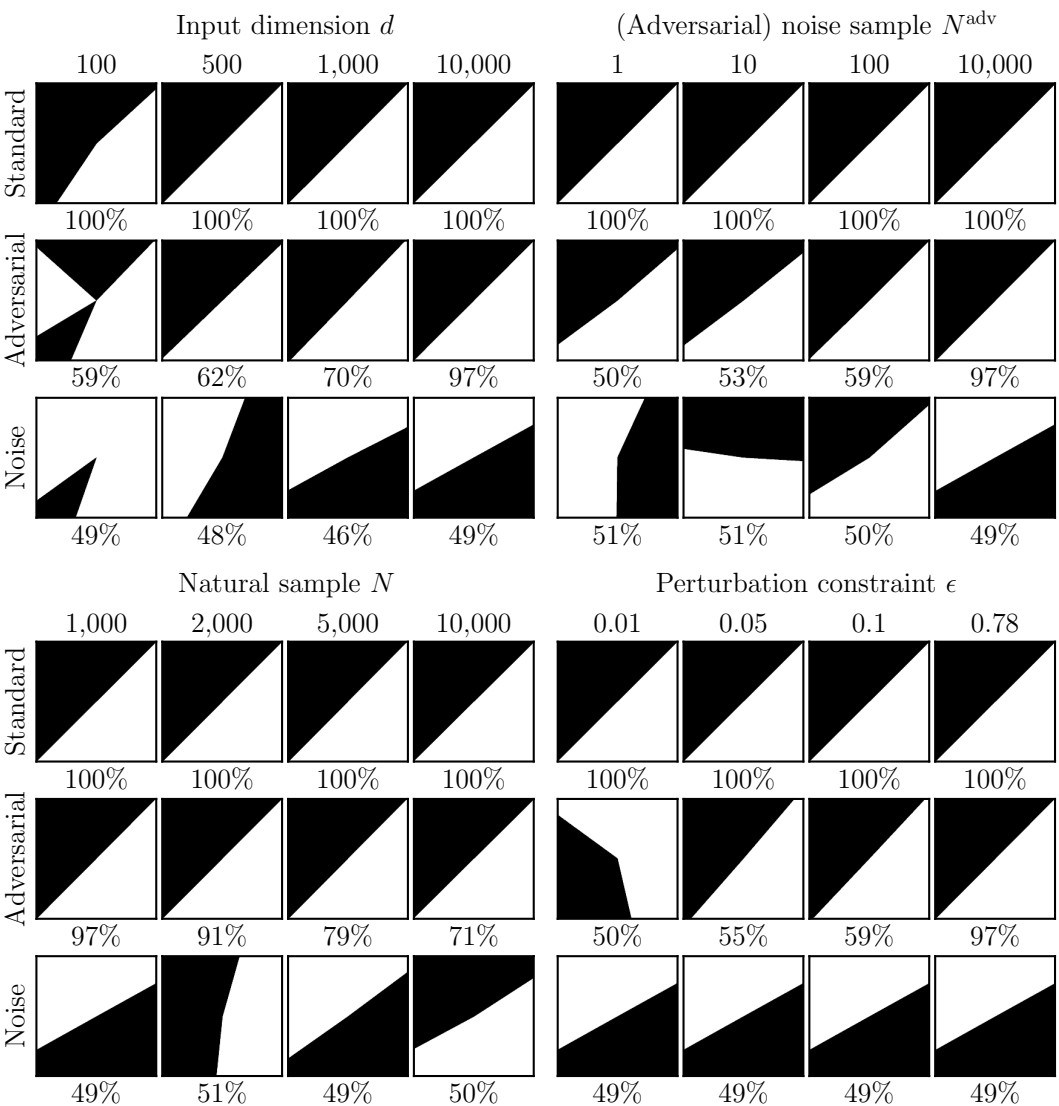

Figure A7: Decision boundaries of classifiers trained on artificial datasets based on Gauss noises and $L_2$ adversarial perturbations. The description is the same as Fig. A6.

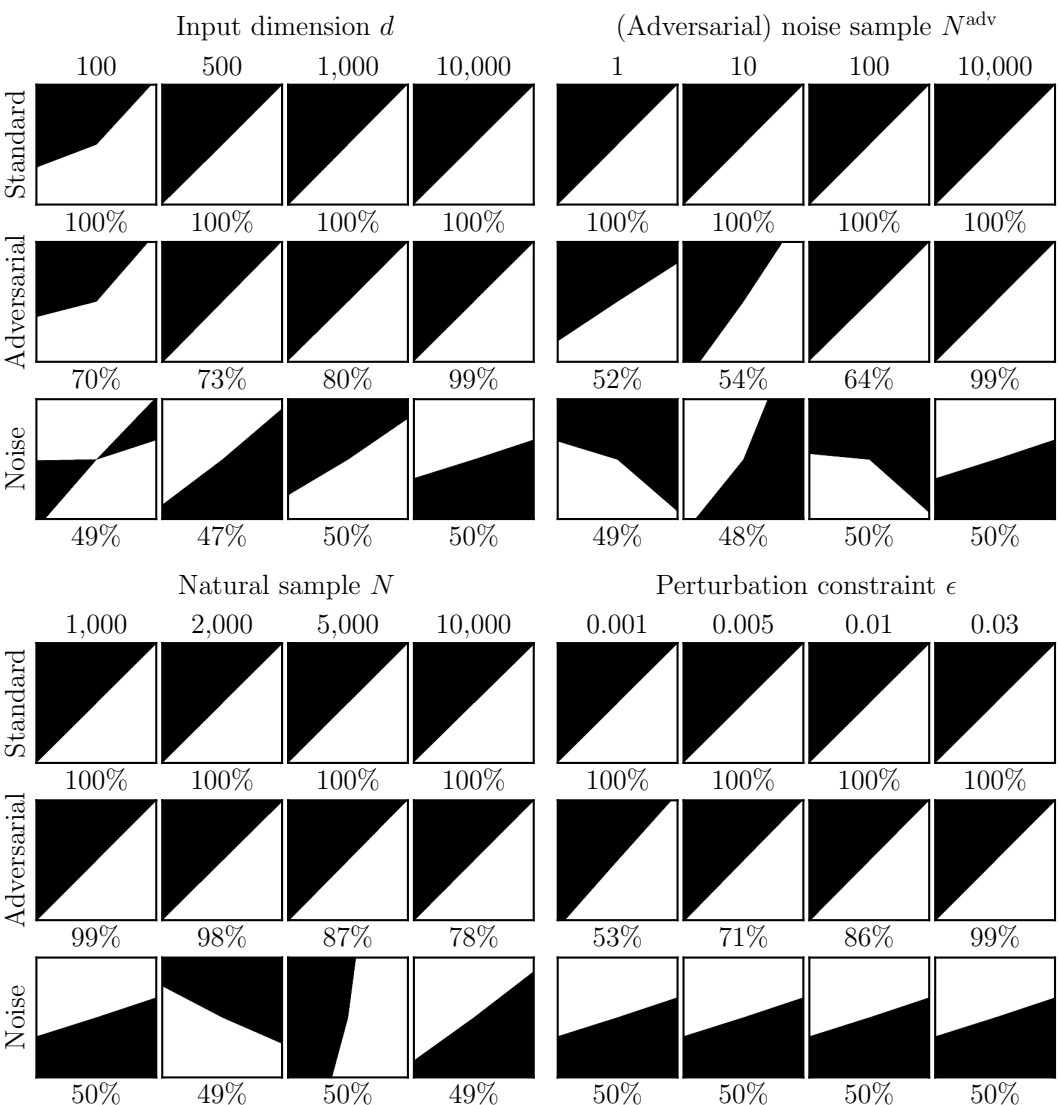

Figure A8: Decision boundaries of classifiers trained on artificial datasets based on uniform noises and $L_\infty$ adversarial perturbations. Each variable was varied based on $d = 10,000$, $N^{\mathrm{adv}} = 10,000$, $N = 1000$, and $\epsilon = 0.03$. The description is the same as Fig. 1.

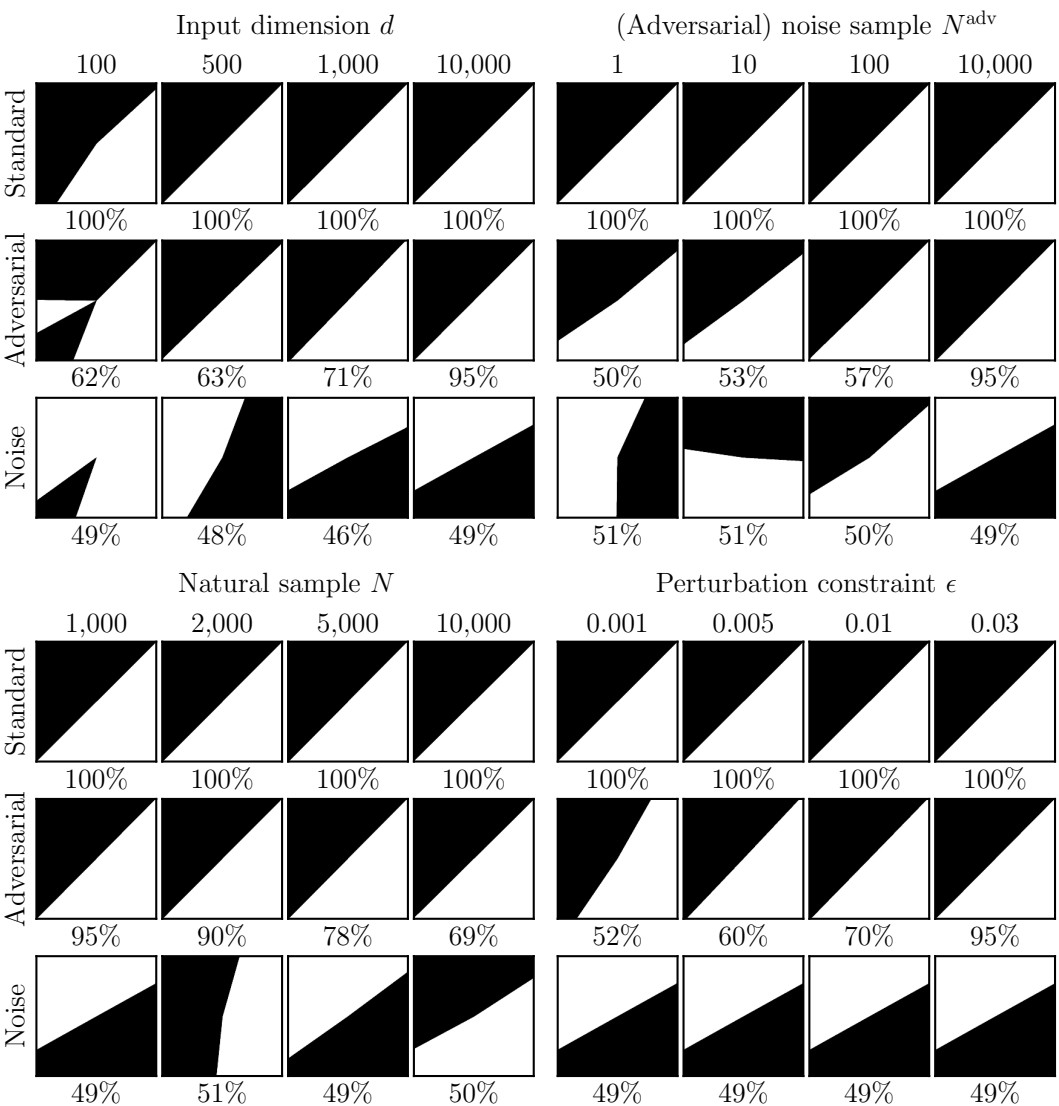

Figure A9: Decision boundaries of classifiers trained on artificial datasets based on Gauss noises and $L_\infty$ adversarial perturbations. The description is the same as Fig. A8.

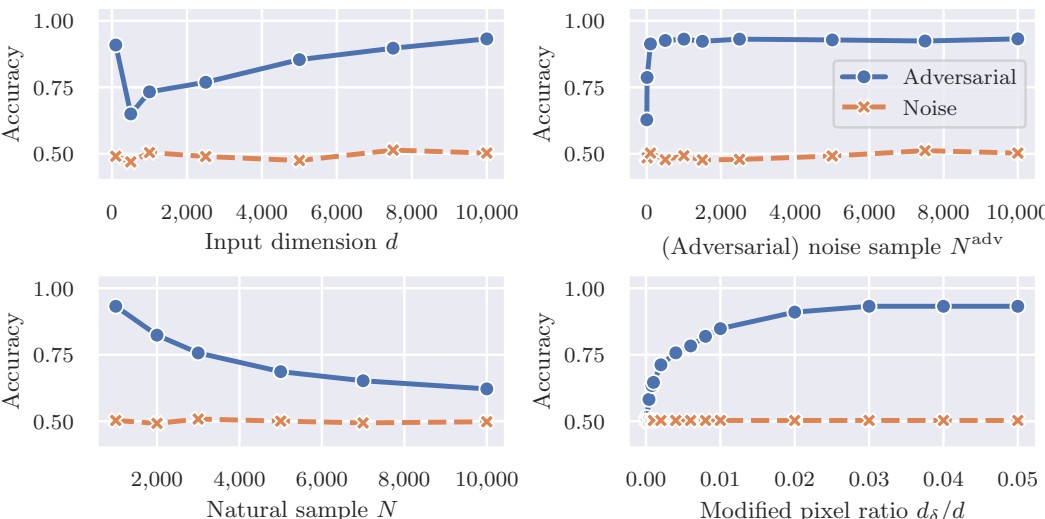

Figure A10: Accuracy of classifiers trained on artificial datasets based on uniform noises and $L_0$ adversarial perturbations. The description is the same as Figs. 2 and A4.

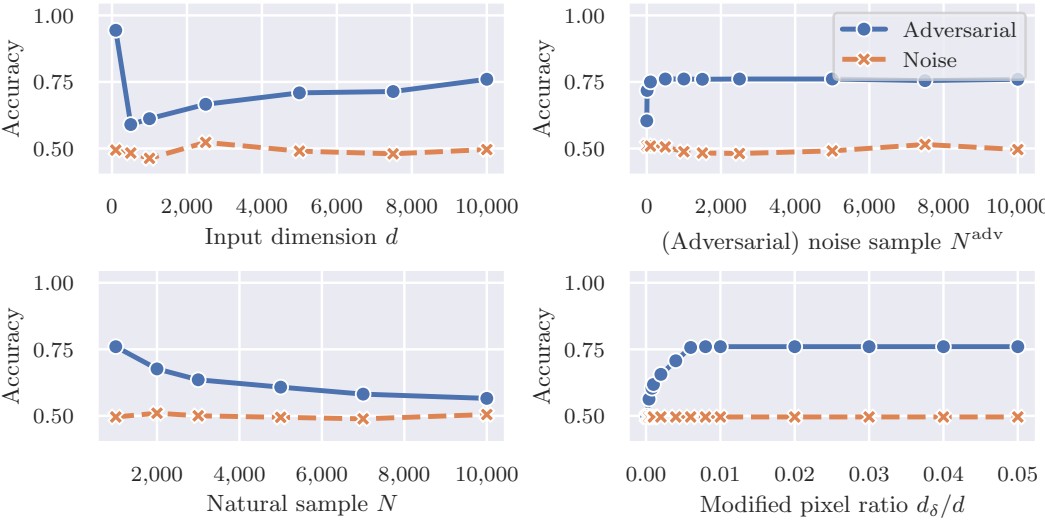

Figure A11: Accuracy of classifiers trained on artificial datasets based on Gauss noises and $L_0$ adversarial perturbations. The description is the same as Figs. 2 and A5.

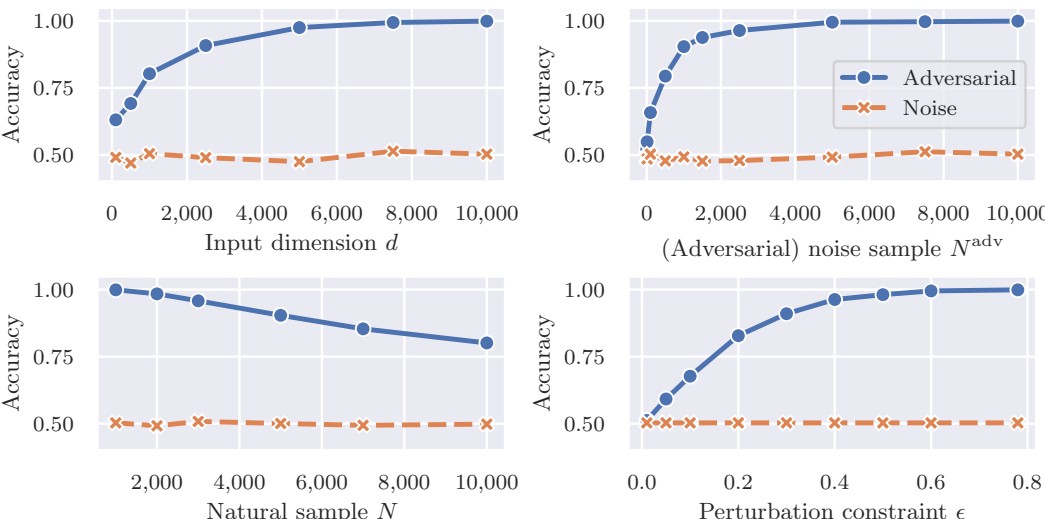

Figure A12: Accuracy of classifiers trained on artificial datasets based on uniform noises and $L_2$ adversarial perturbations. The description is the same as Figs. 2 and A6.

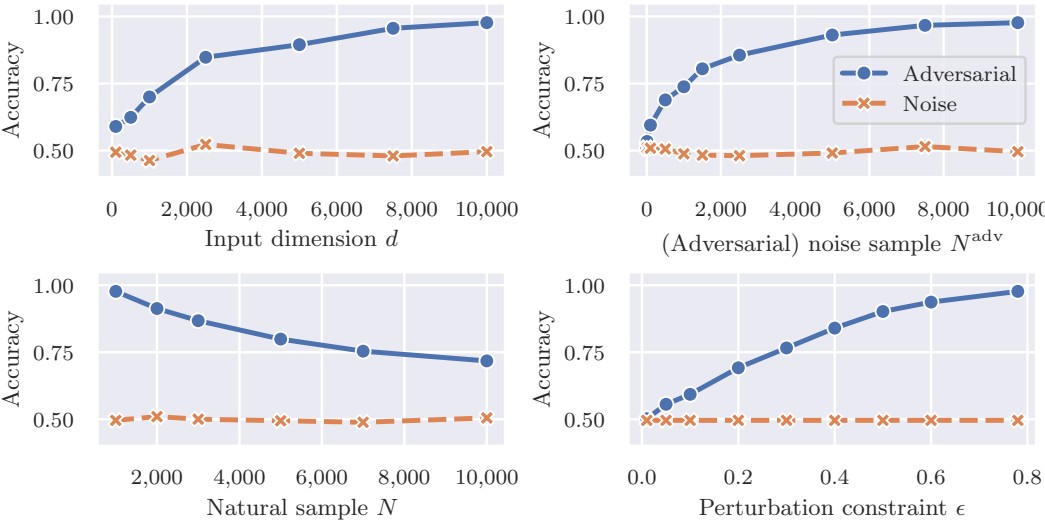

Figure A13: Accuracy of classifiers trained on artificial datasets based on Gauss noises and $L_2$ adversarial perturbations. The description is the same as Figs. 2 and A7.

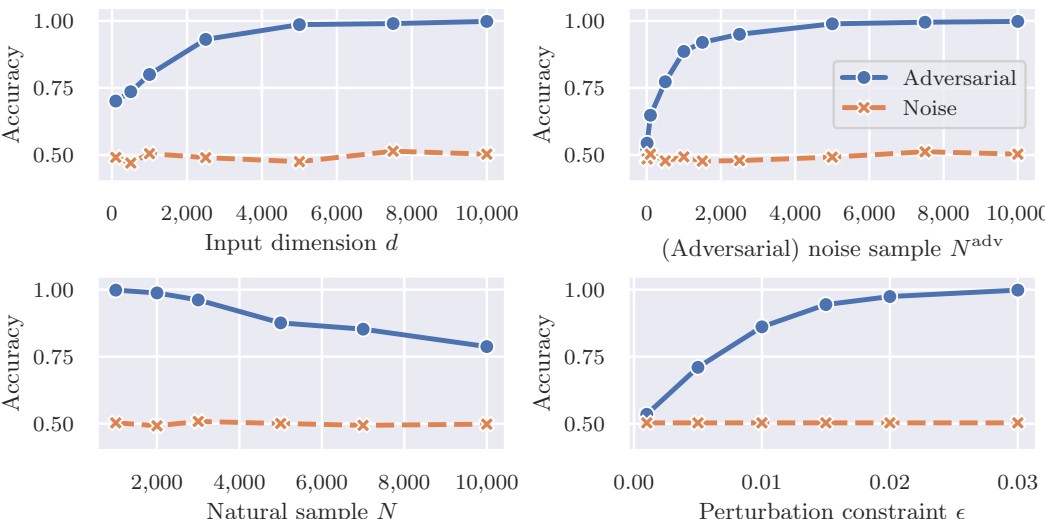

Figure A14: Accuracy of classifiers trained on artificial datasets based on uniform noises and $L_\infty$ adversarial perturbations. The description is the same as Figs. 2 and A8.

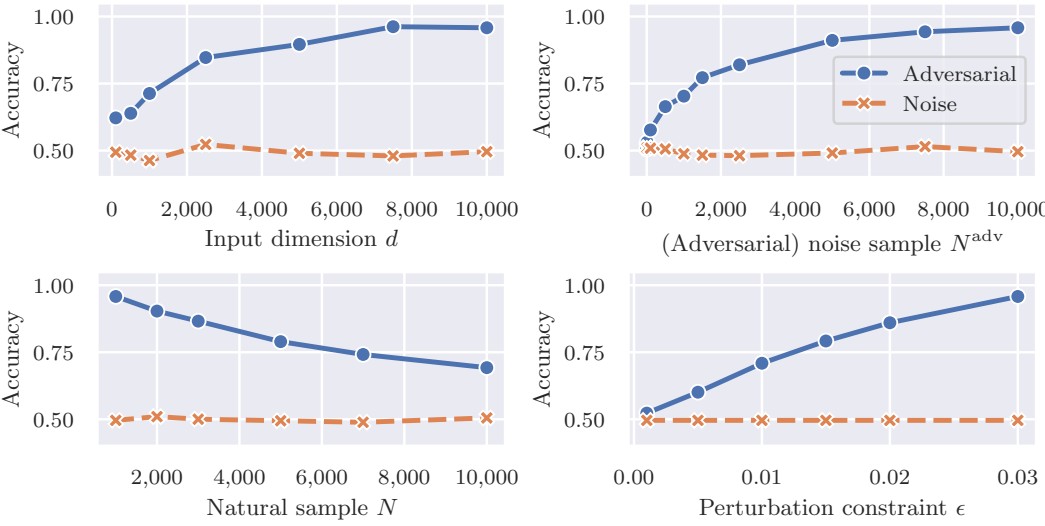

Figure A15: Accuracy of classifiers trained on artificial datasets based on Gauss noises and $L_\infty$ adversarial perturbations. The description is the same as Figs. 2 and A9.

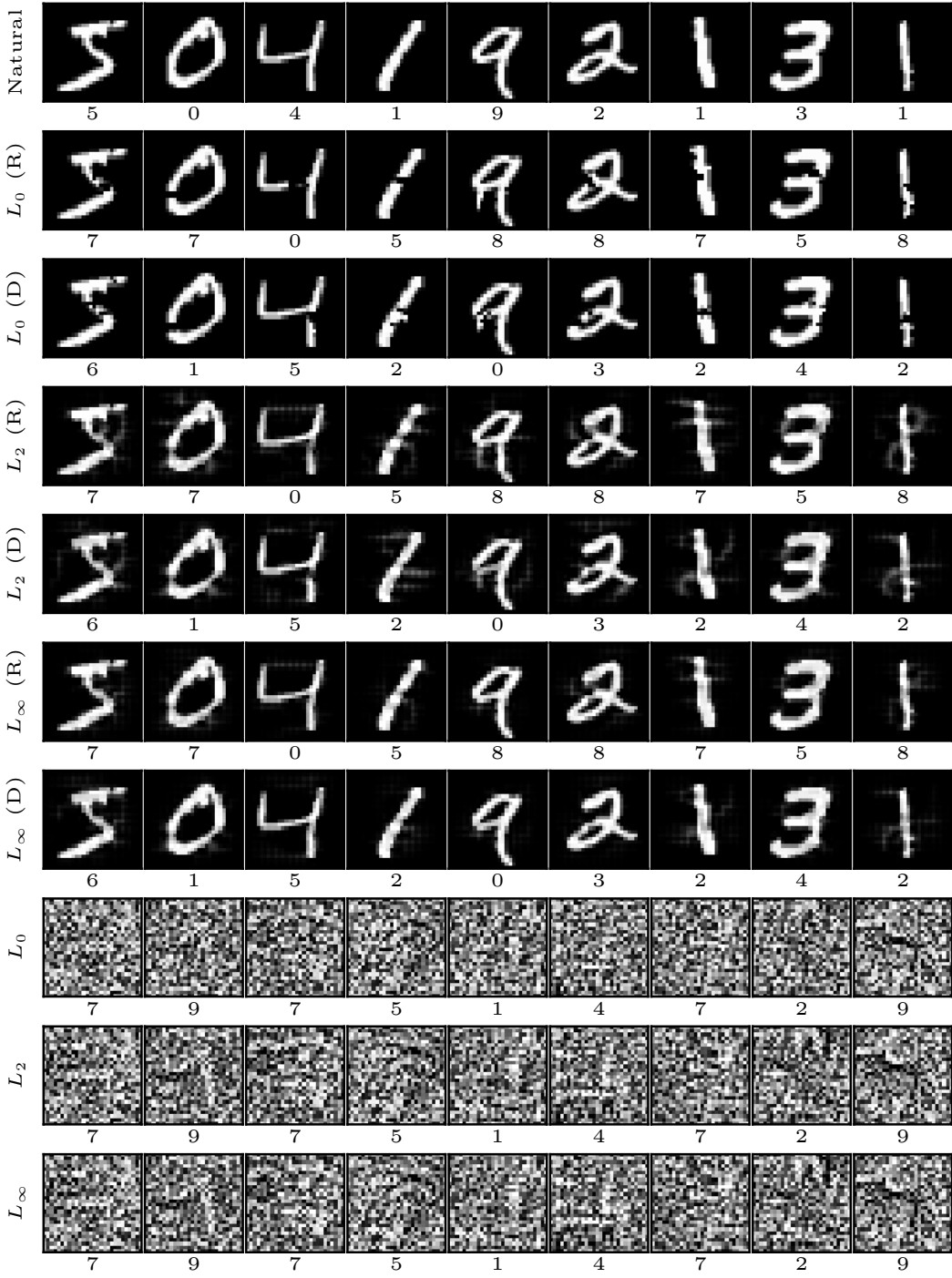

Figure A16: Natural images, perturbations on natural images, and perturbations on uniform noises for MNIST. Below the natural images and adversarial examples are their respective original and target labels in adversarial attacks. The "R" indicates that a target label was randomly chosen from the nine labels that differ from an original label. The "D" indicates that a target label was deterministically chosen as the next sequential label after an original label.

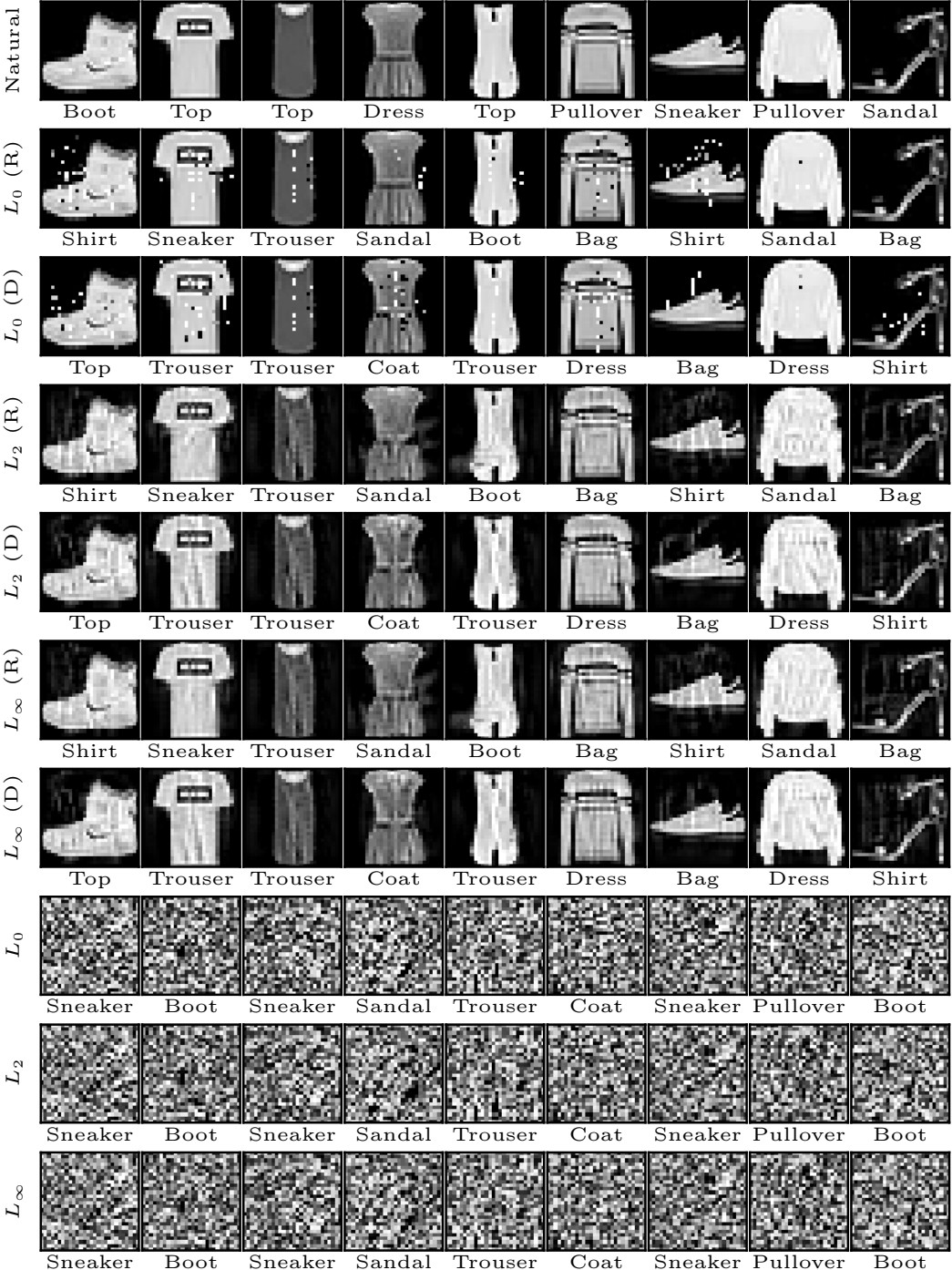

Figure A17: Natural images, perturbations on natural images, and perturbations on uniform noises for Fashion-MNIST. The description is the same as Fig. A16.

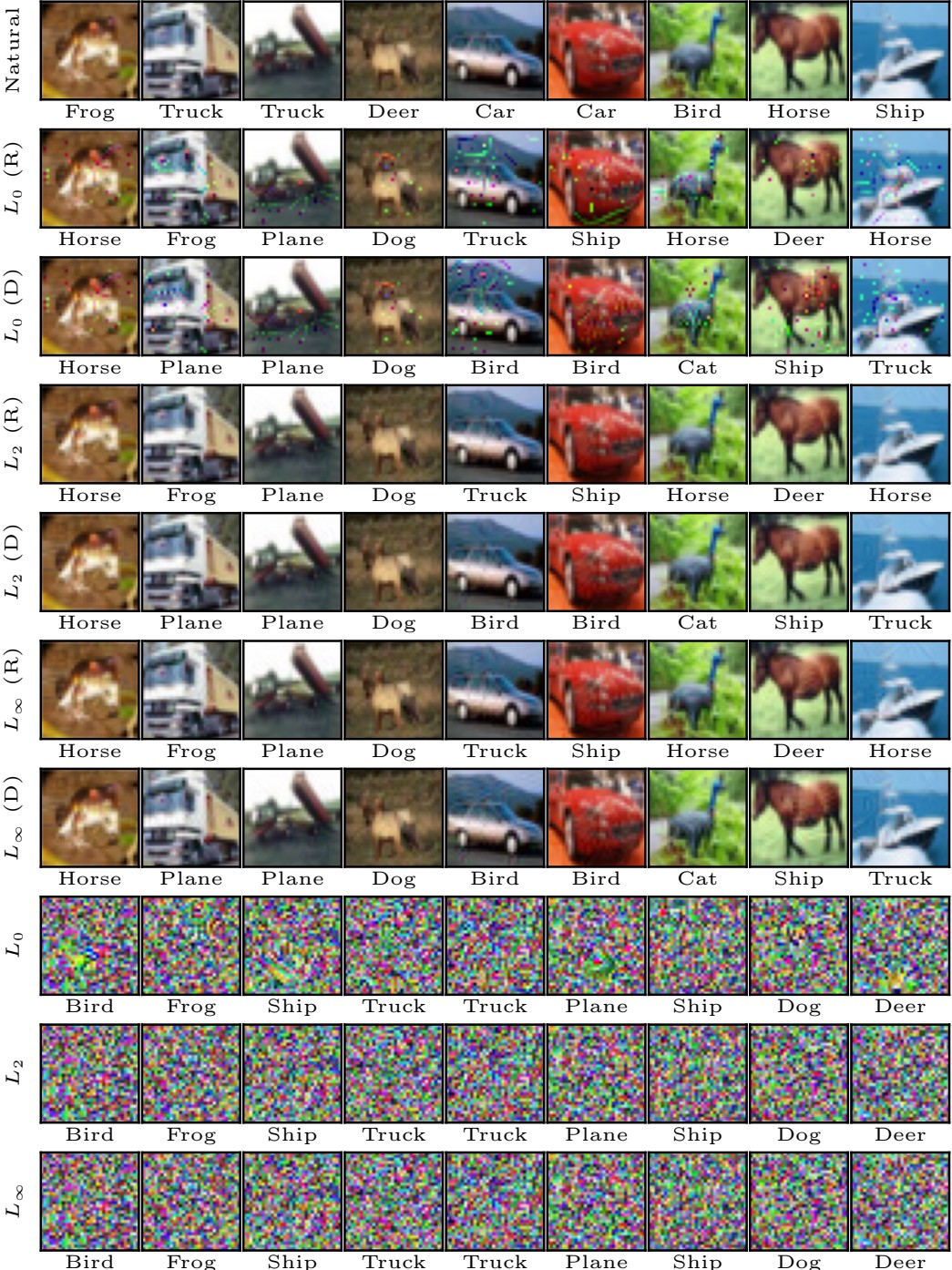

Figure A18: Natural images, perturbations on natural images, and perturbations on uniform noises for CIFAR-10. The description is the same as Fig. A16.

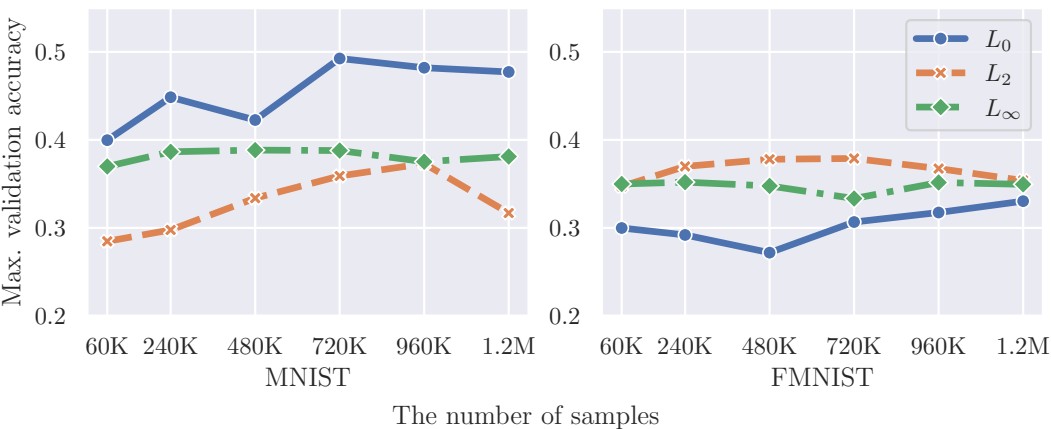

Figure A19: Accuracy of classifiers trained on MNIST and Fashion-MNIST in uniform noise scenario. The vertical axis represents the maximum validation accuracy across all epochs. The horizontal axis represents the number of adversarial samples $N^{\mathrm{adv}}$.

