# OpenReview forum: "Theoretical Understanding of Learning from Adversarial Perturbations"
_ICLR.cc/2024/Conference — ICLR 2024 poster_

### Official Review · Reviewer_75Lo · 2023-10-25

**Soundness:** 2 fair
**Presentation:** 3 good
**Contribution:** 3 good
**Rating:** 6
**Confidence:** 4

**Summary:**

In this paper, the authors focus on the embedded information in adversarial perturbations. Motivated by the prior work (Ilyas et al. 2019), they theoretically demonstrate that the adversarial perturbation can be represented as a weighted combination of the natural datapoints. And the model trained on these adversarial perturbations (with incorrect labels) can efficiently learn the data features and generate well on the natural data. Specifically, they provide some theorems showing the boundary of the model learning from the purturbations is partially determined by the model learning from the natural data. Some experiments are conducted to validate their theory.

**Strengths:**

* The writing is generally clear.
* Although there are some assumptions about the dataset and they only focus on one-hidden-layer neural network, the authors provide some insights into this research field.
* The problem of why network learning from perturbation can generate well is worth exploring. This can shed some light on other interesting phenomena, such as the transferability of adversarial examples.

**Weaknesses:**

My main concerns about this paper are:
* In Section 4.1, the authors generate the adversarial examples according to the linear model. I understand the decision boundaries of the linear model $f^{bdy}(z)$ and the one-hidden-layer neural network are the same (Theorem 3.3). However, I don't think we can directly replace the neural work with the linear model to generate adversarial examples since adversarial examples are generated by optimizing the loss, which is different. Can the authors provide more theoretical explanations for this?
* Although the authors claim that they focus on one-hidden-layer neural networks, they make some approximations and actually focus on linear models. So the contribution of this paper is reduced.

Some minor concerns:
* In the third paragraph of the introduction, the authors say, "Nevertheless, neural networks can learn to accurately classify benign samples from such adversarially perturbed samples with seemingly incorrect labels". Can you provide more explanation for this? Does it mean we can train a model that can accurately distinguish the adversarial examples from the clean sample?
* In this work, the authors show that the model trained on adversarial perturbations with incorrect labels (i.e., $y^{adv}$) can generate well on the clean data. But as we know, traditional adversarial training is based on adversarial examples with correct labels (i.e., $y^{natural}$). So, why do these two methods both generate well on the clean data?
* I would like to know the adversarial accuracy of the method in your experiments.
* In Eq. (2), does $x_n$ refer to the training data $x_n$? It seems that $x_n$ here is new test data that is different from the training data $x_n$. Otherwise, the gradient calculation in Eq. (2) should be reconsidered. Please check this out.
* The assumption in Theorem 4.2 is somewhat strong: $|\sum\lambda_n y_n <x_n,z>|=\Theta(g(N,d))$ if $\sum\lambda_n  |<x_n,z>|=\Theta(g(N,d))$. It is intuitive that the first term is much smaller than the second term. What would your theorem be without this assumption?
* In Section 5.1, I have some confusion about some statements. For example, in Fig 1, the authors mentioned standard data and noisy data. But in their setting, there are two kinds of noisy data: the dataset is generated randomly and can be viewed as some noises; the adversarial perturbations are superposed on random noises. I am confused as to what "standard data" and "noise" refer to. I strongly recommend the authors make more clear clarifications in the paper.
* As the authors claim in the experiments, there are some counterexamples. I think more discussions about these counterexamples are needed. Is it because the effect of learning from mislabeled natural data dominates the effect of learning from perturbation?
* Since there are negative effects of learning from natural data or noises, why doesn't the learner solely learn from the perturbations?

**Questions:**

Please refer to the Weakness part.

---

> ### Author Response · Authors · 2023-11-13
> **Response to Reviewer 75Lo [1/1]**
>
> We appreciate the reviewer’s thorough review of our work.
>
> **First, we would like to clarify that our research is NOT related to adversarial training. Please kindly refer to the general comment.**
>
> ### Main concerns
> > the authors generate the adversarial examples according to the linear model. ... I don't think we can directly replace the neural work with the linear model ... since adversarial examples are generated by optimizing the loss
> >
> > Although the authors claim that they focus on one-hidden-layer neural networks, they make some approximations and actually focus on linear models. So the contribution is reduced.
>
> In Appendix D.1, we discuss the perturbations generated by optimizing the exponential loss on the one-hidden-layer neural network $f$ itself rather than the linear model $f^\mathrm{bdy}$. The results are consistent with those in the main text, i.e., those for the perturbations targeting the linear model. Please refer to it.
>
> ### Minor concerns
> > the authors say, "Nevertheless, neural networks can learn to accurately classify benign samples from such adversarially perturbed samples with seemingly incorrect labels". Can you provide more explanation for this? Does it mean we can train a model that can accurately distinguish the adversarial examples from the clean sample?
> >
> > the authors show that the model trained on adversarial perturbations with incorrect labels can generate well on the clean data. But as we know, traditional adversarial training is based on adversarial examples with correct labels. So, why do these two methods both generate well on the clean data?
> >
> > I would like to know the adversarial accuracy
>
> **Our research focus is NOT adversarial training. Please kindly refer to the general comment.**
>
> > In Eq. (2), does $x_n$ refer to the training data $x_n$? It seems that $x_n$ here is new test data that is different from the training data.
>
> Yes, in Eq. (2), $x_n$ denotes the training data. We would appreciate more details on your interpretation of $x_n$ as new test data.
>
> > The assumption in Theorem 4.2 is somewhat strong. It is intuitive that the first term is much smaller than the second term. What would your theorem be without this assumption?
>
> We acknowledge that this assumption introduced in the natural data scenario (cf. Definition 3.2) is not always reasonable. However, to determine the growth rate of $T_2$, i.e., the second term in Eq. (4), which includes the indeterminable $y_n$, certain assumptions are necessary. While this assumption can be relaxed, for example, to $|\sum\lambda_ny_n\langle x_n,z\rangle|=\Theta(1)$, it inevitably weakens the conclusion of Theorem 4.2.
>
> To circumvent these issues, we employed the noise data scenario. This approach allows us to validate the success of learning from perturbations without relying on such unreasonable assumptions (cf. Theorem 4.4). This feasibility stems from the fact that in the noise data scenario the base data (i.e., uniform noises) have well-defined probabilistic characteristics. This differs from the natural data scenario, where the probabilistic nature of the base data (i.e., natural data) and the corresponding labels cannot be defined.
>
> > In Section 5.1, I have some confusion about some statements. For example, in Fig 1, the authors mentioned standard data and noisy data. But in their setting, there are two kinds of noisy data: the dataset is generated randomly and can be viewed as some noises; the adversarial perturbations are superposed on random noises. I am confused as to what "standard data" and "noise" refer to. I strongly recommend the authors make more clear clarifications in the paper.
>
> The reviwer is correct; "standard data" refers to images that might appear as noise to the human eye in this section. We will enhance the clarity of our experimental setup and the nature of the images in the revised version.
>
> > As the authors claim in the experiments, there are some counterexamples. I think more discussions about these counterexamples are needed. Is it because the effect of learning from mislabeled natural data dominates the effect of learning from perturbation?
>
> Yes, following Theorems 4.1 and 4.3, the dominance of learning effects from mislabeled natural data over learning effects from perturbations is a plausible explanation. However, the theoretical evaluation of this effect is challenging in datasets such as MNIST, Fashion-MNIST, and CIFAR-10, where the data distribution cannot be quantitatively assessed.
>
> > Since there are negative effects of learning from natural data or noises, why doesn't the learner solely learn from the perturbations?
>
> This is because classifiers cannot discern what is "negative" within a given dataset and are trained solely to minimize the loss on the dataset. Learning from mislabeled natural data or noises is counterproductive from our perspective to achieve high accuracy on a test dataset, but it serves as a useful feature to strongly fit a mislabeled training dataset.

---

### Official Review · Reviewer_doyy · 2023-10-29

**Soundness:** 2 fair
**Presentation:** 1 poor
**Contribution:** 2 fair
**Rating:** 5
**Confidence:** 3

**Summary:**

This paper consider orthogonal training data, and prove that one hidden layer neural network trained with mislabeled data should have the sae dicision boundary with training on the natural data. Moreover, the adversarial perturbation can be treated as the weighted sum of benign training data.

**Strengths:**

This paper provides a theoretical justification regarding an empirical phenomenon that first observed in Ilyas et a. 2019. To the best of my knowledge there’s no theoretical work has been focused on this direction before.

**Weaknesses:**

I’m not exactly following the motivation of this work. Normally when people consider adversarial training to gain robustness, at each iteration the adversarial training examples is generated based on the current model weight, yet in this paper, from the definition 3.2, it seems that the adversarial examples of training samples is generated beforehand and independent of the current model weight, and it’s fixed during the latter training procedure. Therefore it’s unclear to me whether the perturbation considered in this paper is actually adversarial or not. The author claims the motivation comes from Ilyas et a. 2019, whereas in that paper there’s also a math formulation of generating the designed adversarial perturbation, which seems different from the procedure described in definition 3.2.

It seems this paper’s theoretical results largely build on the result from Frei et al 2023. Yet the discussion of the existing result and the corollary is limited. It’s unclear to me why corollary 3.4 can be directly derived from Theorem 3.3. Theorem 3.3 requires clean training data to be nearly orthogonal, then for corollary 3.4 you should also prove that the adversarial example is almost orthogonal. Moreover, the adversarial perturbation considered depends on the decision boundary, but it’s unclear to me whether the coefficient $\lambda$ in the decision boundary is computable. The network only considers Leaky ReLU, yet it’s unclear why cannot generalize to other activation functions and what’s the technical difficulty.

The statement of each theorem is not self-contented. If $R_{min}, R_{max}, p_{max}$ is used all over the place, then they should be introduced in the notation part. The explanation of each theorem and corollary is not clear. Why we require equation (3)? Any intuitive explanation or simply one requirement for the proof to go through? When N goes to infinity, equation (3) is saying $p_{max}$ is negative, which is kind of counterintuitive. Similarly, what does equation (6) hold mean intuitively?

**Questions:**

I believe the writing of the paper can be largely improved. Most of the questions I have here and in the previous section are mainly because of the unclear writing or cannot find any pointers.

1. Can the author provide any intuition as to why mislabeled data provide the same classifier direction as clean data? Is it mainly because of the orthogonality of training data so that even if the label is noisy the data is still orthogonal?

2. For the discussion of theorem 4.1 ``Consistent Growth of Two Effects”, it’s unclear to me whether estimating the growth rate of T1, T2 by switching the absolute operator and the summation is reasonable. The magnitude could have a huge difference with proper label $y_n$, so why assume the rate of switching the operator is reasonable?

3. It’s unclear from a theoretical perspective in Appendix G why the standard decision boundary only depends on non-robust features as shown in equation A157.

4. For Figure 2, and other figures, does the accuracy clean accuracy or adversarial robust accuracy? What’s the procedure to generate adversarial perturbation? Does it relate to the decision boundary? What’s the point of considering random selection vs. deterministic selection? Does the perturbation size actually align with what indicates by the theorem such as $O(d/N)$ or $O(\sqrt{d/N})$?

---

> ### Author Response · Authors · 2023-11-13
> **Response to Reviewer doyy [1/2]**
>
> We would like to thank the reviewer’s insightful comments.
>
> **First, we would like to clarify that our research is NOT related to adversarial training. Please kindly refer to the general comment.**
>
> ### Misinterpretation
> > when people consider adversarial training ...
>
> **Our research focus is NOT adversarial training.**
>
> > the motivation comes from Ilyas et a. 2019, ..., which seems different from definition 3.2
>
> The reviewer may refer to the "Robust training" in section 2 of (Ilyas et al.), which is not the focus of our study. Learning from perturbations is described in Section 3.2 of (Ilyas et al.).
>
> > why mislabeled data provide the same classifier direction as clean data? because the orthogonality of training data so that even if the label is noisy the data is still orthogonal?
>
> It seems this question arises from a misunderstanding of our experimental procedure as adversarial training. For the explanation of the success of learning from mislabeled data, please refer to the general comment.
>
> This is because classifiers learn from mislabeled adversarial examples, not just mislabeled data. Following the hypothesis of Ilyas et al., adversarial perturbations contain invisible data features to humans. While learning on mislabeled adversarial examples seems nonsensical to us, classifiers rely on these invisible but generalizable features and can construct the same decision boundaries as those of standard classifiers.
>
> ### Theoretical framework (Theorem 3.3 and Corollary 3.4)
> > the discussion of the existing result and the corollary is limited
>
> The discussion has been summarized in Appendix A.
>
> > why corollary 3.4 can be directly derived from Theorem 3.3
>
> Corollary 3.4 is the special case of Theorem 3.3 for $\\{(x^\mathrm{adv}\_n,y^\mathrm{adv}_n)\\}^{N^\mathrm{adv}}\_{n=1}$. In other words, we replace $\\{(x\_n,y\_n)\\}^N\_{n=1}$ with $\\{(x^\mathrm{adv}\_n,y^\mathrm{adv}\_n)\\}^{N^\mathrm{adv}}\_{n=1}$. This change naturally leads to a modification in definitions, such as $R\_\mathrm{max}:=\max\_n\\|x\_n\\|$ becoming $R^\mathrm{adv}\_\mathrm{max}:=\max\_n\\|x^\mathrm{adv}\_n\\|$. Essentially, Corollary 3.4 is Theorem 3.3 with "adv" added to the variables.
>
> > for corollary 3.4 you should prove that the adversarial example is almost orthogonal.
> >
> > what do Eq. 3 and Eq. 6 mean?
>
> The orthogonality condition of adversarial examples is $\gamma^3{R^\mathrm{adv}\_\mathrm{min}}^4/(3N^\mathrm{adv}{R^\mathrm{adv}\_\mathrm{max}}^2)\geq p^\mathrm{adv}\_\mathrm{max}$ in corollary 3.4. This condition depends on the definition of perturbations and the base images (natural images or uniform noises). For geometry-inspired perturbations (cf. Eq. (2)) on natural images, this condition is transformed into Eq. (3), and for those on uniform noises, Eq. (6).
>
> > it’s unclear whether the coefficient $\lambda$ in the decision boundary is computable
>
> (Theoretical perspective) $\lambda$ is uniquely determined but is not computable (cf. Appendix A). However, the lower and upper bounds are computable, which is enough to derive our theorems.
>
> (Practical perspective) The perturbation Eq. (2) is not computable in practice, as $\lambda$ and the boundary are not computable. Note that this perturbation form was chosen only because it represents the simple results. Moreover, in Appendix D.1, we derived consistent results with those in the main text for the perturbations using the network $f$ directly, which are based on the computable network output and thus practical to implement.
>
> > why cannot generalize to other activations
>
> This is because Theorem 3.3 is applicable only to Leaky ReLU (cf. the proof of Theorem 3.2 in (Frei et al., 2023)).

---

> > ### Comment · Reviewer_doyy · 2023-11-21
> >
> > Dear Author,
> >
> > Thank you for your response. I still have some remaining concerns.
> >
> >
> > Let me rephrase my question regarding the relationship between theorem 3.3 and corollary 3.4. After Theorem 3.3, the author writes "This theorem only requires training data to be nearly-orthogona”. My understanding this refers to $\gamma^3 R_{min}^4/(3N R_{max}^2) ≥ pmax$ in the statement of theorem 3.3, which is an **assumption**. In Corollary 3.4, the author directly replaces it with its adversarial version, meaning the author assumes, the training data, after adding the perturbation, is still nearly orthogonal. I regard this as a very strong assumption. I thought this was proven given the same assumption from the statement of theorem 3.3, but the author responded by saying they just replaced everything with the adversarial counterpart. ``For geometry-inspired perturbations (cf. Eq. (2)) on natural images, this condition is transformed into Eq. (3), and for those on uniform noises, Eq. (6).” Eq(3) and Eq(6) are assumption, can they be proved given the same assumption as in the statement of theorem 3.3? Directly assuming it leads to the point that $p_{max}$ can be negative, which is the next part.
> >
> >
> > In terms of $p_{max}$ being negative, the author responds by ``If the input dimension $d$ is sufficiently larger than $N$, $p_{max}$ will not be negative." This means the statement holds conditioned on high-dimensionality requirements such as d >> n. It covers a rather strict area as not many applications are satisfied. I’m even fine with such an assumption (or called condition), but it has to be crystal clear and properly discussed, instead of buried in some equations.

---

> ### Author Response · Authors · 2023-11-13
> **Response to Reviewer doyy [2/2]**
>
> ### Main results
> > When $N$ goes to infinity, Eq. (3) is saying $p_\mathrm{max}$ is negative?
>
> This is true if $N$ increases infinitely while keeping other variables constant. However, Eq. (3) also indicates $O(d/N)\geq p_\mathrm{max}$. If the input dimension $d$ is sufficiently larger than $N$, $p_\mathrm{max}$ will not be negative.
>
> > estimating the growth rate of T1, T2 by switching the absolute operator and the summation is reasonable?
>
> We acknowledge that this assumption introduced in the natural data scenario (cf. Definition 3.2(a)) is not always reasonable. However, to determine the growth rate of $T_2$, i.e., the second term in Eq. (4), which includes the indeterminable $y_n$, certain assumptions are necessary. While this assumption can be relaxed, for example, to $|\sum\lambda_ny_n\langle x_n,z\rangle|=\Theta(1)$, it inevitably weakens the conclusion of Theorem 4.2.
>
> To circumvent these issues, we employed the noise data scenario (cf. Definition 3.2(b)). This approach allows us to validate the success of learning from perturbations without relying on such unreasonable assumptions (cf. Theorem 4.4). This feasibility stems from the fact that in the noise data scenario the base data (i.e., uniform noises) have well-defined probabilistic characteristics. This differs from the natural data scenario, where the probabilistic nature of the base data (i.e., natural data) and the corresponding labels cannot be defined.
>
> ### Others
> > why the standard boundary only depends on non-robust features in Eq. A157
>
> Under the assumption that standard classifiers focus exclusively on non-robust features, we consider that training classifiers on $\\{(x\_n,y\_n)\\}\_n$ is equivalent to training on $\\{(x^\mathrm{non}\_n,y\_n)\\}\_n$. Thus, by applying Theorem 3.3 to $\\{(x^\mathrm{non}\_n,y\_n)\\}\_n$, we derived Eq. (A157).
>
> > For Figure 2, and other figures,
> > 1. clean accuracy or adversarial robust accuracy?
> > 2. What’s the procedure to generate perturbation?
> > 3. Does it relate to the decision boundary?
> > 4. What’s the point of random vs. deterministic selection?
> > 5. Does the perturbation size align with what indicates by the theorem?
>
> 1. As we do not focus on adversarial training, robust accuracy is not considered in this study.
> 2. Since we cannot practically obtain the decision boundary (cf. the comment above), we employed the PGD attack (Madry et al., 2018). The detailed experimental setup can be found in Appendix H.
> 3. Yes. The accuracy in Figure 2 (left) is the same as that below the boundary in Figure 1 (middle).
> 4. This followed the settings in (Ilyas et al., 2019). Random selection eliminates the correlation between visible features and labels, allowing for a purer observation of the effects of learning from perturbations (cf. the general comment). Experiments with deterministic selection, where visible features are anti-correlated with labels, strongly suggest that invisible perturbations contain data information. Please also refer to Section 3.2 in (Ilyas et al., 2019).
> 5. Yes. In Figure 2 (right), we changed the number of noise samples with a fixed perturbation size to focus on the effect of the sample size variation. This fixed perturbation size was determined by the case with $N^\mathrm{adv}=10,000$. Since the perturbation was controlled by $\epsilon=O(\sqrt{1/N^\mathrm{adv}})$, this fixed size was suitable even for smaller $N^\mathrm{adv}$. The left panel keeps $N^\mathrm{adv}$ constant and varies input dimensions and perturbation sizes according to $O(\sqrt{d})$ to avoid excessively strong L2 perturbations in lower dimensions.

---

> ### Author Response · Authors · 2023-11-21
>
> Thank you for the reviewer's valuable feedback.
>
> > My understanding this refers to $\gamma^3 R^4_\mathrm{min} / (3 N R^2_\mathrm{max}) \geq p_\mathrm{max}$ in the statement of theorem 3.3, which is an assumption. In Corollary 3.4, the author directly replaces it with its adversarial version, meaning the author assumes, the training data, after adding the perturbation, is still nearly orthogonal. I regard this as a very strong assumption.
>
> The reviewer might be misunderstanding that Theorem 4.1 requires not only the orthogonality of the adversarial samples (i.e., Eq. (3)) but also that of the natural samples (i.e., $\gamma^3 R^4_\mathrm{min} / (3 N R^2_\mathrm{max}) \geq p_\mathrm{max}$). We should note that it only assumes the former (i.e., the orthogonality of the adversarial samples). If the reviewer still considers the assumption to be strong, we would like the reviewer to futher clearify it.
>
> > Directly assuming it leads to the point that $p_\mathrm{max}$ can be negative, which is the next part.
>
> Yes, if $N$ goes to infinity (with other variables being constant), $p_\mathrm{max}$ becomes negative, and thus, we cannot use Eq. (4). The core implication of Eq. (3) (and Theorem 4.1) is that if $\epsilon=O(\sqrt{d/N})$ and $p_\mathrm{max}=O(d/N)$ hold, learning from perturbations can be interpreted as Eq. (4). Namely, the decision boundary Eq. (4) consists of two terms, the effects of learning from mislabeled natural data and learning from perturbations, demonstrating the consistency between perturbation learning and standard learning (cf. Theorem 4.2). This is described in Sections 4.2 and 4.3.
>
> Lastly, we have a few minor remarks.
> - The above conditions, $\epsilon=O(\sqrt{d/N})$ and $p_\mathrm{max}=O(d/N)$, contain the case of $N\to\infty$.
> - Eq. (3) does not necessarily require $d \gg n$ (this case was provided as an example).
> - "Assume sufficiently large $N$" in Theorem 4.1 might lead misunderstanding. The more accurate representation is described as "If $N>C^2/R^2_\mathrm{min}$ and Eq. (3) with C=..., ...".
> - Please refer to Theorem B.2 for $N\leq C^2/R^2_\mathrm{min}$.

---

### Official Review · Reviewer_qaZD · 2023-11-01

**Soundness:** 3 good
**Presentation:** 3 good
**Contribution:** 2 fair
**Rating:** 6
**Confidence:** 3

**Summary:**

The paper investigates learning from adversarial perturbations, which is motivated by the observation that neural networks trained on adversarial examples can generalize to natural test data. To that end, recent results on the implicit bias of the gradient flow of training single-hidden-layer networks are employed to show that, under certain assumptions, the decision boundaries of natural examples and adversarial examples align with each other.

**Strengths:**

- The idea to apply the implicit bias results to learning from adversarial perturbations is novel.
- The theoretical results admit interesting and relevant interpretations (eg. effect of learning misslabled data vs perturbation data).

**Weaknesses:**

- It is unclear what the motivation for studying the uniform perturbation model is.
- Overall, the investigated model seems to be quite simple and theoretical assumptions restrictive (see also questions below).
- The text is rather densely written (e.g. subsection 4.2 and 4.3), which makes understanding main statements and insights rather difficult (e.g. Theorem 4.2).

**Questions:**

- In the limitations paragraph it is mentioned that model is simple (one-hidden layer network and orthogonal training data). Could the authors extend their discussion on the limitations (e.g. attack model), and if possible, mention ideas on how to overcome the limitations in the future (e.g. $\epsilon \in O(d/N)$)?
- The paper expands on established settings on 'learning from adversarial perturbations' (Definition 1.1. and Definition 3.2) by also discussing perturbations on uniform noises. In this 'perturbation on uniform noise' setting, i.e. $x_n^{adv} = X_n + \eta_n$, with $X_n$ being uniform noise instead of a natural data point. What is the motivation for studying this setting? Why not considering $x^{adv}_n = x_n + X_n + \eta_n$ (with $x_n$ the actual data points), which would be a more practical setting?

---

> ### Author Response · Authors · 2023-11-13
> **Response to Reviewer qaZD [1/1]**
>
> We would like to thank the reviewer’s careful reading.
>
> To address the concerns of some reviewers who might misunderstand our work as related to adversarial training, we have provided a detailed clarification in the general comment. Please also refer to it.
>
> ### Noise data scenario
> > what the motivation for studying the uniform perturbation model
>
> We consider the noise data scenario (cf. Definition 3.2(b)) with two motivations.
>
> First, this is to prevent unintentional leakage of useful features. In the natural data scenario (cf. Definition 3.2(a)), we labeled adversarial examples with incorrect classes. However, there remains the possibility that these images contain features related to the incorrectly assigned labels. For example, a frog image labeled as a horse may contain horses in the background. As long as we consider natural images, it is inevitable that unintentional alignment between images and labels will occur, causing classifiers to generalize from leaked features rather than perturbations. In contrast, uniform noises lack natural object features, allowing us to explore the purer dynamics of learning from perturbations.
>
> Second, this is to validate the success of learning from perturbations under weaker conditions than in the natural data scenario. In the natural data scenario, the probabilistic nature of the base data (i.e., natural data) cannot be defined. For example, the distribution of frog images cannot be precisely determined. This uncertainty requires us to impose the assumption $|\sum\lambda_ny_n\langle x_n,z\rangle|=\Theta(g(N,d))$ if $\sum\lambda_n|\langle x_n,z\rangle|=\Theta(g(N,d))$. In the noise data scenario, thanks to the well-defined probabilistic properties of the base data (i.e., uniform noises), we can justify the success of perturbation learning under more reasonable assumptions (cf. Theorem 4.4).
>
> > Why not considering $x^\mathrm{adv}_n=x_n+X_n+\eta_n$, which would be a more practical?
>
> First, it is difficult to define what is "practical" in the context of learning from perturbations. This experiment aims to understand the origins of adversarial examples and the information contained in adversarial perturbations. From this perspective, both the natural and noise data scenarios align with the objectives, and we cannot determine the superiority. Similarly, we cannot determine which is better, $x^\mathrm{adv}_n=X_n+\eta_n$ or $x^\mathrm{adv}_n=x_n+X_n+\eta_n$. Please also refer to the general comment.
>
> From the perspective of a tractable theoretical analysis, the scenario where perturbations are on solely uniform noises can provide a more theoretically manageable formulation of learning from perturbations, as described in the comment above. Thus, we choosed $x^\mathrm{adv}_n=X_n+\eta_n$ rather than $x^\mathrm{adv}_n=x_n+X_n+\eta_n$.
>
> ### Assumptions on networks and training data
> > the investigated model seems to be quite simple and theoretical assumptions restrictive
> >
> > it is mentioned that model is simple (one-hidden layer network and orthogonal training data). Could the authors extend their discussion on the limitations (e.g. attack model), and if possible, mention ideas on how to overcome the limitations in the future (e.g. $\epsilon\in O(d/N)$)?
>
> If the reviewer regard "attack model" as the form of perturbation, we have explored three additional forms of perturbation in Appendix D that were not discussed in the main text in detail. Please refer to them.
>
> The constraints of using a one-hidden-layer neural network, orthogonal training, and the resulting $\epsilon\in O(d/N)$ are based on the theoretical framework, Theorem 3.3 (Frei et al., 2023). To remove these constraints, we would need to extend Theorem 3.3 in the context of the implicit bias of gradient descent rather than learning from adversarial perturbations. Previous research [1,2] has demonstrated the implicit bias of unbiased deep ReLU networks under more relaxed dataset assumptions. Thus, we believe that these results could serve as a starting point to justify learning from perturbations under more relaxed assumptions.
>
> [1] Kaifeng Lyu and Jian Li. Gradient descent maximizes the margin of homogeneous neural networks. In ICLR. 2020.
>
> [2] Ziwei Ji and Matus Telgarsky. Directional convergence and alignment in deep learning. In NeurIPS. 2020.
>
> ### Others
> > The text is rather densely written, which makes understanding main statements and insights rather difficult
>
> We acknowledge that due to space constraints, some sections of our paper, particularly the explanation following Theorem 4.1, have become overly dense. We plan to address this by eliminating some redundant expressions.

---

> > ### Comment · Reviewer_qaZD · 2023-11-22
> >
> > Thanks to the authors for addressing my questions, particularly for elaborating on the motivation for the uniform noise model. I'll keep the positive score.

---

### Official Review · Reviewer_K8gK · 2023-11-04

**Soundness:** 2 fair
**Presentation:** 3 good
**Contribution:** 2 fair
**Rating:** 5
**Confidence:** 3

**Summary:**

This paper theoretically studies the learning of adversarial perturbations generated by geometric-inspired attacks with respect to a one-hidden-layer neural network and orthogonal training data. Specifically, two settings are considered for learning from adversarial perturbation: 1) perturbation on natural data and 2) perturbation on uniform noise. Theoretically, they show that the decision boundaries of models learned from adversarial perturbation are consistent with clean decision boundaries, except for examples strongly correlated with many natural samples.

**Strengths:**

Theoretical understanding of how neural networks learn in the presence of adversarial perturbations is a challenging but important task. The paper is technically solid, with clearly presented assumptions and theoretical results. Theoretical results are accompanied by adequate discussions and high-level explanations of the proof idea. The considered setting where adversarial perturbations are added to uniform/Gaussian noise is new, which nicely supports the paper's key argument that generated adversarial perturbations contain abundant information about the underlying data structure.

**Weaknesses:**

While I appreciate the theoretical nature of this paper, the motivation and some considered settings for studying adversarial perturbations in the context of learning need to be more convincing from my perspective. In the abstract, the paper claims that the phenomenon “neural networks trained on mislabeled samples with adversarial perturbations can generalize to natural test data” is counter-intuitive.  I do not understand why this is a counter-intuitive phenomenon and how this phenomenon motivates the problem of learning from adversarial perturbations. If I understand correctly, the adversarial perturbations are generated with respect to some pre-trained neural networks and are not necessarily “mislabeled” (unless the perturbation size is large). In addition, the definition of mislabeled data needs to be clarified because, typically, human is considered the underlying ground truth in judging whether an input image is mislabeled. However, you use a pre-trained neural network as the reference model for defining whether an input is mislabeled. I am also confused by the statement in the introduction, “A classifier, trained on natural samples with adversarial perturbations labeled by such incorrect classes, can generalize to unperturbed data,” which I believe is not the key message that Ilyas et al. (2019) wants to convey. From my perspective, the learnability of adversarially perturbated natural data is because the added perturbation is not large enough to destroy the useful generalizable features of the natural data. I hope the authors can clarify my concerns in the rebuttal on why the learnability from the added adversarial perturbations is worth studying, which will largely decide my judgement on whether this paper reaches the ICLR acceptance bar or not.

Moreover, the imposed assumptions of orthogonal training data and the use of geometric-inspired attacks need to be better discussed. What is the mathematical definition of orthogonal training data? I do not understand the claim that the orthogonality of training data is a common property of high-dimensional data. Does this claim hold for any high-dimensional data distribution or specific distributions? In addition, I believe geometric-inspired attacks are less commonly studied in the adversarial ML community. How will the conclusion of your theoretical findings change if the perturbations are generated by PGD attacks (Madry et al., 2018)?

**Questions:**

Apart from the above questions, I would like to know how the degree of orthogonality of training data affects your theoretical results. If the underlying data distribution only partially satisfies this assumption, can you use similar proof techniques to characterize the learning from adversarial perturbations?


============ Post Rebuttal Comments ===========

I appreciate the authors' feedback. In summary, I will keep my score unchanged. I think the authors misunderstood my major concern, which has nothing to do with adversarial training or adversarial robustness. What is unconvinced from my perspective is the motivation for studying the setting of learning from adversarial perturbations. I do not find a convincing answer in the rebuttal, and I can hardly understand the benefits gained from studying this problem. The final conclusion drawn from the paper is that standard learning from adversarially perturbed inputs (with possibly incorrect labels) yields consistent predictions. Because of the assumptions on how the adversarial perturbations are generated, I am not sure if the paper's theoretical evidence is sufficient to support such general conclusion. In my opinion, the way adversarial perturbations are generated and the perturbation budget will highly affect the final conclusion.

---

> ### Author Response · Authors · 2023-11-13
> **Response to Reviewer K8gK [1/1]**
>
> We appreciate the reviewer's fruitful suggestions and questions.
>
> **First, we would like to clarify that our research is NOT related to adversarial training. Please kindly refer to the general comment.**
>
> ### Counterintuitiveness of learning from adversarial perturbations
> > adversarial perturbations are not necessarily mislabeled
>
> This is true if it implies that adversarial examples may not always deceive a network.
>
> > the definition of mislabeled data needs to be clarified ... you use a pre-trained network as the reference model for defining whether an input is mislabeled.
>
> We do **NOT** use a pre-trained network as the reference model. We consistently regard an image as mislabeled when it has a label different from the human prediction, i.e., an originally annotated label. For example, a natural image of a frog labeled as a horse is mislabeled. Similarly, an adversarial image of a frog (an image of a frog that is manipulated to lead classifier predictions to a horse but still seems to be a frog to humans) labeled as a horse is considered mislabeled.
>
> > the learnability of adversarially perturbated natural data is because the added perturbation is not large enough to destroy the useful generalizable features of the natural data.
> >
> > why this is a counter-intuitive
>
> The reviewer might have misunderstood our focus. Please refer to the general comment.
>
> If the perturbation size is not excessively large, it does not destroy the useful generalizable features of natural data. **Thus, learning from perturbations is counterintuitive.** Recall that our experiments, and Ilyas's, labeled images contrary to these generalizable features. **Since perturbations do not destroy generalizable features, these labels appear entirely nonsensical.** For example, frog images were labeled as horses. In fact, classifiers trained on this dataset classify frog images in this set as horses. However, they classify frog images in *test sets* as frogs, i.e. they achieve a counterintuitively high accuracy on test datasets.
>
> > how this phenomenon motivates the problem
>
> As previously described, classifiers can achieve high test accuracy on standard datasets, even when trained on mislabeled adversarial images. This counterintuitive phenomenon led Ilyas et al. to propose a hypothesis: adversarial perturbations may contain features of the mislabeled class that are imperceptible to humans, and classifiers can learn data structures from them.
>
> This intriguing hypothesis can explain not only why learning from perturbations succeeds but also why adversarial images can deceive classifiers: classifiers might respond to invisible horse features in adversarial frog images and thus misclassify frogs as horses. Moreover, this hypothesis accounts for other puzzling phenomena of adversarial images. Please refer to Section 2.
>
> Although this hypothesis seems reasonable, there has been no theoretical justification for its basis, i.e., the success of learning from adversarial perturbations. In this study, we are the first to provide the theoretical understanding and justification for it.
>
> ### Orthogonality of training data
> > 1. mathematical definition of orthogonal training data
> > 2. how the degree of orthogonality of training data affects your theoretical results
> > 3. orthogonality of training data is a common property of high-dimensional data? Does this claim hold for any high-dimensional data or specific distributions?
>
> 1. The orthogonality condition of training data is defined in Theorem 3.3 as $\gamma^3R^4_\mathrm{min}/(3NR^2_\mathrm{max})\geq p_\mathrm{max}$ and extended to adversarial datasets as $\gamma^3{R^\mathrm{adv}\_\mathrm{min}}^4/(3N^\mathrm{adv}{R^\mathrm{adv}\_\mathrm{max}}^2)\geq p^\mathrm{adv}\_\mathrm{max}$. Due to $R^\mathrm{adv}\_\mathrm{min}$, $R^\mathrm{adv}\_\mathrm{max}$, and $p^\mathrm{adv}\_\mathrm{max}$, the latter condition depends on the definition of perturbations and the base images (natural images or uniform noises). For geometry-inspired perturbations on natural images, this condition is transformed into Eq. (3), and for those on uniform noises, into Eq. (6).
> 2. Our theoretical results hold even if the training data are not perfectly orthogonal, i.e., $p^\mathrm{max}>0$.
> 3. Statistically, two high-dimensional random vectors (e.g., with i.i.d. Gaussian entries) are nearly orthogonal. Although natural images are not random tensors, it has been empirically observed that they tend to be nearly orthogonal to each other. Please also refer to [1].
>
> [1] Sven-Ake Wegner. Lecture Notes on High-Dimensional Data.
>
> ### Perturbation form
> > How will the conclusion of your theoretical findings change if the perturbations are generated by PGD attacks (Madry et al., 2018)?
>
> In Appendix D, for different forms of perturbations, we derived the consistent results with those in the main text. The perturbations generated by PGD attacks are discussed in Appendix D.1. Please kindly refer to it.

---

### Author Response · Authors · 2023-11-13
**General comment [1/2]**

We appreciate the reviewers' constructive comments. Please kindly refer to the response to each reviewer. If our answers require further explanation or clarification, we are more than willing to provide them.

---

**Our primary concern is a possible misunderstanding among the reviewers regarding the focus of our research. We clarify the following points:**

### Our focus is NOT adversarial training.

### Our research is NOT related to adversarial training.

Our study explores the counterintuitive phenomenon of learning generalizable features from adversarial perturbations, distinctly different from adversarial training, which aims to train classifiers to be robust against adversarial attacks. **Other than the fact that we also use adversarial examples, there is no overlap between adversarial training and our work.** We have defined our concept of learning from adversarial perturbations twice (cf. Definition 1.1 and 3.2) and would like to highlight this again.

**Learning from adversarial perturbations (Ilyas et al., 2019)** investigates the characteristic of adversarial examples and perturbations. The primary focus of their experiment is not on practical applications such as enhancing robustness. Given a dataset $\mathcal{D}:=\\{(x_n,y_n)\\}$ and a trained classifier $f$, we create adversarial examples $x^\mathrm{adv}_n$ such that $f$'s prediction becomes $y^\mathrm{adv}_n\ne y_n$, i.e., $f(x_n)=y_n$ and $f(x^\mathrm{adv}_n)=y^\mathrm{adv}_n$, thereby forming a new dataset $\mathcal{D}^\mathrm{adv}:=\\{(x^\mathrm{adv}_n,y^\mathrm{adv}_n)\\}$. These adversarial examples $x^\mathrm{adv}_n$ appear indistinguishable from natural images $x$ to the human eye, making $\mathcal{D}^\mathrm{adv}$ seemingly mislabeled. However, a classifier $g$ trained from scratch on $\mathcal{D}^\mathrm{adv}$ surprisingly yields high test accuracy on standard datasets, which are correctly labeled from a human perspective.

**Adversarial training (Madry et al., 2018)** aims to enhance a classifier's robustness. Given a dataset $\mathcal{D}:=\\{(x_n,y_n)\\}$ and an initialized classifier $f$, adversarial training updates $f$'s parameters to minimize the losses over $\\{(x^\mathrm{adv}_n,y_n)\\}$ in each training iteration.

**(Ilyas et at., 2019) and (Madry et al., 2018) are two distinct works with different objectives and procedures.** Our paper seeks to provide a theoretical explanation for the success of learning from adversarial perturbations. This is a novel endeavor distinct from extensive studies on adversarial training. To the best of our knowledge, the reasons behind the success of learning from adversarial perturbations have not been theoretically elucidated before.

**We kindly ask all reviewers, *regardless of whether they initially lean towards acceptance or rejection*, to reconsider their evaluations in light of this clarification.**

---

**Why does learning from perturbations succeed? (supplement)**

We define the training, test, and adversarial dataset as:
$$
\mathcal{D}:=\\{(\mathrm{frog\~img},\mathrm{frog}),
(\mathrm{horse\~img},\mathrm{horse}),
(\mathrm{cat\~img},\mathrm{cat})\\},
$$
$$
\mathcal{D}^\mathrm{test}:=\\{(\mathrm{frog\~img*},\mathrm{frog}),
(\mathrm{horse\~img*},\mathrm{horse}),
(\mathrm{cat\~img*},\mathrm{cat})\\},
$$
$$
\mathcal{D}^\mathrm{adv}:=\\{(\mathrm{frog\~img+},\mathrm{horse}),
(\mathrm{horse\~img+},\mathrm{cat}),
(\mathrm{cat\~img+},\mathrm{frog})\\}.
$$
A trained classifier $f$ can predict the correct labels for natural images (e.g., $f(\mathrm{frog\~img})=\mathrm{frog}$ and $f(\mathrm{frog\~img*})=\mathrm{frog}$) but not for adversarial examples (e.g., $f(\mathrm{frog\~img+})=\mathrm{horse}$). Note that $\mathrm{frog\~img+}$ still seems to be a frog to humans. Counterintuitively, another classifier $g$ trained from scratch on $\mathcal{D}^\mathrm{adv}$ can correctly predict the classes of the natural test images in $\mathcal{D}^\mathrm{test}$ (e.g., $g(\mathrm{frog\~img*})=\mathrm{frog}$).

Why? **Ilyas et al. (2019) hypothesized that adversarial perturbations contain imperceivable data features to humans**. For example, $\mathrm{frog\~img+}$ contains not only visible frog features but also invisible horse features.

Let us consider training on the following adversarial dataset $\mathcal{D}^\mathrm{adv}$. Through training, a classifier $g$ ignores visible features that are uncorrelated with labels and learns invisible features that are correlated with labels. Since natural test images contain invisible features of the corresponding classes (e.g., a frog image contains invisible frog features), this classifier can provide correct predictions for them. As a result, the classifier trained on the dataset that seems completely mislabeled to humans achieves high accuracy on the natural test dataset.

|Data index|Visible features|Invisible features|Label|
|-|-|-|-|
|1|frog|horse|horse|
|2|horse|cat|cat|
|3|cat|frog|frog|
|4|frog|cat|cat|
|5|horse|frog|frog|
|6|cat|horse|horse|
|...|...|...|...|

---

> ### Author Response · Authors · 2023-11-13
> **General comment [2/2]**
>
> **Our contributions (supplement)**
>
> With the above points in mind, we summarize our contributions as follows:
>
> - Revealing how adversarial perturbations represent natural data features (cf. Section 4.1).
> - Justification of learning from perturbations in the natural data scenario (cf. Definition 3.2(a)) where perturbations are superposed on natural data (cf. Theorems 4.1 and 4.2).
> - Justification of learning from perturbations in the noise data scenario (cf. Definition 3.2(b)) where perturbations are superposed on uniform noise (cf. Theorems 4.3 and 4.4, and Corollary 4.5).
>
> For a fair review, we restate that our results were derived under the assumptions of a one-hidden-layer neural network (cf. Section 3.1) and orthogonal training data (cf. Theorem 3.3 and Corollary 3.4).
>
> Here, we briefly describe the contributions with the background mentioned above.
>
> **Contribution 1. Adversarial perturbations as data featurse (supplement)**
>
> We showed that adversarial perturbations are represented as the weighted sum of training samples, indicating abundant data information within the perturbations. This result theoretically supports the feature hypothesis by Ilyas et al. that adversarial perturbations contain data features. Since the feature hypothesis has the potential to explain many puzzling phenomena of adversarial examples (cf. Section 2), it is important to construct the theoretical basis for this hypothesis.
>
> **Contribution 2. Justification of learning from perturbations in natural data scenario (supplement)**
>
> We provided the first theoretical explanation for the success of learning from perturbations (cf. Theorems 4.1 and 4.3). In summary, these theorems claim that classifiers trained on mislabeled adversarial examples $\mathcal{D}^\mathrm{adv}$ produce the consistent predictions with standard classifiers except for the training data points. Since test datasets generally do not contain data similar to the training data, our results effectively explain the counterintuitively high accuracy of perturbation learning on test datasets.
>
> In addition, in response to some reviewers' concerns, we discussed a wide variety of perturbation forms in Appendix D. In particular, we obtained consistent results with the main text's conclusion for gradient-based perturbations using the outputs of the one-hidden-layer network and exponential loss, which has been employed in many experimental and theoretical situations.
>
> **Contribution 3. Justification of learning from perturbations in noise data scenario (supplement)**
>
> We justified the success of perturbation learning not only in the natural data scenario but also in the scenario where perturbations are superposed on noises. This scenario was developed to overcome two limitations of the natural data scenario: feature leakage and some assumptions due to the uncertainty of natural data. For more details, please refer to our response to Reviewer qaZD.
>
> **Other contributions (supplement)**
>
> We also derived the same results for L0 perturbations, i.e., sparse perturbations. The fact that even sparse perturbations, which seem to lack natural data structures, encapsulate data information and allow networks to generalize to natural data is interesting both from a theoretical (cf. Section 4.4 and Appendix D.2) and experimental (cf. Section 5.2 and Table 1) perspectives.
>
> Furthermore, we experimentally confirmed the precise alignment of the decision boundaries across a wide range of perturbations, strongly supporting our theoretical findings. Please refer to Figures 1 and A4--A9.

---

### Author Response · Authors · 2023-11-20
**The discussion period will be closed soon**

Since the discussion period will be closed soon, we would like to kindly ask all the AC and reviewers to read our feedback comments and have further discussions.

Thank you in advance.

---

### Meta-Review · Area_Chair_7o32 · 2023-12-25

**Metareview:**

【Summary of Contributions】 This paper explores why adversarial perturbations can help improve benign test accuracy. There was a hypothesis that the adversarial perturbations contain information of other data points, but the mathematical mechanism was not clear. This paper used a simple model of single-hidden-layer networks and orthogonal training data, and showed that since the adversarial perturbations can be represented as weighted combinations of training data, models trained on adversarial examples with incorrect labels can still effectively learn a decision boundary that is similar to the boundary for the original natural data distribution. The paper's theoretical claims are supported by experiments showing similar decision boundaries when training on mislabeled data and natural data on MNIST.

【Comments on theoretical contributions】A few reviewers find the setting of the paper quite restricted (orthogonal trainign data, 1-hidden-layer net with 2nd layer frozen), but they also acknowledge that for a theoretical investigation such limitation is understandable. I agree with those reviewers who think the paper has revealed some theoretical insight on learning from adversarial perturbations.

   I read the paper, and find that the explicit expressions of perturbations in (2) critical for the tractability of the analysis. For other attacks like FGSM, or more complicatd neural networks, such a closed-form expression is impossible. It is a limitation of the current framework, but also shows that it is nontrivial to identify a setting that is tractable for analysis.

【Confusion of two settings】  Some reviewers confused the problem of learning from adversarial perturbations in this paper and the problem of adversarial training. I think this is partially because some reviewers are not familar with Ilyas et at., 2019, and partially because of the presentation of the paper. There are a few possible places to improve the presentation so as to reduce the chance of confusion:

   i) This paper did not introduce Ilyas et at., 2019 in more detail. The original paper Ilyas et at., 2019 used the title "adversarial examples are not bugs, they are features" to highlight the unusual finding of the paper. But the tone of this paper reads like it is handling the more usual "adversarial attack and defense" setting.

   ii) The first sentence of the abstract "It is not fully understood why adversarial examples can deceive neural networks and transfer between different networks" is a bit misleading. It reads like discussing the problem of "why adversarial examples can deceive different architectures, e.g., they can deceive ResNet and VGG simultaneously". The problem of transferability is not the focus of this paper, so the sentence should be modified.

  iii) In the abstract, sentences like "trained on mislabeled samples with these perturbations can generalize to natural test data" and "various adversarial perturbations ... encapsulate abundant data information and facilitate network generalization" can be understood as both "learning from adversarial perturbation" and "adversarial training". For adversarial training setting, it is also true that the trained network can generalize to test data with perturbations, which can be understood as "generalize to natural test data". Therefore, the confusion is partially because these sentences did not distinguish these two settings.

  iv) In fact, "learning from adversarial perturbations" itself can be understood as "a network trained on adversarial perturbations on training data (with correct labels) can generalize to classify adversarial perturbations on test data", which is adversarial training.  I think the term "learning from adversarial perturbations" did not highlight the major difference from the so-called "adversarial training", which is "learning from changed labels on perturbed data" v.s. "learning from original labels on perturbed data". It may be nice to add a table to compare these two settings.

【Recommendation】Overall, I recommend acceptance, but also suggest the authors to modify the presentation to reduce the chance of confusion on "learning from adversarial perturbations" and "adversarial training".

**Justification For Why Not Higher Score:**

The model is rather limited: orthogonal training data and 1-hidden-layer net.

**Justification For Why Not Lower Score:**

It reveals some insight on the curious phenomenon of "adversarial examples are features, not just bugs".

---

### Decision · Program_Chairs · 2024-01-16

Accept (poster)